# Karyopherins remodel the dynamic organization of the nuclear pore complex transport barrier

Toshiya Kozai[1,2,18], Javier Fernandez-Martinez[3,4,5,18], Larisa E. Kapinos [1], Paola Gallardo [6,15], Trevor van Eeuwen [3], Martin Saladin[1], Roi Eliasian[7], Adam Mazur[1], Wenzhu Zhang[8], Jeremy Tempkin[9,10,11,16], Radhakrishnan Panatala[1,17], Maria Delgado-Izquierdo [5], Raul Escribano-Marin[5], Qingzhou Feng[12,13], Chenxiang Lin [12,13,14], Andrej Sali [9,10,11], Brian T. Chait [8], Barak Raveh[7], Liesbeth M. Veenhoff [6], Michael P. Rout [3] ✉ & Roderick Y. H. Lim [1,2] ✉

Nuclear pore complexes (NPCs) mediate selective exchange of macromolecules between the nucleus and cytoplasm, but the organization of their transport barrier has been a matter of debate. Here we used high-speed atomic force microscopy, complemented with orthogonal in vitro and in vivo approaches, to probe the dynamic behaviour of the NPC central channel at millisecond resolution. We found that nuclear transport factors dynamically remodel intrinsically disordered phenylalanine-glycine (FG) domains tethered within the NPC channel, partitioning the barrier into two zones: a rapidly fluctuating annular region and a highly mobile central plug. Increased FG-repeat density in mutant NPCs dampened barrier dynamics and impaired transport. Notably, NPC-like behaviour was recapitulated in DNA origami nanopores bearing transport factors and correctly tethered FG domains but not in in vitro FG hydrogels. Thus, the rotationally symmetric architecture of NPCs supports a nanoscopic barrier organization that contrasts with many of the bulk properties of in vitro FG-domain assemblies.

Intrinsically disordered proteins (IDPs) display a repertoire of dynamic behaviours that are crucial to diverse biological processes[1]. Without a stable structure IDPs can exist as an ensemble of multifaceted and interconvertible conformations, enabling context-dependent behaviours[2]. However, capturing the in situ behaviour of IDPs at length and time scales inherent to their biological functions remains a major challenge[3]. A notable example is the nuclear pore complex (NPC)[4–9], which regulates the rapid exchange of signal-specific cargoes across the nuclear envelope during nucleocytoplasmic transport[10,11]. Each NPC is assembled from multiple proteins termed nucleoporins (Nups) that form an eightfold-rotationally symmetric core scaffold surrounding an approximately 50-nm-wide conduit termed the central transporter (CNT). A subset of different Nups possess large intrinsically disordered domains adorned with FG repeats projecting from anchor sites on the core scaffold into the CNT[4,5,7]. These so-called FG domains can be categorized into less cohesive 'FxFGs' and more cohesive 'GLFGs', differing in their in vitro characteristics[12] (Supplementary Table 1). FxFG domains are asymmetrically positioned near the nuclear or cytoplasmic peripheries of the NPC, whereas GLFG domains are symmetrically distributed across the CNT midplane (Supplementary Fig. 1). Together, the FG domains create a transport barrier that obstructs non-specific molecules[13,14] while their FG repeats engage in multivalent interactions

with cargo-carrying nuclear transport factors termed karyopherins (Kaps)[15–17]. These interactions mediate NPC transport selectivity and speed, with transit times of up to 200 ms for messenger RNA and approximately 10 ms for proteins[18–22].

Despite having mapped the core scaffold structure[4–9] and the trajectories of individual Kap-cargo complexes[18,21] with nanometre precision, the organization of the NPC transport barrier remains obscure due to its dynamic and disordered nature[23–25]. This issue is of fundamental interest given that NPC dysfunction is linked to several diseases[10] including cancer and neurodegeneration[26]. NPC transport models are largely informed by in vitro studies of FG-domain assemblies and in particular polymer brushes[27–30], liquid-like biomolecular condensates[31,32] and macroscopic hydrogels[33–38]. Surface-tethered polymer chains, including those with free, sticky end-groups[39], can elongate into nanoscopic polymer brushes at sufficiently high surface coverage to repel passive macromolecules from surfaces[40,41] and nanopores[42]. Biomolecular condensates form through phase separation under specific thermodynamic conditions, driving a density transition that organizes macromolecules into dense dynamic liquid-like assemblies that exist in a state of continuous exchange with a dilute phase[43–45]. In contrast, macroscopic hydrogels describe rigid percolated networks with amyloid-like characteristics[46–49]. Consequently, the composition, dynamics and transport functions of the NPC transport barrier remain controversial. Another question is how Kap enrichment modulates FG-domain conformation[7,16,50–53] to enhance NPC transport barrier function, as Kap depletion compromises its integrity[52]. The identity and function of a dense amorphous structure within the CNT[54–57], termed the central plug (CP), also remain unsettled despite suggestions that it comprises a crosslinked meshwork of FG domains[58].

Here we used high-speed atomic force microscopy (HS-AFM)[59] and line scanning (HS-AFM-LS)[60], complemented by orthogonal in vivo and in vitro studies, to elucidate the in situ dynamic organization of the NPC transport barrier of the budding yeast *Saccharomyces cerevisiae* with a temporal resolution of approximately 1 ms, which closely match transport time scales[19,20,22]. We previously used HS-AFM to visualize dynamic FG-domain behaviour within NPCs using spread *Xenopus laevis* oocyte nuclear envelopes[61,62]. In the current work we investigated the CP identity, the impact of transport barrier dynamics on in vivo NPC function and corroborated these attributes using in vitro FG-domain assemblies and in vivo assays. We show that Kaps remodel in situ FG-domain behaviour by stabilizing their dynamic fluctuations at the pore centre, mediating the formation of a mobile cluster that accounts for the nanosized CP. Mutant NPCs with increased FG content showed reduced barrier dynamics and impaired nuclear localization signal (NLS)-cargo transport in vivo despite elevated Kap95 enrichment at the pore, suggesting its sensitivity to FG composition and an overtightening of the transport barrier. In contrast, overexpressed Nup FG domains (NupFGs) did not readily concentrate at NPCs in vivo, indicating poor coalescence with the transport barrier. To benchmark our findings against in vitro NPC models, we examined FG-domain hydrogels and found that they formed static amyloid-like aggregates, revealing irregularly shaped holes (approximately 50 nm in size), yielding a sponge-like architecture that contrasts sharply with the dynamic nanostructures of native NPCs. In contrast, DNA origami nanopores bearing scaffold-tethered FG domains[63–67] successfully recapitulated NPC-like behaviour, including the formation of CP-like nanoclusters and dynamic barrier properties.

## Results

### High-speed atomic force microscopy analysis of isolated yeast NPCs

To enable gentle tapping on specimens in aqueous environments with forces below 50 pN, HS-AFM uses a nanometre-sharp tip oscillating at about 0.5 MHz with a 10% attenuation of its 2–3 nm free oscillation amplitude[68]. In this way HS-AFM is adept at capturing dynamic biomolecular movements at scan rates of approximately 100 ms per image

frame, achieving nanometre resolution in the $xy$ plane and approximately 0.1 nm precision in the $z$ direction[59]. At these extreme frequencies, the impulse or energy transferred to the specimen is negligibly small, ensuring that mechanical disturbances to delicate biological specimens, such as individual IDP molecules, are minimized[69]. This contrasts sharply with conventional AFM techniques, as the impulse applied to NPCs by HS-AFM in the current work is at least four orders of magnitude smaller than in previous AFM studies[70,71].

Isolated native yeast (wild type, WT) NPCs effectively maintain the in vivo CNT environment and are well-suited for direct examination of the transport barrier[4,5,7]. They represent the contracted form of the normal range of NPC diameters, retain an intact configuration of FG domains and bind Kaps[72]. After isolation, essentially all WT NPCs bore a dense CP (termed +CP WT NPCs) measuring 26.4 ± 8.0 nm in diameter (Fig. 1a and Extended Data Fig. 1). Central plugs have been identified in both isolated and in situ NPCs[4,5], and their presence in NPCs lacking the core nuclear basket proteins Mlp1 and Mlp2 (ref. 73) confirms that the CP and the nuclear basket are distinct[74] (Extended Data Fig. 1). The overall size and eightfold rotational symmetry of +CP WT NPCs were also analysed using negative-stain electron microscopy and were found to be consistent with our published cryo-electron microscopy maps[4,5,7] (Extended Data Fig. 1). It was, however, not possible to determine their nucleocytoplasmic orientation as the nuclear basket is not a structurally well-defined stable feature in a majority of NPCs[4,5,7,74]. Nevertheless, FG-repeat asymmetry is not essential for nucleocytoplasmic transport in vivo as FxFG domains can be either inverted[75] or deleted[76] without disrupting NPC function. Therefore, NPC orientation should not influence our analysis, which focuses on the symmetric CNT core predominantly occupied by the GLFG domains[4,5,7] (Supplementary Fig. 1).

Finer details were captured by zooming into the CNT of +CP WT NPCs at 150 ms per frame and an imaging force of 30–40 pN ('zoomed-in image'; Fig. 1b). This revealed FG domain fluctuations that radiated from the core scaffold into the CNT where they collided into, fused with and then detached from the CP, exposing transient voids in the protein density (Supplementary Video 1). Such behaviour evokes highly dynamic liquid-like behaviour that is consistent with multivalent interactions between the Kaps and FG domains[15–17] being continuously broken and formed. In line with this view, 60% of isolated NPCs lacked detectable CPs after 72 h in buffer (−CP WT NPCs) but the average diameters of +CP and −CP WT NPCs remained unchanged (Fig. 1c and Extended Data Fig. 1). Closer examination of −CP WT NPCs revealed that extended FG domain fluctuations intermingled within the CNT but lacked significant cohesion when the CP was absent, suggesting a more open state (Fig. 1d and Supplementary Video 2). This indicates that inter-FG-domain interactions and dynamic FG-domain fluctuations are not mutually exclusive behaviours.

### Capturing NPC transport barrier dynamics at millisecond time scales

HS-AFM captures each pixel sequentially in a rastering 'zig-zag' pattern, which means that different portions of an image correspond to slightly different time points. Thus, changes in molecular conformations or positions occurring at or above the HS-AFM imaging speeds can lead to data asynchronicity and image distortions[77]. To examine this effect, we performed Brownian dynamics simulations of Nsp1 FG domains (Nsp1FG) in a simplified 22-nm-diameter toroidal nanopore, which is distinct from the NPC, and computationally sampled it in a manner mimicking HS-AFM (Supplementary Fig. 2 and Supplementary Video 3). The simulated images displayed continuous features along horizontal lines, reflecting the capture of the collective motion of Nsp1FG along the HS-AFM fast-scan axis ($x$ axis). In contrast, image data along vertical lines seemed more discontinuous due to asynchronicity along the HS-AFM slow-scan axis ($y$ axis). This demonstrates that HS-AFM more accurately captures dynamic molecular movements within individual horizontal lines than across them.

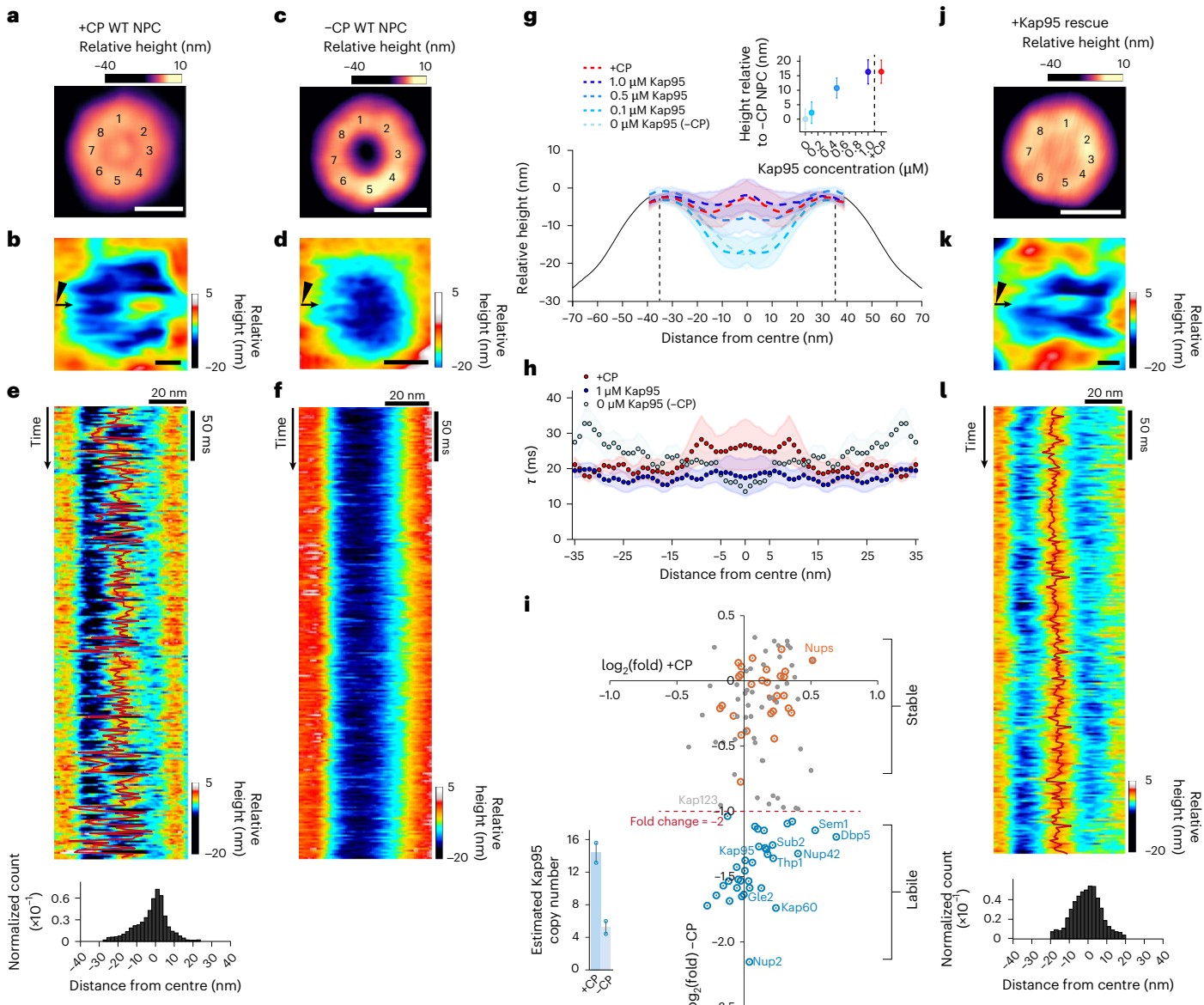

**Fig. 1 | Direct visualization of the in situ NPC transport barrier by HS-AFM.**
**a**, Averaged eightfold symmetric structure (numbered) of a +CP WT NPC.
**b**, Zoomed-in image of a +CP WT NPC showing a CP surrounded by FG-domain fluctuations within the CNT. **c**, Averaged −CP WT NPC resolved by HS-AFM. **a,c**, ≥3 independent isolations, $n$ = 15 NPCs. **d**, Zoomed-in image of a −CP WT NPC revealing FG-domain fluctuations within the CNT. **b,d**, HS-AFM-LS data were collected from left to right (small arrow denotes scan direction). Large tilted arrowheads symbolise the HS-AFM tip. **e**, HS-AFM-LS kymograph of a +CP WT NPC displaying FG-domain fluctuations and CP motion tracking (solid red line; top). Average distribution of CP position in +CP WT NPCs (bottom; ≥3 independent isolations, $n$ = 10). **f**, HS-AFM-LS kymograph detects FG domain fluctuations in a −CP WT NPC. **g**, Mean cross-sectional height profiles of +CP (≥3 independent isolations, $n$ = 10) and −CP (≥3 independent isolations, $n$ = 12) WT NPCs. The CNT is bounded by vertical dashed lines. Kap95 restores the CP in a dose-dependent manner: 0.1 μM (two independent isolations, $n$ = 9 NPCs), 0.5 μM (three independent isolations, $n$ = 9 NPCs) and 1.0 μM (three independent isolations, $n$ = 10 NPCs). The s.d. is shaded. Inset: relative height difference measured along the pore axis at each Kap95 concentration compared with +CP WT NPC and −CP WT NPCs. **h**, Average decay time within the CNT

(demarcated in **g**) as a function of the distance from the pore centre for +CP WT NPCs (≥3 independent isolations, $n$ = 10 NPCs) and following the addition of 1 μM Kap95 (≥3 independent isolations, $n$ = 9 NPCs) to −CP WT NPCs (≥3 independent isolations, $n$ = 12 NPCs). The s.e.m. is shaded. **i**, Relative enrichment of proteins associated with +CP and −CP WT NPCs, determined by label-free mass spectrometry. Proteins showing twofold-higher depletion (dashed red line) in −CP WT NPCs are shown in blue (proteasome components not labelled) and Nups in orange; $n$ = 2 biological replicates. Inset: estimated copy numbers of Kap95 in +CP and −CP WT NPCs. **j**, 1 μM Kap95 restores the CP in −CP WT NPCs (≥3 independent isolations, $n$ = 1 NPCs). **k**, Zoomed-in image of the restored CP in a +Kap95 WT NPC. HS-AFM-LS data were collected from left to right (small arrow denotes scan direction). Large tilted arrowhead symbolises the HS-AFM tip. **l**, HS-AFM-LS kymograph enables tracking of a restored CP in a +Kap95 WT NPC (solid red line; top) giving its average positional distribution (≥3 independent isolations, n = 9 NPCs; bottom). **a–d,j,k**, Scale bars, 50 nm (**a,c,j**) and 10 nm (**b,d,k**). All zoomed-in HS-AFM images were obtained at 0.15 s per frame (**b,d,k**). All kymographs were obtained at 1.875 ms per line (**e,f,l**). Source numerical data are provided.

Informed by this analysis, we applied HS-AFM-LS[60] to quantify the dynamic behaviour within the CNT and reserved HS-AFM images for visualization purposes. Our methodology leveraged the rotational symmetry of the NPC to create kymographs reaching a temporal resolution

of 1.875 ms per line (Fig. 1e,f). This approach improved the experimental time resolution by more than two orders of magnitude, effectively harmonizing it with many NPC transport time scales[18–22]. Kymograph data also allowed for tracking CP movements (Fig. 1e), measuring the

vertical range of both +CP and −CP WT NPC transport barriers (Fig. 1g) and extracting a decay time (τ) using an autocorrelation function (ACF) analysis workflow[67,69] as a readout for dynamic behaviour (Fig. 1h and Extended Data Fig. 2). We further characterized the raw oscillation damping and feedback control signals during HS-AFM-LS, comparing data from −CP WT NPCs with holey defects in lipid bilayers and bare DNA origami nanopores as quality controls (Extended Data Fig. 3).

HS-AFM-LS kymographs revealed dynamic FG domain fluctuations alongside CPs in +CP WT NPCs that preferentially localized at the pore centre despite showing sufficient mobility to explore its width (Fig. 1e). In contrast, random 'bursts' corresponding to FG-domain fluctuations were evident along the pore scaffold in −CP WT NPCs (Fig. 1f). Subsequent ACF analysis showed that +CP WT NPCs exhibited slower dynamics within a radius of approximately 15 nm of the pore axis, reflecting the larger CP mass (Fig. 1h). When the large CP mass was absent, −CP WT NPCs exhibited faster dynamics in this region (lower correlation, smaller τ) but slower dynamics near the scaffold. This inversion in dynamic behaviour probably reflects compaction of FG domains towards their tethering sites[16], resulting in reduced barrier height and a more open pore conformation (Fig. 1g,h).

## Karyopherins reconstitute a central plug-like structure in the NPC

Label-free quantitative mass spectrometry revealed that, following isolation, +CP WT NPCs contain approximately 15–20 Kap95 molecules, the primary nuclear import factor[7]. In contrast, over half of this long-lived endogenous Kap95 fraction, along with its adaptor Kap60, was displaced from −CP WT NPCs (Fig. 1i), consistent with a slow apparent dissociation rate[16,52] (Extended Data Fig. 4). Reduction was also observed for proteasome components, messenger-ribonucleoprotein remodelling factors, Tho/TREX complex members as well as Nup2 and Nup42, which are rapidly exchanging, non-essential FG Nups[72,76]. Although mRNA export factors such as Mex67, Mtr2 and Yra1 were not significantly depleted[7,72], they seem to reside at the NPC periphery during transport[78], suggesting a lesser role in CP formation. Importantly, the quantities of all other NPC components remained stable within the method's precision. Thus, a persistent fraction of Kaps seems to be strongly associated with the CP, which protrudes out of the CNT in +CP WT NPCs and retreats by about 20 nm into the CNT core in −CP WT NPCs (Fig. 1g). Hence, the CP substantially increases the vertical range of the transport barrier.

Providing support for this observation, exogenous Kap95 (0, 0.1, 0.5 and 1.0 μM) restored a dose-dependent CP-like structure[79,80] in −CP WT NPCs, highlighting a scaling relationship between Kap95 concentration and CP formation (Fig. 1g,j–l and Extended Data Fig. 4). At 1.0 μM, Kap95 effectively reproduced the qualitative behaviour and vertical range of the +CP WT NPC transport barrier. However, the overall dynamics of this reconstituted structure were notably faster than those of +CP WT NPCs (Fig. 1h), suggesting that additional nuclear transport factors and other constituents contribute to the native CP mass.

## Increasing FG-repeat concentration attenuates mutant NPC barrier dynamics

Subsequently, we employed HS-AFM to investigate two NPC mutants, each with specific alterations in their FG-repeat concentration. The first mutant was the septuple *nup42ΔFG nup159ΔFG nup60ΔFxF nup1ΔFxFG nup2ΔFxFG nsp1ΔFG&ΔFxFG* maximal FG-deletion NPC[76] (SWY3062; abbreviated ΔFG NPC), which contained mainly symmetric GLFG domains, namely Nup100, Nup116, Nup49, Nup57 and Nup145N (Fig. 2a–f and Supplementary Fig. 1). The ΔFG NPCs have a theoretical FG-repeat concentration of about 27 mM, 51% below the 53 mM FG-repeat concentration in isolated WT NPCs, assuming similar CNT volumes[5,7] (Supplementary Table 2). For the second NPC mutant, we generated a yeast strain that features double FG-domain-length Nsp1 constructs (amino acids (1–591)-(2–565)-(592–end); termed Nsp1FGx2;

Fig. 2g–l and Extended Data Fig. 5). Nsp1, the most abundant FG Nup, contains both FG and FxFG repeats and the theoretical FG-repeat concentration in Nsp1FGx2 NPCs is approximately 69 mM, surpassing WT NPCs by 30% (Supplementary Table 2).

After isolation, +CP ΔFG NPCs and +CP Nsp1FGx2 NPCs retained the dimensions, morphology and non-FG-containing Nup composition of WT NPCs (Fig. 2e,k and Extended Data Figs. 1,5). Both mutants also harboured CPs that were comparable in size, location and height profile (Extended Data Fig. 1). Interestingly, shorter fluctuations were observed along the inner scaffold in −CP ΔFG NPCs, suggesting increased GLFG domain cohesion reminiscent of a dense GLFG ring[81] (Fig. 2c,d and Supplementary Video 4), which might account for the slower dynamics adjacent to the CP in +CP ΔFG NPCs (Fig. 2f and Supplementary Video 5). In contrast, hyperextended FG-domain fluctuations resembling an intertwined sieve-like meshwork[33–37] dominated −CP Nsp1FGx2 NPCs and coalesced with the CP in +CP Nsp1FGx2 NPCs (Fig. 2g–j and Supplementary Videos 6,7). These hyperextended FG domains resulted in inverted dynamic behaviours compared with +CP WT NPCs, with slower fluctuations dominating the pore, except in the central CNT region (Fig. 2l).

## Increasing FG-domain entanglement impairs nucleocytoplasmic transport in vivo

We then tested whether the above mutations altered NPC permeability in vivo. Phenotypic analyses showed a fitness defect in both mutants in comparison to WT cells, which was milder in Nsp1FGx2 and more severe in ΔFG (Extended Data Fig. 5d). Given that Kaps play a role in reinforcing the transport barrier[50–52], we expressed Kap95–mNeonGreen in mutant strains and WT cells to ascertain the level of its enrichment at NPCs in vivo (Fig. 3a). The nuclear envelope:nucleoplasm (NE:N) ratio of Kap95–mNeonGreen decreased by 17% in ΔFG NPCs but increased by 11% in Nsp1FGx2 NPCs compared with WT, aligning with our mass spectrometry analysis (Extended Data Fig. 5b,c). In turn, this lowered the nuclear:cytoplasmic (N:C) ratios of Kap95–mNeonGreen in both mutant strains compared with the WT. These behaviours demonstrate how variations in FG-repeat concentration affect the binding avidity of Kap95 at NPCs, potentially influencing its nucleocytoplasmic partitioning[51]. One possibility is that ΔFG NPCs may be more kinetically accessible to Kap95 due to fewer FG repeats resulting from diminished avidity. Conversely, an increase in the number of FG repeats in Nsp1FGx2 NPCs increases Kap95 avidity and crowding, reducing the translocation efficiency. Providing support for this hypothesis, a simian virus 40 NLS (SV40NLS)–green fluorescent protein (GFP)–protein A (PrA) fusion reporter gave a steady state N:C ratio in Nsp1FGx2 cells that 24% lower than WT cells (Fig. 3b), signifying a transport defect in the Kap60–Kap95 import pathway. For passive permeability, the N:C ratio of a maltose binding protein–4×GFP fusion protein (MG4; 150 kDa)[13] was also less in Nsp1FGx2 cells than WT cells (Fig. 3c). In comparison, ΔFG NPCs displayed selective and passive permeabilities comparable to WT NPCs, as previously demonstrated[13,14]. These results show that additional FG-domain density impairs nucleocytoplasmic transport by an apparent overtightening of the NPC transport barrier.

## FG domains traverse WT and mutant NPCs

Next, we investigated whether GFP-fused NupFGs expressed in vivo exhibit enrichment at NPCs comparable to expressed Kap95, given the known tendency of FG domains to phase separate[82] in vitro into condensates[31,32] or form hydrogels[33–38]. We then overexpressed different GFP-tagged NupFGs in WT cells, including both cohesive (Nup100FG, hNup153FG and Nup116FG) and non-cohesive (Nup159FG, Nsp1FG and Nup60FG) variants[12,83]. Although non-cohesive NupFGs were evenly distributed throughout the cell without forming foci, cohesive NupFGs formed distinct punctate foci (Extended Data Fig. 6a).

As a representative example, Nup100FG, the yeast homologue of Nup98 in higher eukaryotes, formed foci exhibiting liquid-like

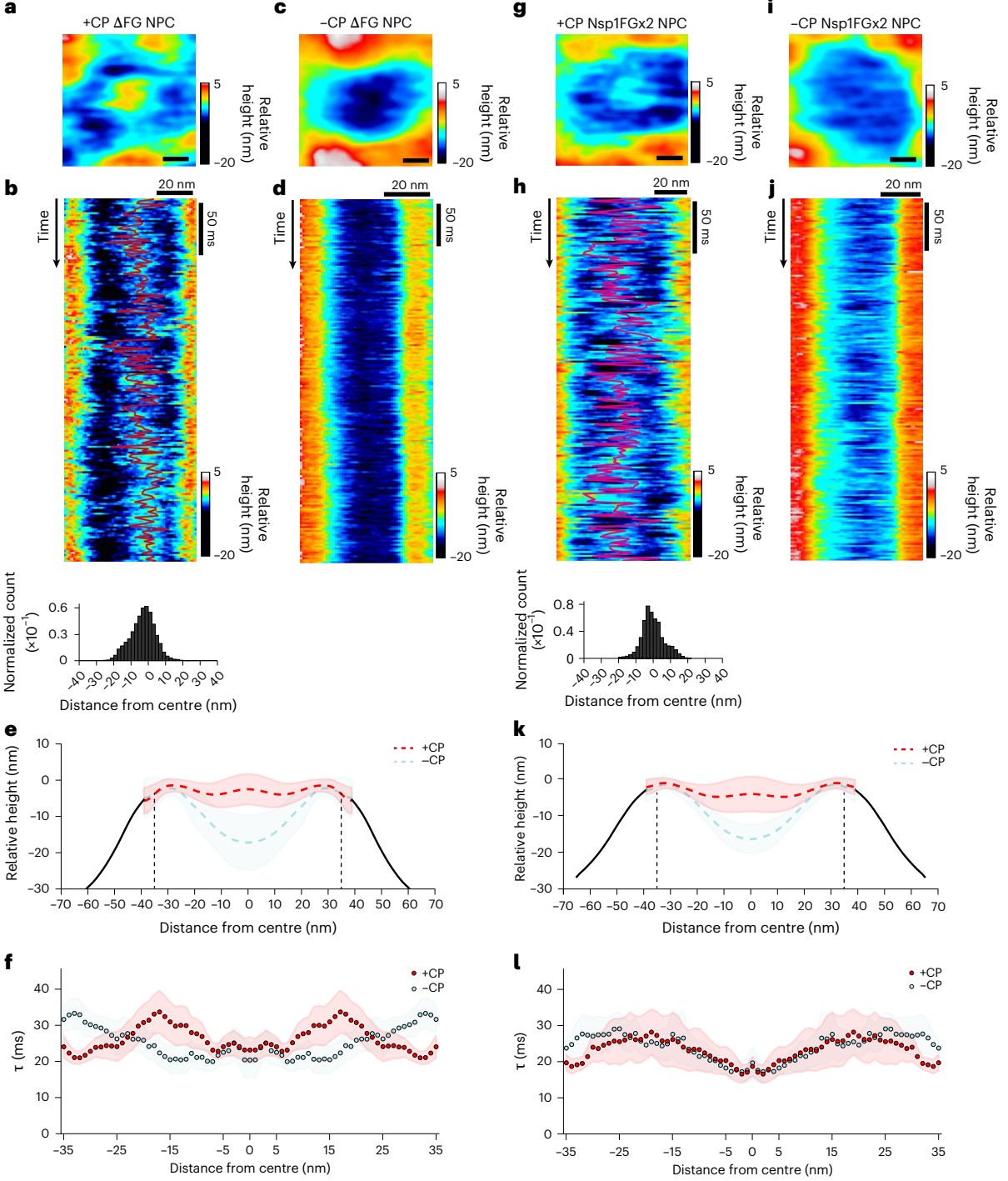

**Fig. 2 | FG-domain mutations disrupt the dynamic barrier properties of yeast NPCs. a**, Zoomed-in image of a +CP ΔFG NPC showing CP and FG-domain fluctuations. **b**, HS-AFM-LS kymograph of a +CP ΔFG NPC displaying FG-domain fluctuations and CP motion tracking (red; top). Average distribution of CP positions in +CP ΔFG NPCs (bottom; ≥3 independent isolations, $n = 11$ NPCs). **c**, FG-domain fluctuations in a −CP ΔFG NPC. **d**, HS-AFM-LS kymograph reveals reduced FG-domain fluctuations in a −CP ΔFG NPC. **e**, Mean cross-sectional height profiles of +CP (≥3 independent isolations, $n = 11$ NPCs) and −CP (≥3 independent isolations, $n = 10$ NPCs) ΔFG NPCs. The CNT is bounded by vertical dashed lines. Shading indicates the s.d. **f**, Average decay time within the CNT (bounded in **e**) as a function of distance from the pore centre for +CP (≥3 independent isolations, $n = 11$) and −CP (≥3 independent isolations, $n = 10$ NPCs) ΔFG NPCs. Shading indicates the s.e.m. **g**, Zoomed-in image of a +CP Nsp1FGx2 NPC showing CP and FG-domain fluctuations. **h**, HS-AFM-LS kymograph of a

+CP Nsp1FGx2 NPC displaying FG-domain fluctuations and CP motion tracking (red; top). Average distribution of CP positions in +CP Nsp1FGx2 NPCs (≥3 independent isolations, $n = 10$ NPCs; bottom). **i**, Hyper-elongated FG-domain fluctuations resembling a meshwork in a −CP Nsp1FGx2 NPC. **j**, HS-AFM-LS kymograph reveals hyper-elongated FG-domain fluctuations in a −CP Nsp1FGx2 NPC. **k**, Mean cross-sectional height profiles of +CP and −CP (≥3 independent isolations, $n = 10$ NPCs each) Nsp1FGx2 NPCs. The CNT is bounded by vertical dashed lines. Shading indicates the s.d. **l**, Average decay time within the CNT (bounded in **k**) as a function of distance from the pore centre for +CP and −CP (≥3 independent isolations, $n = 10$ NPCs each) Nsp1FGx2 NPCs. Shading indicates the s.e.m. **a,c,g,i**, Scale bars, 10 nm. Zoomed-in HS-AFM images were obtained at 0.15 s per frame. All kymographs were obtained at 1.875 ms per line. Source numerical data are provided.

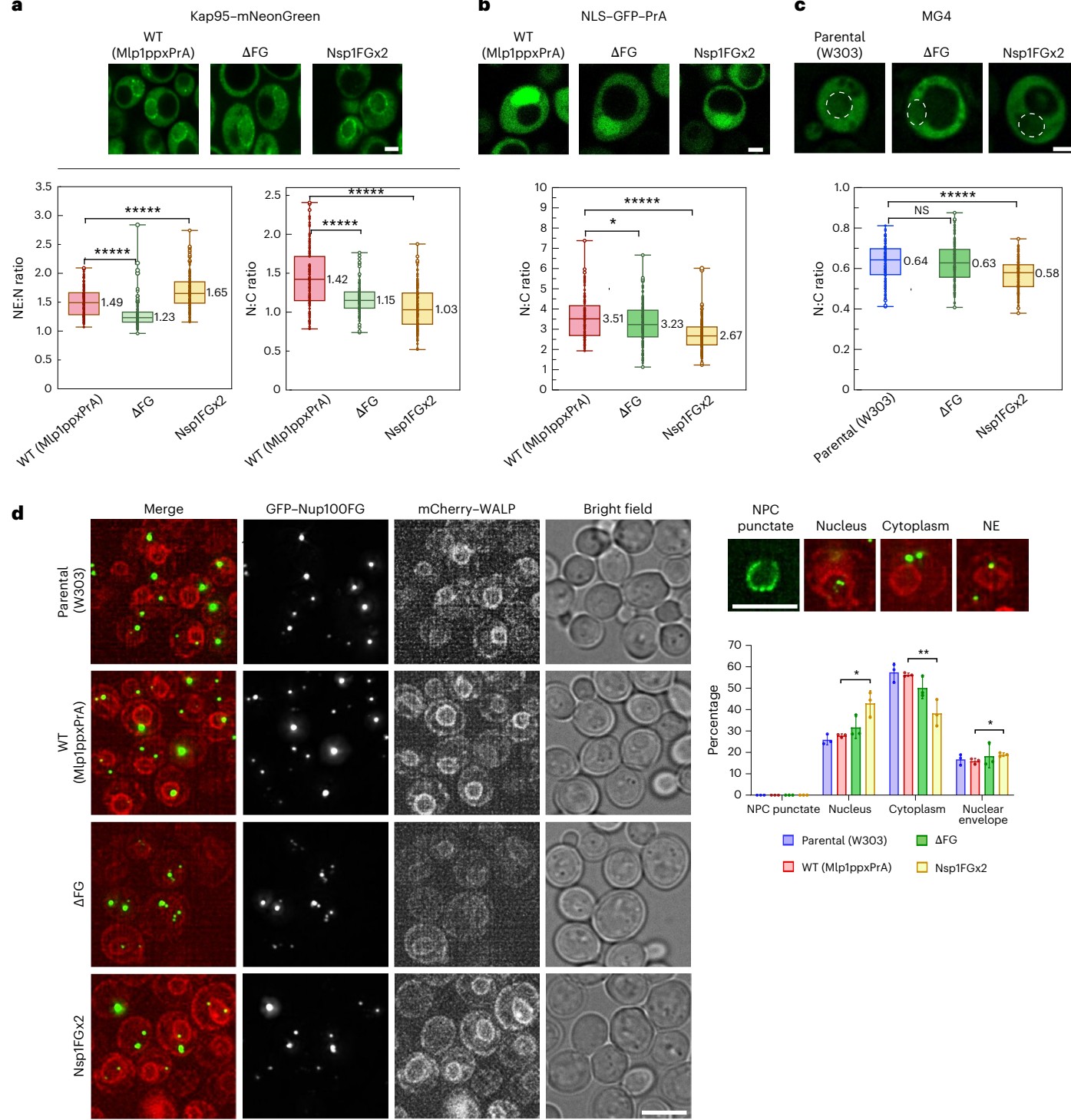

**Fig. 3 | Examination of Kap95 and GFP–Nup100FG domain behaviours in WT and mutant yeast NPCs in vivo. a–c**, In vivo localization of Kap95–mNeonGreen (**a**; $n = 109, 119$ and $138$ cells; NE:N, $P = 1.3 \times 10^{-10}$ for both; N:C, $P = 1.98 \times 10^{-10}$ (left) and $1.3 \times 10^{-10}$ (right)), NLS–GFP–PrA (**b**; $n = 99, 132$ and $215$ cells from left to right; $P = 0.023$ (left) and $1.3 \times 10^{-10}$ (right)) and MG4 (**c**; $n = 120$ for all groups; $P = 0.98$ (left) and $3.2 \times 10^{-6}$ (right)) in the indicated strains ($n = 4$ biological repeats). NE:N and N:C fluorescence intensity ratios are provided. **a,b**, Mlp1ppxPrA is encoded by the WT strains. **c**, The nuclear envelope–endoplasmic reticulum marker mCherry-L-TM was used to locate the nucleus (white dotted circles). **a–c**, Boxplots: the median (horizontal line; exact values are provided next to the line), and the first and third quartiles (box boundaries) are shown; the whiskers represent the minimum and maximum values. $P$ values were obtained using an ordinary one-way analysis of variance (ANOVA) test. **d**, Cellular localization of GFP-Nup100FG, overexpressed for 1 h, in the indicated

strains (left). Constitutively expressed mCherry–WALP-HDEL (red) was used as a nuclear envelope–ER marker. Maximal projection of four $z$-stacks, 0,2 μm interval. From left to right, representative images of GFP-Nup100FG condensates in the nucleus, cytoplasm and nuclear envelope (upper right). For comparison, NPC punctate signifies full-length Nup100–GFP. Percentage of GFP–Nup100FG localized at the indicated locations in a cell (lower right; three biological replicates, $n = 150$ cells for each strain, data represent the mean ± s.d.; $P = 0.012$, $0.0073$ and $0.026$ from left to right). Condensates were classified into mutually exclusive categories; in cells with multiple condensates, each was scored independently; 100–200 condensates were analysed in each replicate. Statistical analysis was performed using a two-tailed unpaired Student's $t$-test. 1,6HD controls and expression levels are in Extended Data Fig. 6. Scale bars, 2 μm (**a–c**) and 5 μm (**d**). *$P < 0.05$; **$P < 0.01$; *****$P < 0.00001$; NS, not significant ($P > 0.05$). Source numerical data are provided.

condensate behaviour, with approximately 90% dissolving in 1,6-hexanediol (1,6HD; Extended Data Fig. 6c). The Nup100FG foci were predominantly located in the cytoplasm and nucleoplasm and less frequently adjacent to the nuclear envelope (Fig. 3d). However, Nup100FG also lacked the punctate nuclear rim staining typically seen with NPC-associated proteins such as GFP-tagged full-length FG Nups, including Nup100–GFP (Fig. 3d and Extended Data Fig. 6d). Even Nup1, which is present in eight copies per NPC (Extended Data Fig. 6d), showed clear nuclear rim staining when fluorescently labelled[7,84], suggesting that significantly fewer than eight copies of Nup100FG were stably retained or present at any moment per NPC. Nuclear rim stainings of Nup100FG were also absent in ΔFG NPCs that contained only cohesive GLFG Nups and Nsp1FGx2 NPCs that displayed a substantially larger FG-repeat concentration (Fig. 3d). These results suggest that: (1) only a subset of FG domains can form condensates in vivo, (2) their presence in both the nucleoplasm and cytoplasm indicates they can traverse the CNT, potentially aided by Kaps, and (3) their lack of enrichment at the nuclear rim, unlike Kap95 (Fig. 3a), indicates that FG domains in the CNT do not form the same type of condensates as visible elsewhere in the same cell, even when the NPC composition is altered to potentially foster such condensation.

## FG hydrogels are holey and heterogeneous

Next, we compared the in situ characteristics of the NPC transport barrier with those of in vitro FG hydrogels[33–37], also termed FG phases[38]. We generated 5% fluorescein-5-maleimide (FMAL)-labelled in vitro hydrogel particles from Nup100FG and Nup116FG, which formed near-spherical particles that were selectively permeable to Kap95 but excluded bovine serum albumin (BSA) passive reporters, mirroring the behaviour of other FG hydogels[33,34,36–38] (Fig. 4a). The Nup100FG and Nup116FG particles were not dissolved by 1,6HD, which disrupts the NPC transport barrier[85,86] (Extended Data Fig. 7). However, 1,6HD prevented hydrogel formation[38] (Extended Data Fig. 7), confirming that, once formed, Nup100FG and Nup116FG hydrogels become increasingly insoluble. In this context, in vitro FG hydrogels display properties that are markedly different from those of in vivo NupFG foci (Extended Data Fig. 6c), emphasizing the need to differentiate between the two.

Subsequent photobleaching quantification revealed that neither Kap95 nor 1,6HD altered the mobile fractions or recovery half-lives of Nup116FG itself within the hydrogels. Of the Kap95 molecules that selectively partitioned into the Nup116FG hydrogels, a mean fraction of approximately 55% remained mobile (Fig. 4b). We then analysed the particle morphology using HS-AFM, achieving spatial and temporal resolution comparable to NPC behaviours. The Nup116FG hydrogel surfaces formed a static assembly resembling granular aggregates with

large variations in height and surface roughness (Fig. 4c–e). Furthermore, their surfaces were densely punctuated by irregularly shaped, holey structures at a density of approximately 80 holes μm$^{-2}$ and an average width of approximately 50 nm, with many coinciding with or exceeding the CNT diameter (Fig. 4f), validating previous reports[47]. Although HS-AFM-LS was challenging due to surface roughness, the corresponding HS-AFM images suggested FG-domain-like fluctuations inside these holes (Fig. 4g).

Consistent with other FG hydrogel studies[46,47] was the presence of approximately 1-μm-long, 25-nm-thick amyloid-like fibrils[49] that protruded from Nup100FG hydrogels (Extended Data Fig. 8). We also added Kap95 to examine how hydrogels facilitate selective transport but found no evidence of CP-like structures or dynamic remodelling such as local 'unfastening', 'dissolving' or 'melting' of crosslinks[33–38], or any increase in hole formation. Instead, the height of the hydrogel increased, probably due to Kap95 binding to surface-exposed FG repeats[47] (Extended Data Fig. 8i).

These findings demonstrate that the static amyloid-like mesostructure of in vitro FG hydrogels[46,47,49] contrasts sharply with the dynamic nanoscopic mechanism of the NPC transport barrier (Fig. 4h,i), highlighting morphological qualities that are inherently non-scalable from bulk in vitro FG-domain assemblies. Interactions between Kaps and FG domains, shaped by the rotationally symmetric architecture of the NPC (Fig. 1), instead suggest that the transport barrier is spatially organized into two temporal zones: a mobile CP and a surrounding annular region characterized by dynamic FG-domain fluctuations.

## An NPC architectural mimic recapitulates the appearance and dynamics of a bona fide NPC

Finally, we investigated whether the geometric organization of FG domains imposed by the NPC scaffold influences Kap binding and whether this behaviour results in CP formation. To test this hypothesis, we tethered 32 copies of Nsp1FG inside a 60-nm-wide, 30-nm-tall DNA origami channel to produce 'Nups organized on DNA' (NuPODs; Fig. 5a and Supplementary Table 2), which closely replicate the size, octagonal cross-section and selective barrier function of NPCs[63–67]. HS-AFM-LS kymograph analysis revealed fluctuations within NuPODs that were qualitatively comparable to −CP WT NPC behaviour but absent in empty DNA origami scaffolds (Fig. 5b,c). Although Nsp1FG chains previously showed sustained cohesion in NuPODs with a width of about 40 nm, the NuPODs generated here, which were approximately 20 nm larger than earlier versions[67], did not exhibit similar behaviour. Nevertheless, the addition of Kap95 to NuPODs induced a mobile, CP-like structure with dose-dependent changes in height and dynamics (0.1 μM versus 1 μM), qualitatively resembling the behaviour when Kap95 was added to −CP WT NPCs (Figs. 5d–g and 1g).

**Fig. 4 | FG hydrogels are static and display a holey irregular morphology.**
**a**, Nup116FG domains (5% FMAL-labelled) form hydrogel particles that are selectively permeable to Kap95 but exclude BSA. Incubation of Nup116FG hydrogels in 1,6HD had no discernible effect on their shape or their exclusionary property against BSA. The associated 1,6HD data are in Extended Data Fig. 7. AF568, Alexa Fluor 568; AF647, Alexa Fluor 647. **b**, Fluorescence recovery after photobleaching (FRAP) analysis of the mobile fraction (left) and recovery times (right) of Nup116FG within Nup116FG hydrogels under the indicated conditions (≥3 experimental replicates; n = 10 control, 15 +Kap95 and 16 +1,6HD hydrogel particles). The corresponding values for Kap95 within the Nup116FG hydrogels are also shown (4 experimental replicates; n = 11 hydrogel particles). Boxplots: the median (horizontal line), and the first and third quartiles (box boundaries) are shown; the whiskers represent the minimum and maximum values. Statistical analysis was performed using an ordinary one-way ANOVA test; mobile fraction, P = 0.71 (left) and 0.21 (right); recovery times, P = 0.22 (left) and 0.99 (right); NS, not significant (P > 0.05). **c**, Zoomed-out (left) and zoomed-in (right) HS-AFM images of a spherical Nup116FG hydrogel. **d**, HS-AFM images show that the surface morphology of Nup116FG hydrogels is static, irregular

and holey (dashed circles). These images were obtained by high-pass filtering the zoomed-in image in **c**. **e**, Surface roughness of Nup116FG hydrogels (three experimental replicates; n = 18 hydrogel particles; mean ± s.d. = 8.6 ± 3.2 nm). **f**, Nup116FG hydrogel hole diameters (three experimental replicates; n = 80 holes; mean ± s.d. = 54.6 ± 15.3 nm). **g**, Selection of zoomed-in images showing Nup116FG hydrogel holes of irregular size and shape obtained at 0.15 s per frame. **h**, Geometry, architecture, size and protein composition define the context-dependent dynamics of NPC barrier function. A subset of Kaps/cargoes binds dynamic FG domains (red arrows emphasise motion) forming a mobile CP along the pore axis. This remodelling increases FG-domain connectivity near the pore centre to improve passive exclusion (dark blue), directing facilitated transport through the annular region of the CNT (light blue). **i**, In vitro FG hydrogels are static amyloid-like aggregates that do not scale to the NPC's nanoscopic organization; irregular holes and uneven FG-domain distribution can create spurious, selectively permeable channels. **a,c,d,g**, Scale bars, 2 μm (**a**), 500 nm (left **c**), 200 nm (right **c,d**) and 10 nm (**g**). Source numerical data are provided. Credit, **i**: Yeung Mai (https://www.crinklepack.com.au).

This indicates that the NPC transport barrier mechanism is critically dependent on the dimensions and geometry of the CNT (Fig. 5d–g and Extended Data Fig. 9). Although our assay did not directly quantify the number of Kap95 molecules residing within the NuPODs, approximately 20 Kap95 molecules were found within Nsp1FG-coated NPC mimics of comparable size[87], aligning with our findings in +CP WT NPCs[7] (Fig. 1i and Extended Data Fig. 5b,c).

## Discussion

Our findings reveal fundamental insights into the context-dependent behaviour of the NPC transport barrier that reconciles the CP with prior observations and model predictions, unifying key aspects of virtual gating[30,88] and Kap-centric control[16,51,52,89–91]. At its core, the rotational symmetry of anchor sites around the circular NPC scaffold restrains but focuses the dynamic FG domains towards the pore

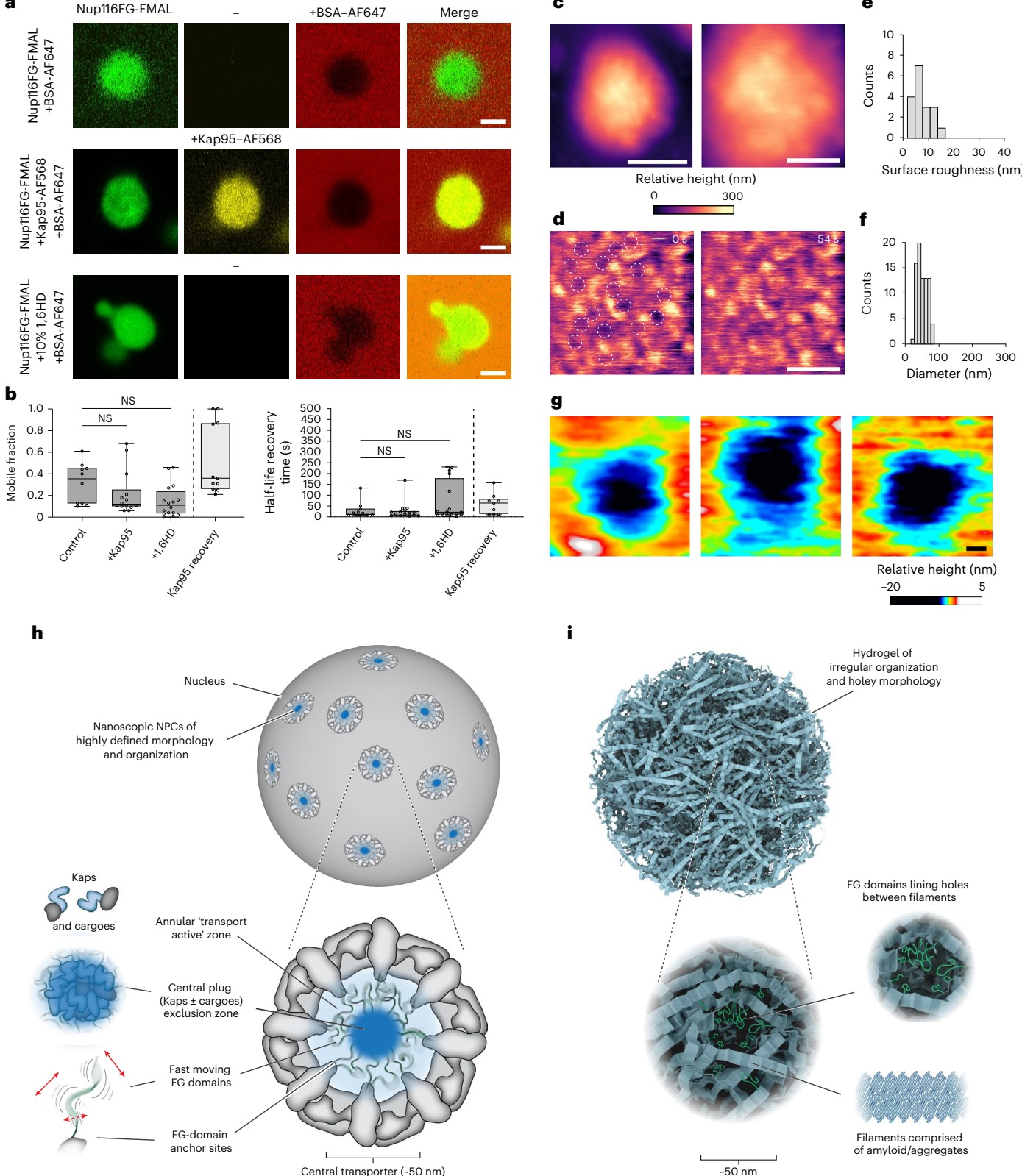

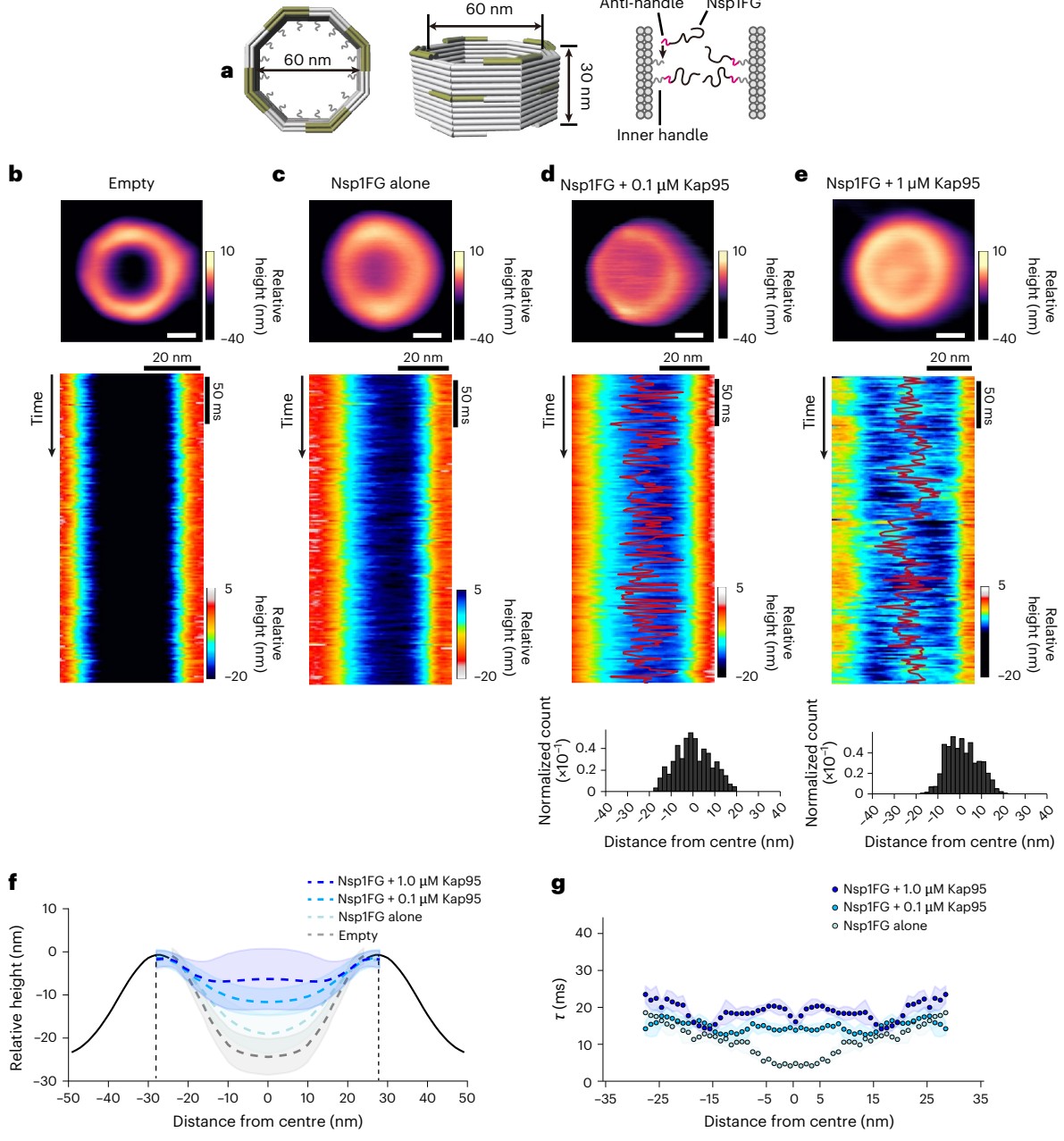

**Fig. 5 | Binding of Kap95 reconstitutes a CP-like structure in Nsp1FG-NuPODs.**
**a**, Schematic description of the DNA origami scaffold. **b**–**e**, Averaged HS-AFM images (top) of an empty DNA origami scaffold (**b**; $n = 13$ NuPODs), an Nsp1FG-functionalized DNA origami scaffold, that is, NuPOD (**c**; $n = 12$ NuPODs) as well as NuPODs in 0.1 µM (**d**; $n = 10$) and 1 µM (**e**; $n = 12$) Kap95. **d**,**e**, CP-like features can be observed. Scale bars, 20 nm. HS-AFM-LS kymographs (middle) showing an empty DNA origami scaffold lacking FG-domain fluctuations (**b**) and Nsp1FG fluctuations within a NuPOD (**c**) as well as NuPODs with Nsp1FG fluctuations and CP-like movements (red) in the presence of 0.1 µM Kap95 (**d**) and 1 µM Kap95 (**e**). Average positional distribution of CP-like movements in 0.1 µM ($n = 10$ NuPODs;

**d** (bottom)) and 1 µM Kap95 (**e** (bottom); $n = 12$ NuPODs). **f**, Mean cross-sectional height profiles comparing empty DNA origami scaffolds ($n = 6$), NuPODs alone ($n = 11$), NuPODs in 0.1 µM Kap95 ($n = 10$) and NuPODs in 1 µM Kap95 ($n = 12$). Shading indicates the s.d. **g**, Average decay time within NuPODs (area bounded by vertical dashed lines in **f**) as a function of distance from the pore centre of standalone NuPODs ($n = 11$), NuPODs in 0.1 µM Kap95 ($n = 10$) and NuPODs in 1 µM Kap95 ($n = 12$). Tau analysis of empty DNA origami scaffolds is inadmissible. Shading indicates the s.e.m.; ≥3 independent replicates for all experiments. All kymographs were obtained at 1.875 ms per line. More data and controls in Extended Data Fig. 9. Source numerical data are provided.

centre[4,5,7], evoking polymer brush-like behaviour[27,28] (Fig. 4h). This creates a highly turbulent environment of fluctuating FG domains within the CNT, consistent with a confined liquid state[32] that persists for days (Fig. 1d,f), in marked contrast to their rapid aggregation to hydrogel and amyloid states in vitro[31,32]. Even ΔFG NPCs dominated by symmetric GLFG domains remain fluid at the CNT centre (Fig. 2f). This underscores how the correct environment seems to be critical for observing bona fide FG-domain behaviour. However, the open pore state in −CP WT NPCs (Fig. 1c–f) suggests that FG domains alone are not sufficient to maintain an effective transport barrier, explaining the loss of selectivity following Kap depletion[51,52,85].

We previously postulated that differences in Kap–FG-domain binding avidity partition the CNT into regions of varying Kap mobility as a function of local occupancy[16,50–52,89–91]. To enhance the NPC transport barrier, a pool of nuclear transport factors and other molecular constituents probably meld together through high-avidity interactions

with FG repeats concentrated near the pore centre, forming a mobile CP structure. On average, the CP would preside near or at the pore centre, spanning approximately 30 nm in diameter and at least 20 nm in height on one lobe of the NPC (Fig. 1e,g). Extending across both lobes of the pore, the entire CP reaches nearly 40 nm in height, consistent with NPC cryo-electron tomographic reconstructions[4]. The CP may serve a dual purpose: to create a central exclusion zone that blocks nucleocytoplasmic traffic along the central axis of the pore (Fig. 1g,h) and to direct facilitated transport[18,92] through an annular lower density region[5,7,93], shaped by the FG-domain fluctuations radiating towards the CP (Fig. 1b,g,h). In this way, Kaps remodel FG-domain behaviour through CP formation, which acts as a 'keystone' establishing dynamic FG-domain connectivity at the pore centre.

Our model strongly complements three-dimensional (3D) minimal photon flux microscopy (MINFLUX) tracking of Kaps through individual NPCs, which revealed transport-active Kaps traversing an annular region of the NPC but not along its central transport axis[21]. Although the approach by Sau and colleagues provided single molecule-resolution tracking, it did not directly resolve the structure or dynamics of the NPC transport barrier. In contrast, our study directly captures these features in detail. We show that the CP—comprising a persistent pool of endogenous Kaps[51]—forms a sterically restrictive structure that establishes the central exclusion zone within the NPC. Given that their experiments were performed in permeabilized cells[18,21], the presence of this CP-bound pool of endogenous Kaps probably prevented exogenous Kaps from accessing the central exclusion zone. In parallel, selective transport may proceed through the surrounding annular region of dynamic FG domains. For this reason, reduced dynamics due to overcrowding within the annular region of +CP Nsp1FGx2 NPCs (Fig. 2l) probably suppressed selective transport (Fig. 3a–c). Together, these findings underscore the need to account for endogenous Kaps in future NPC studies.

The CP may also serve additional roles such as acting as a dynamic reservoir of Kaps that exchange with other regions of the NPC during large structural transitions, such as pore dilation. Conversely, dilated NPCs may recruit additional Kaps to the CP to compensate for a 20% reduction in FG-domain concentration relative to constricted NPCs[5,8,9,94] (Supplementary Table 2), thereby stabilizing the central exclusion zone and maintaining transport barrier function.

The dynamic nature of the NPC is further evident in how NupFGs traverse its transport barrier in vivo (Fig. 3d and Extended Data Fig. 6a). This environment acts as a 'good solvent' for cohesive FG domains like Nup98, allowing it to adopt an expanded conformation within NPCs compared with its collapsed state in solution[32]. Similarly, Nup100FG traverses the NPC without significant enrichment in WT, ΔFG or Nsp-1FGx2 NPCs despite its tendency to form condensates in vivo. Whereas in vitro NupFG condensates and hydrogels rely on homotypic interactions of single-component FG domains[31,32], the NPC environment is defined by a diverse ensemble of FG domains and Kap–cargo complexes engaging in multifaceted interconvertible interactions essential for CNT function. Quinary interactions within the cellular milieu can also attenuate inter-FG-domain interactions[95]. Additional structural factors, including scaffold tethering that constrains the mobility of FG domains, pore geometry that defines their spatial arrangements, and the confined nanoscale volume, crucially also suppress NupFG phase separation[96]. These defining characteristics underscore the context-dependent behaviour of the NPC transport barrier, often overlooked in in vitro assemblies and in vivo NupFG condensates.

The CP was previously proposed to form a pore-spanning sieve-like meshwork or 'selective phase'[58], grossly approximated by macroscopic FG hydrogels[33–38], with FG-domain concentrations in NPCs being expected to exceed 200 mg ml⁻¹ (ref. 38). Here we found that the CP is labile and can dissociate without loss of core FG Nups, indicating that it is not primarily comprised of FG domains. Instead, CP loss correlates with a depletion of transport factors and cargo,

and can be reconstituted by the addition of a single type of transport factor (Fig. 1g–l and Extended Data Fig. 4e–g). Thus, the CP is mainly comprised of transport factors and cargo associating dynamically with FG repeats, consistent with previous findings[7]. It is further evident that the dynamic nanoscale organization of the NPC is inherently context-dependent and not directly scalable from bulk in vitro FG-domain assemblies[46,47,49] (Fig. 4 and Extended Data Fig. 8). Still, how might FG hydrogels exhibit NPC-like characteristics? One possibility is that the holes of approximately 50 nm may serve as FG-gated passageways allowing even large cargoes to access the hydrogel interior[97] (Fig. 4d–g and Extended Data Fig. 8). However, their irregular, asymmetric geometries and non-uniform FG-domain organization would preclude the formation of CP-like structures following Kap binding (Fig. 4i). To be clear, these holes should not be conflated with the hypothesized 'meshes' of the selective phase model[48], which are thought to be an order of magnitude smaller[33–38]. Notably, adding Kap95 did not induce the melting or 'resealing' of FG hydrogels (Extended Data Fig. 8i), which presents a divergence from model predictions.

By contrast, the dynamic intercalation of FG domains within Nsp1FGx2 NPCs more closely approximates a sieve-like meshwork (Fig. 2i). Yet, excessive FG-domain concentration overtightens the transport barrier and reduces transport efficiency in vivo (Fig. 3a–c). Hence, conditions that exceed a threshold FG-domain concentration or constrain the NPC's dynamic state, such as FG-domain ageing and aggregation[31,32], may contribute to pathological outcomes[26,98]. Indeed, Kaps, molecular chaperones and even FG Nups such as Nsp1 can modulate FG-domain phase separation[83,99,100] and may assist in maintaining the liquid-dynamic properties of the NPC transport barrier. Additional uncertainty remains as to whether the NPC transport barrier mechanism can be unambiguously distinguished as a putative FG phase[38] from within the broader continuum of condensate behaviours, ranging from liquid- to solid-like, hydrogel and aggregated amyloid states[101,102]. How these phases compare with macroscopic percolated network hydrogels that form independently of phase separation remains unclear[45]. Notably, highly idealized mutations can convert Nup98 from a solid hydrogel to a liquid state[103]. Hence, greater clarity is needed to delineate these behaviours.

Based on the current evidence, the CP probably represents a liquid-like nanocluster that forms under subsaturating conditions of nuclear transport factors and other constituents[101,104,105], albeit one that is confined within the NPC scaffold by FG-domain tethers (Fig. 4h). Although the molecular structure of the CP remains unknown, recombinant Kap95 restored CP-like structures in −CP WT NPCs in a dose-dependent manner (Fig. 1g and Extended Data Fig. 4e–g) as well as in NuPODs[106] (Fig. 5 and Extended Data Fig. 9). This finding corroborates data from other NPC mimics[87,107,108] (Fig. 5) and supports the role of Kaps in reinforcing the NPC transport barrier against passive macromolecules[51–53]. Promiscuous interactions between FG domains and other nuclear transport factors[90] may allow the native CP mass to vary, facilitating a range of dynamic behaviours. To traverse the pore, large cargoes such as viruses with multiple FG-binding sites[109] or Kap-conjugated vesicles for drug delivery[110] would then outcompete FG domains, displacing and replacing the CP.

Our findings support a model in which a pool of Kaps, rather than FG-domain gelation or phase separation, organizes and reinforces NPC transport barrier dynamics, reconciling single-molecule tracking[18,21] and structural studies[4,5,7]. Achieving optimal transport conditions probably requires a precise balance of FG-domain arrangement, quantity and composition within the nanopore geometry of the NPC, alongside Kaps, in line with theoretical predictions[111]. These nanoscopic insights provide a deeper context-dependent understanding of IDP complexes, transcending phenomenological, bulk-averaged interpretations. How the CP might adapt nucleocytoplasmic transport to cellular needs, such as by integrating other nuclear transport factors into the CP, are compelling questions for future research.

## Online content

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

¹Biozentrum, University of Basel, Basel, Switzerland. ²Swiss Nanoscience Institute, University of Basel, Basel, Switzerland. ³Laboratory of Cellular and Structural Biology, The Rockefeller University, New York, NY, USA. ⁴Ikerbasque, Basque Foundation for Science, Bilbao, Spain. ⁵Instituto Biofisika (UPV/EHU, CSIC) and Fundacion Biofisica Bizkaia/Biofisika Bizkaia Fundazioa (FBB), University of the Basque Country, Leioa, Spain. ⁶European Research Institute for the Biology of Ageing, University of Groningen, University Medical Center Groningen, Groningen, Netherlands. ⁷School of Computer Science and Engineering, The Hebrew University of Jerusalem, Jerusalem, Israel. ⁸Laboratory of Mass Spectrometry and Gaseous Ion Chemistry, The Rockefeller University, New York, NY, USA. ⁹Department of Bioengineering and Therapeutic Sciences, University of California, San Franscisco, San Francisco, CA, USA. ¹⁰Department of Pharmaceutical Chemistry, University of California, San Francisco, San Francisco, CA, USA. ¹¹Quantitative Biosciences Institute, University of California, San Francisco, San Francisco, CA, USA. ¹²Nanobiology Institute, Yale University, West Haven, CT, USA. ¹³Department of Cell Biology, Yale School of Medicine, New Haven, CT, USA. ¹⁴Department of Biomedical Engineering, Yale University, New Haven, CT, USA. ¹⁵Present address: Centro Andaluz de Biología del Desarrollo, Universidad Pablo de Olavide-Consejo Superior de Investigaciones Científicas-Junta de Andalucía, Sevilla, Spain. ¹⁶Present address: Lawrence Livermore National Laboratory, Livermore, CA, USA. ¹⁷Present address: Pentapharm AG, Aesch, Switzerland. ¹⁸These authors contributed equally: Toshiya Kozai, Javier Fernandez-Martinez. ✉e-mail: rout@mail.rockefeller.edu; roderick.lim@unibas.ch

## Methods

### Yeast strains and materials

All *S. cerevisiae* strains used in this study are listed in Supplementary Table 3. Unless otherwise stated, the strains were grown at 30 °C in YPD medium (1% yeast extract, 2% bactopeptone and 2% glucose) supplemented with adenine hemisulfate (40 mg l⁻¹; Sigma).

### Construction of yeast strains

Yeast strains were constructed using standard molecular genetic methods[112]. For the construction of the ΔMlp1ΔMlp2 mutant, the genomic loci of W303 of *Mlp1* and *Mlp2* were replaced with the URA3 and LEU2 markers, respectively. For the construction of the Nsp1FGx2 mutant, the *Nsp1* open reading frame was knocked out from a strain carrying plasmid PLPC19 using a cassette amplified from plasmid pAG25 that introduces a nourseothricin resistance marker (ΔNsp1)[113]. Using NSP1 from the S288C reference genome as a basis, a gene was constructed to effectively double the FG region of Nsp1 by repeating residues 2–565 between residues 591 and 592, resulting in the sequence: Nsp1(1–591)-(2–565)-(592–824) (Nsp1FGx2). In the Nsp1FGx2 gene, the NSP1 intron was removed and the gene was codon optimized and synthesized by Twist Bioscience. The Nsp1FGx2 gene was cloned into the p410 plasmid under control of the Adh1 promoter and terminator. ΔNsp1 yeast were transformed with the p410-Adh1-Nsp1FGx2 and plated on YPD + G418. Single colonies were passaged three times on YPD + G418 at 30 °C and then plated on SC-Ura and SC + 5-FOA plates. Colonies that had naturally lost the PLPC19 plasmid (SC-Ura negative, SC + 5-FOA positive) were chosen for downstream analysis. The sequences for all the oligonucleotides and vectors used in this study are available on request.

### Fitness and growth assays

To analyse their fitness at different temperatures, strains were cultured overnight in liquid YPD medium at 30 °C. The cells were counted and diluted to a final concentration of $2 \times 10^4$ cells ml⁻¹. Four tenfold serial dilutions were made and spotted on YPD plates that were incubated at 25, 30 and 37 °C for 1–2 d. Three biological replicates of each experiment were performed. A ΔNsp1 + Nsp1 strain was also analysed to ensure no deleterious effect of the gene deletion background (not shown). Plates were imaged using a Versadoc imaging system (linear detection range; BioRad). Semiquantitative estimation of fitness was performed using ImageJ (National Institutes of Health)[114] as described previously[115].

For growth curves, strains were cultured as biological replicates as described in the previous paragraph, inoculated in 96-well plates (150 µl) and cultured with 19 Hz agitation at each of the different temperatures for 30–50 h in a Biotek Synergy HT microplate reader. The $OD_{600}$ reads were calibrated[116] and transformed into cell counts for plotting the curves.

### Affinity purification of endogenous *S. cerevisiae* NPCs

We used our previously published method for the isolation of endogenous whole NPCs from *S. cerevisiae*[5,7,117]. Briefly, *Mlp1* was genomically tagged in each strain with PrA preceded by the human rhinovirus 3C protease (ppx) target sequence (GLEVLFQGPS). The cells were cultured at 30 °C in YPD medium until they reached the mid-log phase (approximately $3 \times 10^7$ cells ml⁻¹), harvested, frozen in liquid nitrogen and cryogenically lysed in a planetary ball mill PM100 (Retsch; https://www.ncdir.org/protocols/). Affinity purification was performed in resuspension buffer (20 mM HEPES–KOH pH 7.4, 50 mM potassium acetate, 20 mM NaCl, 2 mM MgCl₂, 0.5% (wt/vol) Triton X-100, 0.1% (wt/vol) Tween-20, 1 mM dithiothreitol (DTT), 10% (vol/vol) glycerol and 1/500 (vol/vol) protease inhibitor cocktail (Sigma)). Native elution by protease cleavage was achieved in a similar buffer without Triton X-100 or protease inhibitors. Protease inhibitors were added to the purified NPCs and either conserved at 4 °C or frozen in liquid nitrogen and stored at −80 °C. For the ΔMlp1ΔMlp2 and WT comparison NPCs, Nup84 was genomically tagged with Protein A and the NPCs were purified in the same way.

Successful incorporation of Nsp1FGx2 was determined by SDS–PAGE analysis and western blotting of purified NPCs as follows. NPCs from Nsp1FGx2, Mlp1-PPX-PrA cells cultured and lysed as described earlier (previous paragraph) were purified using a citrate resuspension buffer (20 mM HEPES pH 7.4, 150 mM NaCl, 250 mM sodium citrate, 1% (vol/vol) Triton X-100 and 1×protease inhibitor cocktail)[72]. Affinity-captured NPCs were eluted from beads by adding 20 µl of 1×lithium dodecyl sulfate loading buffer (Thermo Fisher) and vortexing for 10 min at room temperature. The eluted NPCs were run on an NuPAGE 4–12% gel for 60 min and analysed by Coomassie staining or western blot. For the western blots, the NPCs were transferred overnight onto polyvinylidene difluoride membranes at 4 °C, blocked with 5% powdered fat-free milk in tris-buffered saline with 0.1% Tween-20 (TBST) and probed overnight with primary antibody monoclonal mouse anti-Nsp1 (1:5,000; ab4641, Abcam). The membranes were washed with TBST, reprobed with secondary antibody (anti-mouse; 1:5,000) for 1 h at room temperature, which were then detected with ECL western blot substrate and imaged using an ImageQuant 4000 LAS system.

### Recombinant protein expression and purification of Kap95

Yeast Kap95 was recombinantly purified in-house from *Escherichia coli* BL21DE3. The protein was purified from a Ni-NTA column. First, 3 l BL21DE3 bacteria at optical density at 600 nm ($OD_{600}$) = 0.6 were induced with 0.5 mM isopropyl β-D-1-thiogalactopyranoside and overnight incubation at 21 °C. After harvesting, the bacteria were lysed in lysis buffer (10 mM Tris pH 7.5, 100 mM NaCl, 1 mM DTT and 10 mM imidazole containing DNase, lysozyme, Pefabloc and protease inhibitor cocktail complete) for 1 h at 4 °C. The lysate was sonicated with a stab-sonicator (Branson Digital Sonifier SFX 550) at 30% amplitude, 3 s pulse and 4 s off, for a total of 5 min pulse. The supernatant from the lysate was separated from unlysed debris after centrifugation at 25,000g and 4 °C for 1 h. The supernatant was loaded onto a 5 ml Ni-NTA column (GE Healthcare). After the flow-through and wash steps, the proteins were eluted in a lysis buffer supplemented with 500 mM imidazole.

### Quantitative mass spectrometry analysis of affinity-purified NPCs

The affinity-captured NPCs (5 µg per sample) were concentrated by pelleting at 98,884g (40,000 rpm) at 4 °C for 20 min in a TLA 55 rotor (Beckman). The pelleted NPCs were solubilized with a final concentration of 1×NuPAGE LDS sample buffer (Invitrogen) and 50 mM DTT, and incubated at 72 °C for 10 min. The samples were then cooled to room temperature and treated with a 30 mM (final concentration) iodoacetamide (Sigma) at room temperature in the dark for 30 min. The samples were then loaded into 4% (37.5:1) stacking acrylamide SDS–PAGE gels prepared in-house. The resulting stacked bands (gel plugs) were stained with Coomassie brilliant blue, excised and processed for quantitative mass spectrometry analyses.

Proteins in gel plugs were digested and peptides were extracted as described previously[118]. The purified peptides were bound to C18 StageTips. Peptides eluted from the StageTips were analysed using an LCMS Easy-nLC system coupled to a Thermo Orbitrap Exploris mass spectrometer (Thermo Fisher). The SpectroMine (version 2.8.210609.47784, Biognosys AG) software was used to identify and quantitate the mass spectrometric data by means of label-free quantitation[118]. The protein label-free-quantitation outputs from SpectroMine were further analysed with Microsoft Excel. The absolute stoichiometries of the proteins were determined by normalizing the summed copies of Nup192, Nup188, Nup170, Nup157, Pom152 and Nic96 per NPC to 112 copies (that is, 16 for Nup192, Nup188, Nup170, Nup157 and Pom152, and 32 for Nic96).

### Negative-stain transmission electron microscopy and image processing

For negative-stain analysis of purified yeast NPCs, 5 µl purified sample was adsorbed onto 400-mesh carbon-coated grids rendered hydrophilic

using a glow-discharger at low-vacuum conditions. The grids were washed with three drops of water or TEM buffer (20 mM HEPES, 20 mM NaCl and 1 mM DTT) and subsequently stained with 5-μl drops of 2% (wt/vol) uranyl acetate. The samples were examined with a Tecnai G2 Spirit (FEI Company; operating at 80 kV accelerating voltage) or Talos L120C (Thermo Fisher; operating at 120 kV) transmission electron microscope. On the Tecnai G2 Spirit, transmission electron microscopy images were recorded with an Olympus Veleta camera 4k using the EMSIS Radius software at nominal magnification of ×49,000. On the Talos L120C TEM, images were recorded with a Ceta 16 M Pixel CMOS camera using the TIA software at a nominal magnification of ×28,000. The software packages cryoSPARCv4 (ref. 119) and RELION v.3.1.2 (ref. 120) were used for particle picking and two-dimensional (2D) classification.

### High-speed atomic force microscopy measurements

All HS-AFM data in this study were acquired using a HS-AFM 1.0 system (RIBM) operating in tapping mode with a dynamic proportional-integral-derivative controller using a standard scanner with a $xy$ scan range of $6 \times 6$ μm$^2$. For scan areas smaller than $80 \times 80$ nm$^2$ ($80 \times 80$ pixels; 'zoomed-in' images) the scan speed was 150 ms per frame. For scan areas larger than $200 \times 200$ nm$^2$ ($200 \times 200$ pixels; 'zoomed-out' images) the scan speed was 1–2 s per frame. Pristine QUANTUM-AC10-SuperSharp cantilevers (nanotools GmbH) bearing a tip radius of ≤2 nm, nominal spring constant of 0.1 N m$^{-1}$, resonant frequency of approximately 0.5 MHz and a quality factor of about 2 in buffer were used in all experiments. The typical set-point amplitude ($A_{set}$) was 80–90% of the free cantilever oscillation amplitude $A_{free}$, which was set to 2–3 nm (refs. 68,121).

Droplets (3 μl) containing isolated NPCs were dispensed on freshly cleaved mica surfaces for 3 min at room temperature. Any excess was removed by rinsing the sample with buffer (20 mM HEPES–KOH pH 7.4, 20 mM NaCl and 2 mM MgCl$_2$). As an exception, isolated ΔFG NPCs were immobilized on 3-aminopropyltriethoxy silane-functionalized mica, which was prepared by incubating a droplet (3 μl) of 0.1% 3-aminopropyltriethoxy silane on a freshly cleaved mica surface for 5 min, followed by thorough rinsing with pure water. For CP-reconstitution experiments, −CP WT NPCs were monitored in situ and in real time by HS-AFM while Kap95 was added to the imaging buffer, achieving a final concentration of 0.1, 0.5 or 1.0 μM Kap95. The images were obtained 30 min after the addition of Kap95. For DNA origami pores, more details are provided in the "Sample preparation for HS-AFM measurements on DNA origami pores" section. For in vitro FG hydrogels, details are provided in the "In vitro phase separation of FG hydrogels for HS-AFM" section.

### High-speed atomic force microscopy line scanning measurements

During HS-AFM-LS, the slow-scan axis is disabled and only the fast-scan axis operates. An NPC or DNA origami pore is first centred during HS-AFM imaging before switching to line scanning so that the centre of the NPC or DNA origami pore undergoes repetitive scanning, resulting in a kymograph that spans the pore diameter. Line scans were recorded over 80 pixels (1 nm per pixel for NPCs, 0.625 nm per pixel for empty DNA origami pores and 0.75 nm per pixel for NuPODs) in the fast-scan axis at 1.875 ms per line. At this scan speed, the tip spends 0.9375 ms moving from left to right, then returns in another 0.9375 ms along the same line. As standard practice, only the left-to-right data were analysed to ensure data consistency. At an oscillation frequency ($f_c$) of approximately 0.5 MHz with 2-μs periods, each approximately 1-nm-sized pixel is tapped approximately six times with a tip-sample contact time of 200 ns per tap and 0.1 nm lateral spacing between taps due to the continuous lateral-scanning motion of the tip (Extended Data Fig. 3).

### Nucleoporins organized on DNA origami

**NuPOD assembly.** MBP-Nsp1-DNA[63] (50 mM Tris pH 8.0, 300 mM NaCl and 0.2 mM TCEP) was added to 60 nm DNA origami channels with 32 handles[65] (100 mM Tris–HCl, 20 mM EDTA and 15 mM MgCl$_2$) at double excess over the number of handles (for example, 5 nM origami with 32 handles mixed with 320 nM MBP-Nsp1-DNA) and incubated for 2 h at 37 °C. The MBP was then cleaved through the addition of TEV protease and 1 mM DTT for 1.5 h at 30 °C. The product is a mixture of free MBP, Nsp1FGs and NuPODs.

**Negative-stain transmission electron microscopy, image processing and data analysis of NuPODs.** Samples (5 μl) were adsorbed on 400-mesh carbon-coated grids rendered hydrophilic using a glow-discharger at low-vacuum conditions. They were subsequently stained using 5-μl drops of 2% (wt/vol) uranyl acetate. The grids were examined in a Talos L120C G2 (Thermo Fisher Scientific) transmission electron microscope operating at 120 kV. Images were recorded using a Ceta CMOS camera at a nominal magnification of ×73,000. The images used in the calculation of the intensity ratios between inside and outside underwent a rolling-ball background subtraction to accommodate for uneven staining. A 250 nm × 250 nm area without pores was selected as the average background reference in each image. The average intensity of the area inside the 60 nm pores was measured with a 40-nm-diameter circle. The values were analysed in GraphPad Prism 10.2.3. Statistical significance was calculated using an ordinary one-way ANOVA with Tukey's multiple comparisons test.

**Sample preparation for HS-AFM measurements on DNA origami pores.** Supported lipid bilayers were prepared as described[63]. Dipalmitoylphosphatidylcholine (DPPC, Avanti Polar Lipids) and didodecyldimethylammonium bromide (DDAB, Avanti Polar Lipids) lipids were dissolved in chloroform and combined in a 3:1 molar ratio (DPPC:DDAB). The solvent was evaporated under a steady nitrogen stream inside a fume hood for 1 h, followed by vacuum exposure for at least 4 h. The resulting dry lipid film was resuspended in MilliQ water to achieve a concentration of approximately 1 mg ml$^{-1}$. This solution was then placed in an ultrasonic bath (Elmasonic p30H, Elma Schmidbauer GmbH) maintained at 65 °C. The disruption of large multilamellar vesicles occurred through sonication treatments lasting 15 min using the pulse setting at 80 kHz. The lipid dispersion solution was loaded into an Avanti mini-extruder kit (Avanti Polar Lipids) while ensuring that the temperature remained at 65 °C. The lipid solution was passed through a 100-nm-pore-diameter Nucleopore Track-Etch membrane supported by two PE drain discs (Global Lifesciences Solutions UK Ltd). Repeating the extrusion process at least 20 times then yielded small unilamellar vesicles.

A 2.7 μl droplet of vesicle buffer (55.5 mM MgCl$_2$ and 55.5 mM CaCl$_2$) was placed onto a freshly cleaved mica sheet. Next, 0.3 μl of small unilamellar vesicles were then gently mixed into the droplet. The sample stage was then subjected to 65 °C for 20 min. After that, the temperature was gradually reduced back to room temperature over a span of 20 min. This led to the rupture of vesicles, resulting in the formation of a positively charged gel-phase-supported lipid bilayer. Any surplus vesicles present in the supernatant were washed away by initial rinsing with water and then exchanged with 10 mM phosphate buffer (PB, pH 7.0) containing 40 mM MgCl$_2$. To ensure a pristine and uniform lipid surface, this rinsing process was gently repeated 3–5 times before introducing 2–4 μl of NuPODs (0.5–1 nM) for subsequent imaging purposes. Free Nsp1FG and MBP in solution was removed by first washing the sample with Tris-EDTA buffer (100 mM Tris–HCl, 20 mM EDTA and 40 mM MgCl$_2$) and a subsequent buffer exchange with PB before imaging them using HS-AFM. The estimated concentration of Nsp1FG per nanopore is approximately 0.6 mM (Supplementary Table 2). Dose-dependent binding was examined by incubating NuPODs in 0.1 or 1 μM Kap95, followed by rinsing before HS-AFM analysis and negative-stain transmission electron microscopy. The images were obtained 30 min after the addition of Kap95.

## HS-AFM data processing

All HS-AFM 2D images were corrected for drift using an in-house Python-based software that also converted the file into TIFF format[61]. HS-AFM images were analysed in ImageJ (1.53c) and self-written analysis routines in Python. Image filtering, contrast adjustment and height/diameter measurements were performed by ImageJ. First, a flattening filter with first-order polynomial plane was applied to all HS-AFM images to compensate for $xy$-tilting. Subsequently, the heights of all HS-AFM images are set relative to the NPC or NuPOD scaffold by subtracting the average height of the scaffold. Displayed HS-AFM zoomed-in images were treated by a 2D Gaussian filter with a s.d. of 1 pixel, followed by bicubic interpolation with a scale factor of two in the $x$ and $y$ direction. HS-AFM images of NupFG hydrogels (Fig. 4d and Extended Data Fig. 8d) were processed with a 20-nm high-pass filter.

HS-AFM-LS data were processed by self-written routines in Python. Full kymographs were separated into 600-ms segments, followed by autocorrelation function analysis (details in subsequent sections), corrected for drift in $z$ and tilt in the fast-scan axis, and set the height relative to the NPC or NuPOD scaffold. In addition, a Fast Fourier transform filter was applied to eliminate periodic noise at frequencies of 50 and 150 Hz, which had root mean square (RMS) amplitudes of approximately 0.09 and 0.08 nm, respectively. These amplitudes (about 8% of the value at the centre of the NPC) are negligibly small and thus have no discernible impact on the HS-AFM-LS dynamic analysis.

## Average HS-AFM images of NPCs and NuPODs

We manually selected image stacks of NPCs scanned with a pixel size of 1 nm per pixel. After drift correction, the raw images in each stack were averaged over a corresponding pixel position. The average images were then rotated and aligned to the reference image by phase cross-correlation. Subsequently, all images were averaged over.

## Estimation of decay times from autocorrelation functions in HS-AFM-LS data

Each kymograph was divided into 600-ms segments before ACF analysis, that is, 320 lines. This segmentation was chosen to align with the short time scales underlying FG-domain dynamics. The ACFs $G(\tau)$ were applied to all kymographs to quantify height changes, expressed as[67]:

$$G(\tau) \equiv \sum_{n=1}^{N-m} \frac{(z_n - \bar{z})(z_{n+m} - \bar{z})}{N - m}$$

where $m$ and $n$ refer to the number of scan lines in the kymograph, $\tau \equiv m\Delta t$, $N$ is the total number of scan lines, $z_n$ is the pixel intensity at the given scan line and $\bar{z}$ is the mean value of $z_n$. The larger the $m$, the lower the number of data points that are available to compute the ACF. Therefore, time-lag length of ACF was limited to less than $N/2$ for the ACF accuracy. Each ACF was normalized to the value at lag zero and an ACF heat map was generated from each kymograph. Given that the centre of the kymograph on the fast-scan axis is the centre of the NPC, the heat map was split at the centre and averaged over the corresponding radii to produce an averaged heat map. The $y$ and $x$ axis are the radius and time lag, respectively. Subsequently, the average ACF plot at each radius was fitted with a single exponential decay model to estimate a decay time ($\tau$; Extended Data Fig. 2). To exclude possible intrinsic HS-AFM noise, the average ACF plot was fitted starting at lag one because the autocorrelation coefficient at this point is primarily influenced by intrinsic noise. Furthermore, given that autocorrelation coefficients become less reliable with increasing time lag, only those lying above the 95% confidence interval threshold were fitted[122]. To illustrate average $\tau$ values as a function of distance from the centre, the $\tau$ values estimated from each HS-AFM line scan were averaged over distance intervals of 1 nm.

## Measurements of NPC and central plug diameters

The NPC and CP diameters were measured from 'zoomed-out' images of individual NPCs (for example, Extended Data Fig. 1) using Fiji.

Cross-sectional line profiles were extracted across opposing pairs of octants, yielding four scaffold profiles per NPC. The NPC diameters were determined by measuring peak-to-peak distances across the highest points of the scaffold. The CP diameters were measured from the same line profiles. The resulting NPC and CP diameter values were plotted as histograms and fitted with a single Gaussian function to obtain the mean and standard deviation.

## Average height profiles

The average cross-sectional profile of the NPC (Figs. 1g and 2e,k) was obtained by averaging over all cross-sectional line profiles extracted from four opposing pairs of octants per NPC, across all NPCs. All average height profiles at the CNT were extracted from kymographs at each distance from the pore centre and averaged along the time axis. This process was repeated across multiple NPCs. The resulting average height profile was then overlaid onto the scaffold's average profile. An exception was made when calculating the average cross-sectional profiles of the CNT in Fig. 1g because HS-AFM images and not kymographs were obtained in some instances. In this case the average cross-sectional profile of the CNT was obtained by averaging all cross-sectional line profiles extracted from four opposing pairs of octants per NPC, across all NPCs in each condition.

## Central plug-tracking and positional distribution

Custom Python routines were employed to track the movement of the CP or Kap95 along the fast-scan axis in the kymograph (line-by-line). A low-pass filter was first applied to the kymographs along the fast-scan axis, facilitating efficient identification of the local maximum position of the CP or Kap95 over time. Furthermore, the detection range along the line is limited specifically to the CT region, excluding the scaffold region. Assuming movement of the CP or Kap95 within the line, the following criteria were employed. If only one local maximum was detected, its position was taken. In cases where two or more local maxima were detected, the position of the highest local maximum was taken. If no local maximum was detected, the line was skipped. Trajectories from all kymographs were then plotted as a histogram depicting the distribution as a function of distance from the centre in nanometres, based on the pixel size.

## HS-AFM-LS signal analysis

HS-AFM control signals were extracted using a PicoScope 5000 series 5444D MSO oscilloscope (Pico Technology) and its software (PicoScope v7.1.50) from supported lipid bilayers (on mica) with holes, bare DNA origami pores and −CP WT NPCs. The holey lipid bilayer and the bare DNA origami pore feature pore-like depressions and were chosen as quality controls for the −CP WT NPC (Extended Data Fig. 3).

**Estimation of applied force.** The average magnitude of tip force, $F$, exerted onto each sample was calculated using the equation

$$F = \frac{k_c A_{free}}{2Q} \sqrt{1 - \left(\frac{A_{real}}{A_{free}}\right)^2},$$

where $k_c$ is the HS-AFM cantilever spring constant (0.1 N m$^{-1}$) and $Q$ is its quality factor (approximately 2)[59]. $A_{free}$ and $A_{real}$ are the free cantilever oscillation amplitude and the real-time cantilever amplitude, respectively, where $A_{free}$ is typically 6 nm (Extended Data Fig. 3d).

**Estimation of energy loss.** To assess the impact of tip-sample contact, the mean energy loss per tap of each HS-AFM cantilever oscillation ($E$) can be calculated using[123]

$$E = \frac{k_c A_{free}^2}{2Q} \left\{1 - \left(\frac{A_{real}}{A_{free}}\right)^2\right\}.$$

For $A_{free}$ = 3 nm, the average energy losses are about 16 $k_{B}T$ for lipid bilayer holes, and 6 $k_{B}T$ for bare DNA origami pores as well as −CP WT NPCs (where $k_{B}$ is Boltzmann constant and $T$ is 298 K; Extended Data Fig. 3). For comparison, the repulsive energy barrier associated with the FG domains of NPCs is estimated to be approximately 10 $k_{B}T$ (refs. 124,125). This allows for the HS-AFM to reliably image the 'soft' NPC barrier without energetically altering it. The energy transferred from the tip disperses among multiple degrees of freedom into the surrounding buffer[59,68]. Intrinsically disordered proteins, with more degrees of freedom than ordered proteins, distribute this energy more efficiently, reducing their susceptibility to tip-sample contact damage[69].

## In vitro phase separation of FG hydrogels for HS-AFM

Nup100FG and Nup116FG hydrogels were prepared as previously described[100]. Nup100FG (100 μM; amino acids 1–580) and Nup116FG (110 μM; amino acids 1–725) were purified[83] and kept in a storage buffer (100 mM Tris–HCl, 2 M guanidine–HCl, pH 8.0 and 10% glycerol). For HS-AFM experiments, Nup100FG or Nup116FG assemblies were formed by placing a droplet (2.9 or 3.2 μl) of TBS buffer (150 mM NaCl and 50 mM Tris–HCl, pH 8) on freshly cleaved mica, followed by mixing with 0.1 μl of 100 μM Nup100FG or 110 μM Nup116FG, respectively. This yielded a final concentration of 3.3 μM for each protein. The samples were incubated at room temperature for 1 h and then gently rinsed with the same buffer used for dilution before immediate imaging. Subsequent HS-AFM measurements were carried out on particles with average hole densities of approximately 31 holes μm$^{-2}$ ($n$ = 14) for Nup100FG and 87 holes μm$^{-2}$ ($n$ = 9) for Nup116FG. In some cases, Kap95 was added to the imaging buffer to achieve a final concentration of 1 μM Kap95.

## FG-hydrogel permeation assay

FG hydrogels were prepared as described earlier with unlabelled Nup100FG or Nup116FG supplemented with 5% of their FMAL-labelled counterparts (Nup100FG-FMAL and Nup116FG-FMAL) at the carboxy-terminal cysteine. The total NupFG concentration in droplets containing phase-separated hydrogel particles was 3.3 μM, consistent with samples prepared for HS-AFM. The NupFG concentration within hydrogel particles was estimated by fluorescence calibration of FMAL[126], yielding 9.9 ± 6.1 mM for Nup100FG and 16 ± 7 mM for Nup116FG. Kap95 was labelled with Alexa Fluor 568 C5 maleimide (A20341, Thermo Fisher; Kap95–AF568) and purified on a spin column (Princeton separations). The labelling degree was 1:1, verified using a NanoDrop spectrophotometer (Thermo Fisher). For transport and FRAP assays, Kap95–AF568 in 150 mM NaCl and 100 mM Tris pH 8 was added to the solution containing preformed NupFG particles. The final concentration of Kap95–AF568 in Nup116FG particles was 1.58 ± 0.2 μM, confirmed using fluorescent calibration of Kap95–AF568. As a control, 30 μM BSA labelled 1:1 with Alexa Fluor 647 C2 maleimide (A20347, Thermo Fisher; BSA–AF647) was used. Hydrogel solubility was tested by adding 10% 1,6HD (Sigma) to Nup100FG-FMAL or Nup116FG-FMAL particles. Separately, 9.7 μl of 10% 1,6HD in TBS was mixed with 0.3 μl of 110 μM Nup100FG-FMAL or Nup116FG-FMAL to assess whether 1,6HD interfered with hydrogel formation. Solubility tests were also performed in TBS buffer containing 10% polyethylene glycol 3350 (Sigma-Aldrich, P4338-2KG) to examine the effects of crowding agents on hydrogel stability.

## Fluorescence recovery after photobleaching

A point-scanning confocal LSM880 inverted microscope with an Airyscan detector (Zeiss) built on a Zeiss Axio Observer stand (Intelligent Imaging Innovations GmbH) was used to perform FRAP measurements. The system is equipped with a 1.4 numerical aperture (NA) ×63 Plan-Apochromat oil-immersion objective (Plan-Apochromal ×63/1.4 Oil DIC M27), an EMCCD camera (Evolve(R) 512, Photometrics) and a humidified climate control system at 25 °C. A round 1 μm$^2$ region within each condensate was chosen for bleaching. Bleaching was

performed using three 10-ms pulses per pixel with a solid state 488 nm laser (10 mW) for Nup100FG-FMAL and Nup116FG-FMAL or a 555 nm laser (10 mW) for Kap95–AF568. Sequential images were acquired at intervals of 5 s, capturing five frames before and 115 images after the bleaching event. The sample was illuminated at 30% power (10 mW) with either the 488 nm or 555 nm laser, as appropriate.

Image analysis was conducted using ImageJ, ensuring that movies were free of oversaturated pixels (HiLo) and corrected for condensate mobility (stack registration). For each frame, time stamps and fluorescent intensities of the bleached area, whole condensate and background were extracted and compiled in an EXCEL file. Recovery curves were plotted and analysed using the easyFRAP software[127,128].

## In vivo nucleocytoplasmic transport reporter assays

Strains expressing SV40NLS–GFP-PrA were cultured at 30 °C on synthetic dropout medium supplemented with 0.25 mg ml$^{-1}$ adenine hemisulfate. Strains expressing Kap95–mNeonGreen were cultured at 30 °C on YPD supplemented with 0.2 mg ml$^{-1}$ adenine hemisulfate. Overnight cultures with an OD$_{600}$ of 0.6–1.0 were diluted 20-fold into fresh medium and cultured for 4 h. The cells were collected at 23$g$ for 5 min, resuspended in a small amount of fresh medium, placed into eight-well poly-L-lysine-coated slides (Ibidi GmBH) and visualized using a point-scanning confocal LSM880 inverted microscope with an Airyscan detector (Zeiss) built on a Zeiss Axio Observer stand. Passive permeability for MG4 in the W303, ΔFG and Nsp1FGx2 strains was determined as described[13]. For the Δnsp1 p410-Nsp1FGx2:KanMX strain, the medium was supplemented with 200 μg ml$^{-1}$ G418. Before microscopy analysis, 1 ml cell yeast culture was centrifuged and a fraction of the cells were resuspended and mounted on glass slides.

## In vivo localization assay of NupFG domain biomolecular condensates

**Strains and growth conditions.** The budding yeast *S. cerevisiae* strains used in this study are listed in Supplementary Table 3. To assess the localization of the overexpressed NupFG domains, cells containing the pGAL-GFP-NupFG::His3 plasmids were cultured at 30 °C for one day on synthetic defined (SD) medium lacking histidine, containing 2% glucose (wt/vol), and then cultured for one day on medium containing 2% raffinose (wt/vol) as the carbon source. On the day of the experiment, NupFG expression was induced in exponentially growing cells with 1% (wt/vol) D-galactose for 1 h. pACM021–GFP plasmid backbone, derived from pUG34, was used for the overexpression of the different NupFG domains. FG domain segments were selected as described[12], including Nup153FG[83] (details in Supplementary Table 4).

To assess the localization of the different full-length FG Nups tagged with GFP at the C terminus and expressed under their endogenous promoters, cells were cultured in 2% D-glucose SD medium lacking histidine.

To assess the localization of the overexpressed GFP–Nup100FG in the W303, Mlp1, ΔFG and Nsp1FGx2 stains, cells containing the genome-integrated TEFF-mCherry-WALP-HDEL::Leu2 (nuclear envelope–endoplasmic reticulum marker) and the replicative plasmid pGAL-GFP–Nup100FG::Ura3 were cultured in SD medium lacking leucine and uracil. For the ΔNsp1 p410-Nsp1FGx2:KanMX strain, the medium was supplemented with 200 μg ml$^{-1}$ G418. Overexpression of Nup100FG was induced as explained previously (1% (wt/vol) D-galactose for 1 h) before imaging.

For assessing the solubility of GFP–Nup100FG condensates, GFP–Nup100FG was overexpressed for 1 h in the indicated strains and cells were treated for 10 min with 10% 1,6HD (240117, Sigma-Aldrich), an aliphatic alcohol that dissolves liquid particles, before imaging.

**Imaging and data analysis.** Imaging was done on a DeltaVision deconvolution microscope (Applied Precision (GE)) using InsightSSITM solid state illumination at 488 and 594 nm and an Olympus UPLS Apo

×100 oil objective with 1.4 numerical aperture (NA). Detection was done with a PCO-edge sCMOS camera (Photometrics). Image stacks (30 stacks of 0.2 µm) were deconvolved using softWoRx (Cytiva, Resolve3D softWoRx-Acquire version 7.0.0 release RC6). Data were analysed using ImageJ. Unless otherwise stated, three independent repeats of each experiment were performed. For assessing Nup100FG localization, condensates of fifty cells were analysed for each replicate using ImageJ. For assessing Nup100FG solubility to 10% 1,6HD, the condensates of untreated cells and cells treated with 10% 1,6HD were quantified. Further details about the number of samples can be found in the corresponding figure legends. Graphs and statistical analyses were generated using the Prism software (GraphPad, version 10.4.2).

**Expression levels.** To assess GFP–Nup100FG protein levels by western blotting, budding yeast cells overexpressing GFP–Nup100FG for 1 h were collected by centrifugation, washed with PBS and frozen in liquid nitrogen. Cell pellets were resuspended in HEPES buffer (50 mM HEPES pH 7.5, 100 mM NaCl, 2.5 mM MgCl$_2$, 10 mM DTT and 10% glycerol) supplemented with protease inhibitors (10 mM phenylmethylsulfonyl fluoride and cOmplete-EDTA protease inhibitor cocktail) and broken with glass beads using a fast-prep homogenizer. The lysate was clarified by centrifugation at 1,500g for 3 min. The concentration of total protein in whole-cell extracts was determined using a Pierce BCA protein assay kit (10678484, Thermo Scientific).

Equal amounts of whole-cell lysates of the strains W303, Mlp1, ΔFG and Nsp1FGx2 were diluted in 2×protein loading dye buffer and boiled for 15 min. Subsequently, the samples were separated via SDS–PAGE (Stain free gels, 1610183, BioRad) and transferred to polyvinylidene difluoride membranes for 1 h. The membranes were blocked with 2.5% BSA in PBS-T (PBS containing 0.1% Tween-20), incubated overnight with primary (monoclonal mouse anti-GFP; 1:2,500; sc-9996, Santa Cruz) and secondary (anti-mouse; 1:2,500; sc-516102, Santa Cruz) antibodies, and revealed with ECL using the Chemidoc imaging system (BioRad). Band intensities following immunoblotting and detection using chemiluminescence reagents (ECL) were quantified with ImageJ (1.54p). Individual results from all three repeats are shown together with the mean and s.d. For statistical analysis, we used an unpaired Student's t-test. Further details on the number of samples can be found in the corresponding figure legends.

### Depletion and rebinding of Kap95 at isolated WT NPCs

**Depletion of endogenous Kap95.** Endogenous Kap95 (endoKap95) was analysed in NPCs isolated from an Mlp1ppxPrA pNup84–mCherry strain expressing Kap95–GFP under its native promoter. The isolated NPCs were incubated in PBS buffer at 4 °C for 0, 24, 48 and 72 h. After each incubation, a 4-µl droplet containing isolated NPCs was deposited onto a clean poly-L-lysine (0.01%)-treated coverglass along with 0.2-µm TetraSpeck beads (T7280, Thermo Fisher) as positional markers. The samples were incubated at room temperature for 10 min, followed by BSA passivation and immunostaining. For staining, the samples were first incubated with monoclonal mouse anti-mCherry (1:100; (4B3) MA5-32977, Thermo Fisher) conjugated to AF568 maleimide (A20341, Thermo Fisher), followed by gentle rinsing with PBS buffer. Next, polyclonal rabbit antibodies to eGFP (1:20; CAB4211, Thermo Fisher) conjugated to AF647 hydroxylamine (A30632, Thermo Fisher) were applied for 1 h at room temperature, followed by a final rinse in PBS buffer. To assess the non-specific binding of the anti-mCherry and anti-eGFP antibodies, control samples were subjected to the same protocol in the absence of isolated NPCs (Extended Data Fig. 4a,b).

**Rebinding of exogenous Kap95.** The capacity to rebind exogenous Kap95 (exoKap95) was analysed in NPCs isolated from an Mlp1ppxPrA Nup82–GFP strain after incubation in PBS buffer at 4 °C for 0, 24, 48 and 72 h. After each incubation, a 4-µl droplet containing isolated

NPCs was deposited onto a clean poly-L-lysine (0.01%)-treated coverglass along with 0.2-µm TetraSpeck beads as positional markers. The samples were incubated for 10 min at room temperature, followed by BSA passivation and immunostaining. For staining, the samples were first incubated with monoclonal mouse antibodies to GFP (1:20; 11814460001, Roche) and secondary antibodies (AF568-conjugated anti-mouse; 1:200; A11004, Thermo Fisher), followed by gentle rinsing with PBS buffer. Next, 100 nM Kap95 labelled with FLUX680 maleimide (FX680-0003-1MG, abberior; exoKap95–FLUX680) was applied for 30 min at room temperature and rinsed with PBS buffer. To assess the non-specific binding of monoclonal mouse anti-eGFP, AF568-conjugated anti-mouse and Kap95–FLUX680, control samples were subjected the same protocol in the absence of isolated NPCs (Extended Data Fig. 4c,d).

**Microscopy.** Samples were chemically fixed with 4% formaldehyde before measurements. Confocal measurements were performed using a point-scanning confocal LSM880 inverted microscope with an Airyscan detector (Zeiss) built on a Zeiss Axio Observer stand. The system is equipped with a 1.4 NA ×63 plane apochromat objective (Plan-Apochromal ×63/1.4 Oil DIC M27), EMCCD camera (Evolve(R) 512, Photometrics) and a humidified climate control system at 25 °C. The images were taken using 2% power of solid state 488 nm laser (10 mW) for GFP or 555 nm laser (10 mW) for AF568-labelled antibodies and 0.5% power of 633 nm laser (5 mW) for Kap95–FLUX680. The collected images were analysed using ImageJ. The mean fluorescence was measured within the regions of interest where the signals for NPCs (laser 568 nm) and endoKap95 or exoKap95 (laser 633 nm) were detected and co-localized. The region-of-interest selection was done partially using the ImageJ plug-in 'particle analysis' or additionally verified manually.

**Analysis of endoKap95 depletion.** Mean fluorescence intensities of anti-eGFP–AF647 were measured at NPCs co-localized with anti-mCherry–AF568. The observed signals were significantly higher than the baseline signal of the anti-eGFP–AF647 signal when NPCs were absent (Extended Data Fig. 4a–d), confirming specificity. The dissociation of the persistent endoKap95 fraction from isolated NPCs follows the equation:

$$[\text{endoKap95}]_t = [\text{endoKap95}]_0 \times e^{-\frac{\ln 2}{t_{1/2}} \times t}$$

where $[\text{endoKap95}]_t$ represents the remaining endoKap95 in NPCs after an incubation time $t$, $[\text{endoKap95}]_0$ is the initial amount of strongly bound endoKap95 at $t = 0$ and $t_{1/2}$ is the half-life of the bound complexes. Assuming the anti-eGFP–AF647 signal linearly scales with $[\text{endoKap95}]_t$ at $t = 0, 24, 48$ and $72$ h, the fitted half-life is $t_{1/2} = 20.1$ h, which is comparable to previous estimates[16].

**Analysis of exoKap95 rebinding.** Mean fluorescence intensities of exoKap95–FLUX680 co-localized with anti-mouse–AF568 were measured at isolated NPCs incubated in PBS for $t = 0, 24, 48$ and $72$ h. The capacity for NPCs to rebind exoKap95 after depleting endoKap95 can be modelled by:

$$[\text{exoKap95}]_t = [\text{exoKap95}]_0 +$$

$$([\text{exoKap95}]_\infty - [\text{exoKap95}]_0) \times \left(1 - e^{-\frac{\ln 2}{t_{1/2}} \times t}\right)$$

where $[\text{exoKap95}]_t$ represents the amount of exoKap95 that bound to NPCs that had been pre-incubated in PBS for a time $t$ (the $y$- and $x$-axis variables shown in Extended Data Fig. 4d, respectively), $[\text{exoKap95}]_0$ is the initial amount bound to NPCs at $t = 0$, $[\text{exoKap95}]_\infty$ is an extrapolated value that estimates the saturated amount of bound exoKap95 and $t_{1/2}$ is fixed to 20 h (from "Analysis of endoKap95 depletion").

## Simplified NPC model

A simplified NPC model was implemented in our open-source Integrative Modeling Platform[129] software (http://integrativemodeling.org)[130–132] using a previously published protocol for Brownian dynamics simulations[7,14,133]. The simulated components include a static toroidal ring, the nuclear envelope and 32 flexible FG domains enclosed within a bounding box of $150 \times 150 \times 150$ nm$^3$. The configuration of these components is fully specified by a configuration vector **X** that includes their spatial coordinates, their orientation vectors and the values of some auxiliary variables, as detailed in the section "Interactions among model components". The nuclear envelope was represented as a 30-nm slab with a toroid pore (major radius, $r_{major} = 11$ nm; minor radius $r_{minor} = 7.5$ nm). The FG domains were anchored across the torus symmetrically in four layers, each consisting of eight FG domains. Each FG domain was represented as a flexible string of beads (spherical particles), each one of which has a radius of 8 Å and encompasses 20 residues. An FG domain was divided into 12 FxFG-like beads in the C terminus and four GLFG-like beads in the amino terminus and was anchored to the C terminus of its globular anchor domain. Each bead also has a single specific interaction site on its surface, representing an FG motif that may interact with other FG motifs. Molecular weights were computed from bead radius ($r_{bead}$) by assuming a uniform protein density ($\rho$) of 1.38 g cm$^{-3}$ (ref. [134]), resulting in molecular weight = $\frac{4}{3}\pi r_{bead}^3 \rho$ kDa. Alternatively, the bead radius could be interpreted as the Stokes radius of the molecule, resulting in molecular weight = $(r_{bead}/6.6)^3$ kDa (ref. [135]), leading to nearly identical values for the molecular weight.

## Interactions among model components

Interactions among the model components were quantified using a coarse-grained potential energy function $U(\mathbf{X})$[133], where **X** is the configuration vector:

$$U(\mathbf{X}) = U_{excluded} + U_{FG-polymer} + U_{FG-cohesiveness}$$

where **X** was omitted on the right side of the equation for simplicity. We now explain each one of these terms in turn.

### Excluded volume term.

$$U_{excluded} = -k_{ex}\left(\sum_{i,j|\hat{d}_{i,j}<0} \hat{d}_{i,j} + \sum_{i|\hat{d}_{i,NE}<0}\hat{d}_{i,NE} + \sum_{i|\hat{d}_{i,BB}<0}\hat{d}_{i,BB}\right)$$

$U_{excluded}$ is a linear excluded-volume potential that penalizes overlaps between model components. The minimal distance between the surfaces of the $i$th and $j$th particles (with radii $r_i$ and $r_j$, respectively) is denoted by $\hat{d}_{i,j}$; it is equal to $d_{i,j} - r_i - r_j$, taking a negative value when the two particles intersect and zero when they touch externally. Similarly, $\hat{d}_{i,BB}$ denotes the minimal distance between the surface of the $i$th particle and the simulation bounding box. The minimal distance between the surface of the $i$th particle and the nuclear envelope is denoted by $\hat{d}_{i,NE}$ and $k_{ex}$ is a force constant of 10 kcal mol$^{-1}$ Å$^{-1}$, allowing a soft overlap of 1–2 Å between the coarse-grained representations of the model components.

### FG polymer term.

$$U_{FG-polymer} = 0.5 \sum_{FG \in FGs} \sum_{i=0}^{n_{FG}-1}\left[k_{b1}(d_{i,i+1} - d')^2 + k_{b2}(d' - d'')^2\right]$$

$U_{FG-polymer}$ is a harmonic bonded interaction potential accounting for the spring-like nature of flexible polymers in general[136] and disordered FG-repeat domains in particular[15,17,23,28,137,138]. The distance between the centres of the $i$th and $(i+1)$th beads in a single FG-repeat domain is denoted by $d_{i,i+1}$. The indirect coupling of $d_{i,i+1}$ to $d'$ via $d'$ was used to include information about the empirical relaxation time for the end-to-end distance between consecutive repeats, analogously to

the coupling of fast-moving variables to auxiliary slow-moving variables in temperature-accelerated molecular dynamics simulations[139,140]. The value of $d'$ was typically set to 30.4 Å. The value of the first coupling coefficient $k_{b1}$ was set to guarantee a small standard deviation $\sigma_1 = 2.0$ Å of the difference between $d_{i,i+1}$ and $d'_{i,i+1}$. The value of $k_{b1}$ was computed by equating the normal and Boltzmann distributions of the spring length, $e^{-0.5(d_{i,i+1}-d')/\sigma_1^2} = e^{-0.5k_{b1}(d_{i,i+1}-d')/k_B T}$. From this equality, we inferred that $k_{b1} = k_B T \cdot \sigma_1^{-2}$ and $\sigma_1 = \sqrt{k_B T k_{b1}^{-1}}$. Distance-dependent contributions to configurational entropy to the right-hand term can be neglected because $d_{i,i+1} - d' \ll d_{i,i+1}$. Similarly, the value of the second coupling coefficient $k_{b2}$ was typically set to 0.0075 kcal mol$^{-1}$ Å$^{-2}$ to guarantee a standard deviation of $\sqrt{k_B T k_{b2}^{-1}} = 8.9$ Å for the distance between two consecutive beads in the FG-repeat domains. The diffusion coefficient $D_{i,i+1}$ of the auxiliary variable $d'$ was set to $k_B T \cdot \tau^{-1} \cdot k_{b2}^{-1}$ Å$^2$ fs$^{-1}$, where $\tau$ is the estimated relaxation time of the distance between two consecutive FG repeats; it was typically set to 50 ns ($50 \times 10^6$ fs) for all FG repeats unless stated otherwise. This computation is based on the relaxation time of a polymer chain being equal to $\zeta \cdot k_{b2}^{-1}$ (ref. [136]), where $\zeta$ is the hydrodynamic friction coefficient of the Brownian dynamics, which is equal to $k_B T \cdot D_{i,i+1}^{-1}$ ('Dynamics of the model').

### FG cohesiveness term.

$$U_{FG-cohesiveness} = \sum_{i,j} L(d_{i_0, j_0})$$

$U_{FG-cohesiveness}$ is an isotropic non-bonded linear interaction potential representing cohesive interactions between pairs of interaction sites on FG repeats. $L(d_{i_0,j_0})$ is a truncated linear restraint on $d_{i_0,j_0}$, the distance between the interaction sites on the $i$th and $j$th FG beads, respectively. It is equal to $0.5(k_{\psi_i} + k_{\psi_j})d_{i_0,j_0}$ when $d_{i_0,j_0} < 0.5(R_{\psi_i} + R_{\psi_j})$ or zero otherwise, where $\psi_i$ and $\psi_j$ are the flavours of the $i$th and $j$th FG beads, respectively; $k_{\psi_i}$ and $k_{\psi_j}$ are the force coefficient for the corresponding flavours, parameterized as $k_{GLFG-like} = 1.47$ kcal mol$^{-1}$ Å$^{-1}$ and $k_{FSFG-like} = 1.32$ kcal mol$^{-1}$ Å$^{-1}$; and $R_{\psi_i}$ and $R_{\psi_j}$ are the maximal range for cohesive interaction of FG repeats for the corresponding flavours, parametrized to 6 Å for all FG flavours.

## Dynamics of the model

Brownian dynamics simulations[141] were implemented in IMP as described previously[7,14,133]. All simulations were conducted at 297.15 K. The force vector $\mathbf{f}(\mathbf{X}_i)$ is acting on the $i$th particle with coordinates X$_i$. It is equal in magnitude and opposite in direction to the gradient $\nabla U(\mathbf{X}_i)$. The coordinates of all non-static particles were updated at each time step using the following discrete integration equation:

$$\mathbf{X}_i(t + \Delta t) = \mathbf{X}_i(t) + \frac{D}{k_B T}\Delta t \mathbf{f}(\mathbf{X}_i) + \sqrt{2D\Delta t}R$$

where $t$ denotes time, $\Delta t$ is the integration time step, $D$ is a translational diffusion coefficient assigned to each bead in units of Å$^2$ fs$^{-1}$ and R is a standard normal random variable with a mean of zero and a s.d. of one. The term $\frac{D}{k_B T}$ is denoted by $\zeta$, the hydrodynamic friction coefficient for the diffusing particles. Torques between interaction sites on FG repeats were integrated similarly to the integration of translational forces but using a rotational diffusion coefficient $D_{rot}$ instead of $D$, specified in units of rad$^2$ fs$^{-1}$. The sum of the torques was multiplied by $\frac{1}{k_B T}D_{rot}\Delta t$, followed by adding it to a random rotation about a uniformly sampled rotation axis; the magnitude of the rotation is a normal random variable with mean zero and s.d. of $\sqrt{6D_{rot}\Delta t}$, approximating an independent rotation around three rotational degrees of freedom of magnitude $\sqrt{2D_{rot}\Delta t}$ each.

To approximate the differences between diffusion rates of molecules of different sizes, we defined the Stokes radius of each bead to equal its radius $r$. We then assigned each bead a translational diffusion coefficient in units of Å$^2$ fs$^{-1}$ using the Stokes–Einstein equation, $D = \frac{k_B T}{6\pi\eta r}$,

where $\eta = 0.92$ mPa·s is the dynamic viscosity of water at 297.15 K. Each bead was also assigned a random rotational diffusion coefficient $D_{rot} = \frac{k_B T}{8\pi\eta r^3}$, in units of rad$^2$ fs$^{-1}$, using the Einstein–Stokes–Debye equation and multiplying η by a constant factor of 3.33 to account for increased viscosity in the crowded molecular environment[142].

## NPC simulator

We conducted a total of 50 independent simulations of the NPC, each lasting a minimum of 200 μs following a warm-up time of 10 μs. The warm-up time was discarded from the HS-AFM analysis to allow the system to reach equilibrium. In total, we simulated the system for at least 10 ms across 300 independent simulations, using some 5,000 h of CPU time on the Hebrew University School of Computer Science and Engineering Phoenix HPC cluster of Linux computers.

## HS-AFM simulator

We developed an HS-AFM simulator that computes simulated HS-AFM images from each NPC simulation trajectory. The simulator mimics the HS-AFM rasterization process, where the HS-AFM tip scans the surface of the NPC pixel after pixel and line after line (Supplementary Fig. 2). The simulated HS-AFM images were computed as follows. First, the full simulation trajectory was broken into consecutive trajectory intervals of time length $\tau_{tip} = 10$ ns, to account for the tip movement times. Second, a 3D density map was computed for each of the consecutive trajectory intervals; the density map is computed from the distribution of particle positions in an $n_{grid} \times n_{grid} \times n_{grid}$ voxel grid, with a voxel size of $1 \times 1 \times 1$ nm$^3$. Third, an $n_{grid} \times n_{grid}$ 2D non-rasterized height map was computed from each density map; the height at each $(x, y)$ coordinate corresponds to the hypothetical HS-AFM signal expected if the tip would be positioned at that coordinate. Finally, a rasterized HS-AFM image is computed from consecutive non-rasterized height maps, by simulating the scanning moving of the tip over the $xy$ plane. The tip initiates at the origin point $(0, 0)$ and advances horizontally along the axis at a rate of $\tau_{tip}$ ns per pixel. We note that the HS-AFM tip is not simulated explicitly, and it does not physically interact with the molecules. Thus, a simplifying assumption of the HS-AFM simulator is that the HS-AFM image acquisition does not influence the NPC dynamics.

## Density map computation

The density maps are computed from the corresponding trajectory interval of length $\tau_{tip}$ as follows. Space is discretized into a voxel grid of $n_{grid} \times n_{grid} \times n_{grid}$ around the centre of the pore. A 3D array of the same size, delineating the densities, is initialized with all entries set to zero. For each simulation component (for example, FG bead), the value of the voxel corresponding to the centre of the component in the grid is incremented by one, occurring twice per $\tau_{tip}$ interval.

## Generation of 2D height maps

The 2D height maps were computed from the density maps as follows. First, a Gaussian filter was applied to the raw density map. This procedure was employed to emulate a tip radius of 1.5 nm, surpassing the dimension of an individual pixel (1 nm). The filtering process entails the utilization of a $3 \times 3$ square convolution kernel with a s.d. (σ) of one. Within each vertical column, we emulated the descent of the tip by aggregating the densities of individual components in the simulations. This summation proceeds from the top and progresses downward along the column until reaching the central plane of the pore. Each density is assigned a weight, contingent on the original component type and other pertinent factors.

The $Z$ value assigned to a column is the maximum height at which the cumulative sum of weighted densities surpasses a pre-determined threshold $D_{max} = 0.001$, as stipulated by the following formula.

For a certain time interval:

$$Z(x,y) = max\left\{ z \in \{40, 41, \cdots, 80\} : \sum_{j=z}^{80} \sum_{i=1}^{n_{FG}} (w_{FG}(i) D_{FG}(x,y,j,i)) \geq D_{max} \right\}$$

where $n_{FG}$ is the number of FG chains, $D_{FG}(x, y, z, i)$ is the density of $i$th FG chain in the voxel $x, y, z$ and $w_{FG}(i)$ is its weight.

## FG chain weights

To address the absence of vertically aligned FG chains in realistic HS-AFM images, a penalty is imposed. This penalty is quantified through the introduction of a 'verticality score'. This score is computed on a per-FG-chain basis during each statistical interval. It is determined by the $z$-axis distance between the highest and lowest voxels wherein the density of the FG chain exceeds zero within the specified time interval. As we would like to penalize high verticality scores, this verticality score is then substituted into the density function $f_{P_1}$ of the normal distribution $N(\mu_1, \sigma_1^2)$. The mean $\mu_1$ was set to zero so that horizontal chains are the most heavily weighted and $\sigma_1$ was set to the root-mean-square end-to-end distance $\sigma_1 = \sqrt{n_{beads} - 1} \, l_{rest} = \sqrt{15} \times (30.4) = 117.738$ Å, where $n_{beads}$ is the number of beads per FG chain and $l_{rest}$ is the resting length between two beads. The final weighting for the $i$th FG chain with a verticality score of $v(i)$ is given by $w_{FG}(i) = f_{P_1}(v(i))$.

## Tip movement along $xy$

In our imaging process, we simulated the movement of a scanning tip across the sample surface, resulting in rasterized images. The tip initiates at the origin point $(0, 0)$ and advances horizontally along the $x$ axis at a rate of $\tau_{tip} = 10$ ns per pixel. This speed, although faster than the experimental HS-AFM speeds, was chosen to balance resolution with practical processing time considerations. The pixel at coordinate $x = 0$, $y = 0$ is obtained from the first non-rasterized map corresponding to time interval 0–10 ns (assuming $\tau_{tip} = 10$ ns), the pixel at coordinate $x = 1, y = 0$ is computed from the next map corresponding to time interval 10–20 ns and so on until the entire line of $n_{grid}$ is complete. On reaching the horizontal boundary of the image, the tip ascends one pixel on the $y$ axis and resets to the $x$-axis origin. The horizontal return time to the starting position is equal to the forward traversal time and thus takes $n_{grid} \times \tau_{tip} = 400$ ns. The reset time between consecutive images was also set at 400 ns.

## Calculation of next-neighbour distance for WT and ΔFG NPCs

We extracted the anchor point coordinates for the FG-repeat domain from the integrative localization map of the yeast NPC, which has a precision of 9 Å (ref. 7). For the WT analysis, we included the anchor sites of Nup1.601–636 (that is, residues 601–636 of Nup1), Nup2.301–350, Nup49.201–269, Nup57.201–286, Nup60.351–398, Nup100.551–575, Nup116.751–775, Nup145.201–225 and Nup159.1082–1116; we omitted the anchor site of Nup42 from the analysis because it is likely to fluctuate spatially. In addition, the anchor sites of Nsp1, Nup159, Nup1 and Nup60 were omitted in the deletion-mutant analysis. For a single anchor site, the nearest-neighbour distance is the minimum Euclidean distance to any other anchor site within a single spoke of the NPC, including other copies of the same anchor site.

## Statistics and reproducibility

The data were tested for normality using a built-in function of GraphPad Prism software (version 10.4.2 or 10.6.1). $P$ values were obtained using an unpaired two-tailed Student's $t$-test or an ordinary one-way ANOVA test, as appropriate; details are provided in the figure legends. No statistical methods were used to pre-determine sample size. No data were excluded from the analyses. The experiments were not randomized. The investigators were not blinded to allocation during experiments and outcome assessment. All experiments were repeated independently at least three times unless otherwise stated.

## Reporting summary

Further information on research design is available in the Nature Portfolio Reporting Summary linked to this article.

## Data availability

Yeast strains and plasmids generated in this study will be distributed without restriction on request. Additional data related to this paper can be found in the following public repositories: HS-AFM data files are available via *Zenodo* at https://doi.org/10.5281/zenodo.15684361 (ref. 143) and mass spectrometry data files are available in PRIDE (accession no. PXD069104). Source data are provided. All other data supporting the findings of this study are available on reasonable request from the corresponding authors.

## Code availability

The full code for the HS-AFM simulator can be found in https://github.com/ravehlab/HS-AFM-Simulation. Codes used for CP tracking are available at https://github.com/toshiya-kozai/Single-Particle-tracking-on-AFM-kymograph.

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

## Acknowledgements

R.Y.H.L. is supported by the Schweizerischer Nationalfonds zur Förderung der Wissenschaftlichen Forschung (Swiss National Science Foundation; grant numbers 310030_201062 and IZCOZ0_220223), the Biozentrum and the Swiss Nanoscience Institute. T.K. is supported by a Swiss Nanoscience Institute PhD Fellowship. J.F.-M. acknowledges grant funding from the National Science Foundation (NSF-1818129) and Spanish Ministry of Science, Innovation and Universities (projects PID2020-116404GB-I00 funded by MCIU/AEI/10.13039/501100011033, and PID2023-149015NB-I00 funded by MCIU/AEI /10.13039/501100011033/FEDER, UE) as well as the Fundación Biofísica Bizkaia and the Basque Excellence Research Centre (BERC) programme of the Basque Government. T.v.E. is supported by an Anderson Center for Cancer Research Fellowship at The Rockefeller University. M.D.-I. is supported by the Investigo Program (Lanbide-Servicio Vasco de Empleo) funded by Next Generation EU (Plan de Recuperación, Transformación y Resiliencia) and a Predoctoral Fellowship from Programa Predoctoral de

Formación de Personal Investigador No Doctor del Departamento de Educación del Gobierno Vasco. B.T.C., M.P.R. and A.S. are supported by NIH P41 GM109824. M.P.R. is supported by NIH R01 GM112108 and R01 GM117212. A.S. is supported by NIH R01 GM083960. P.G. and L.V. are supported by the Netherlands Organization of Scientific Research (grant numbers OCENW.GROOT.2019.068 and VI.C.192.031). Q.F. and C.L. are supported by NIH R01 AI162260. R.E. and B.R. are supported by a Minerva Center grant on Cell Intelligence, Israeli Science Foundation grant 385/24 and a Hebrew University Center for Interdisciplinary Data Science Research grant. We thank A. Steen, T. Otto and T. Bergsma for providing advice and purified Nup100FG and Nup116FG, and E. C. R. Barrientos for providing the plasmid mCherry–WALP. We also acknowledge M. Hondele, B. Fahrenkrog and P. Upla as well as the Imaging and Bio-EM facilities at the Biozentrum for support. Special thanks to Y. Mai for permission to use the photo shown in Fig. 4i.

## Author contributions

R.Y.H.L. and M.P.R. conceived and coordinated the study. T.K., J.F.-M., L.E.K., T.v.E., P.G., J.T., Q.F., C.L., A.S., B.R., L.M.V., M.P.R. and R.Y.H.L. designed the research. T.K., J.F.-M., T.v.E., P.G., L.E.K., M.S., R.E.-M., A.M., W.Z., J.T., R.P., M.D.-I., Q.F., R.E. and B.R. performed the research. T.K., J.F.-M., T.v.E., P.G., L.E.K., M.S., R.E., A.M., W.Z., M.D.-I., Q.F., C.L., B.R., B.T.C., L.M.V., M.P.R. and R.Y.H.L. analysed the data and conceptualized the results. T.K., J.F.-M., T.v.E., B.T.C., L.M.V., M.P.R. and R.Y.H.L. wrote and revised the manuscript with contributions from all authors.

## Competing interests

The authors declare no competing interests.

## Additional information

**Extended data** is available for this paper at https://doi.org/10.1038/s41556-025-01812-9.

**Correspondence and requests for materials** should be addressed to Michael P. Rout or Roderick Y. H. Lim.

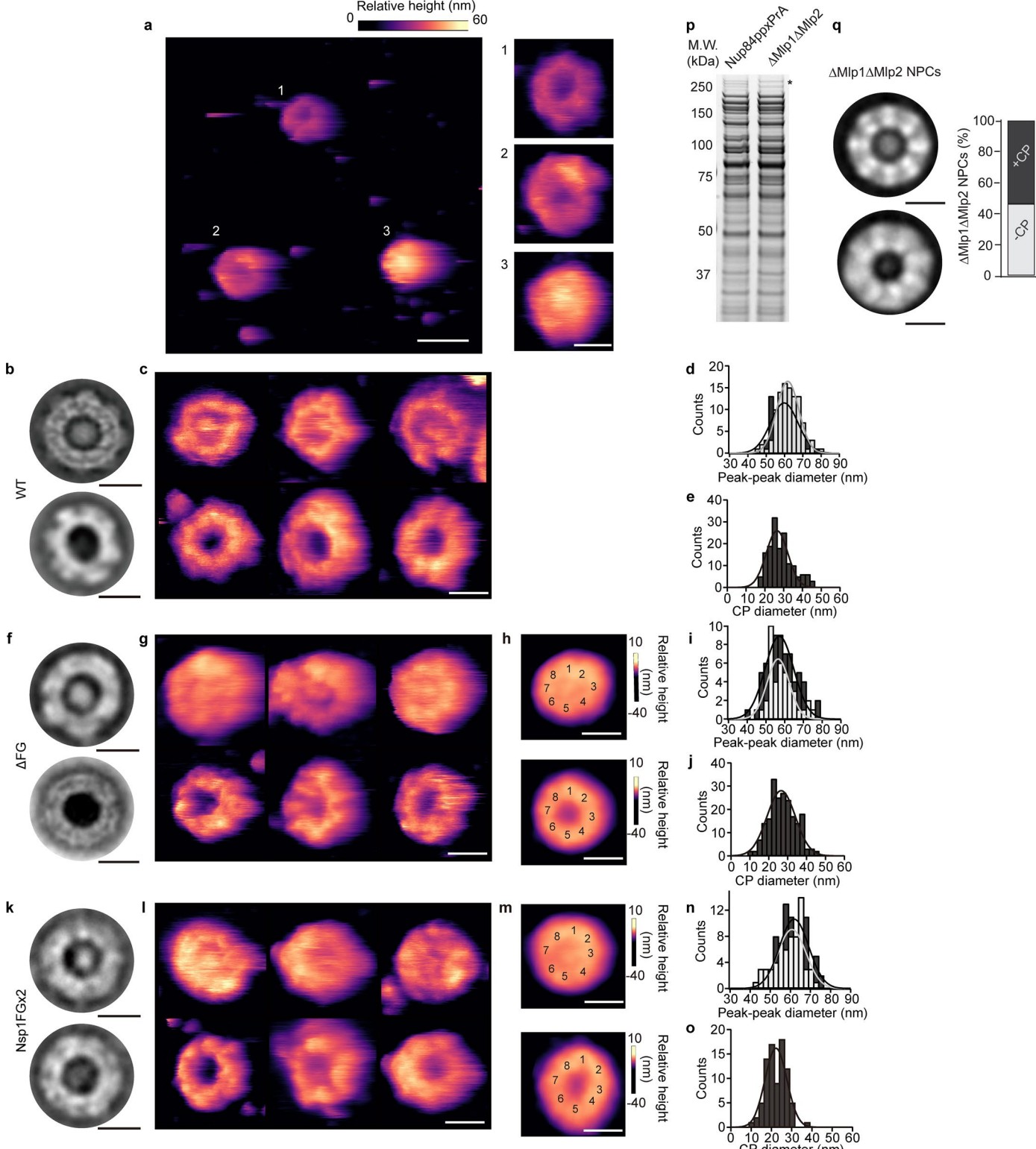

**Extended Data Fig. 1 | See next page for caption.**

**Extended Data Fig. 1 | WT and mutant NPCs retain their overall structure. a**, Raw zoom out image showing three NPCs. Scale bar, 100 nm. Corresponding zoom-in images identify two −CP WT NPCs and a single +CP WT NPC. Scale bar, 50 nm. **b**, 2D class average negative-stain TEM images of +CP (upper; $n$ = 172 NPCs) and −CP (lower; $n$ = 76 NPCs) WT NPCs. **c**, Collage of raw HS-AFM images of +CP (upper) and −CP (lower) WT NPCs. **d**, Peak-to-peak diameters for +CP WT NPCs (black; mean value ± s.d. = 61.8 ± 6.8 nm; $n$ = 21 NPCs) and −CP WT NPCs (grey; mean value ± s.d. = 63.0 ± 5.9 nm; $n$ = 22 NPCs). Statistical significance was determined using a two-tailed unpaired $t$-test (p = 0.108). **e**, CP diameters (mean value ± s.d. = 29.4 ± 7.7 nm; $n$ = 52 NPCs). **f**, 2D class average negative-stain TEM images of +CP (upper; $n$ = 133 NPCs) and −CP (lower; $n$ = 17 NPCs) ΔFG NPCs. **g**, Collage of +CP (upper) and −CP (lower) ΔFG NPCs. **h**, Averaged HS-AFM images of +CP (upper; $n$ = 12 NPCs) and −CP (lower; $n$ = 11 NPCs) ΔFG NPCs. **i**, Peak-to-peak diameters for +CP ΔFG NPCs (black, mean value ± s.d. = 59.7 ± 6.6 nm; n = 17 NPCs) and −CP ΔFG NPCs (grey, mean value ± s.d. = 59.0 ± 7.5 nm; $n$ = 10 NPCs). Statistical significance was determined using a two-tailed unpaired $t$-test

(p = 0.321). **j**, CP diameters (mean value is 26.5 ± 10.5 nm; $n$ = 60 NPCs). **k**, 2D class average negative-stain TEM images of +CP (upper; $n$ = 29 NPCs) and −CP (lower; $n$ = 62 NPCs) Nsp1FGx2 NPCs. **l**, Collage of +CP (upper) and −CP (lower) Nsp1FGx2 NPCs. **m**, Averaged HS-AFM images of +CP (upper; $n$ = 10 NPCs) and −CP (lower; $n$ = 9 NPCs) Nsp1FGx2 NPCs. **n**, Peak-to-peak diameters for +CP Nsp1FGx2 NPCs (black, mean value ± s.d. = 62.7 ± 7.0 nm; $n$ = 19) and −CP Nsp1FGx2 NPCs (grey, mean value ± s.d. = 60.2 ± 7.0 nm; $n$ = 16 NPCs). Statistical significance was determined using a two-tailed unpaired $t$-test (p = 0.98). **o**, CP diameters (mean value ± standard deviation = 22.2 ± 7.5 nm; n = 28 NPCs). Scale bars (b to m), 50 nm. **p**, SDS–PAGE analysis of WT and *ΔMlp1ΔMlp2* NPCs purified via Nup84 affinity purification shows loss of Mlp protein band (*). **q**, Left, 2D class average negative-stain TEM images of +CP (upper; $n$ = 156) and −CP (lower; $n$ = 57) *ΔMlp1ΔMlp2* NPCs. Right, Average distribution of NPCs from *ΔMlp1ΔMlp2* between +CP and −CP class averages from total of 1,266 NPCs. Source numerical data and unprocessed blots are provided.

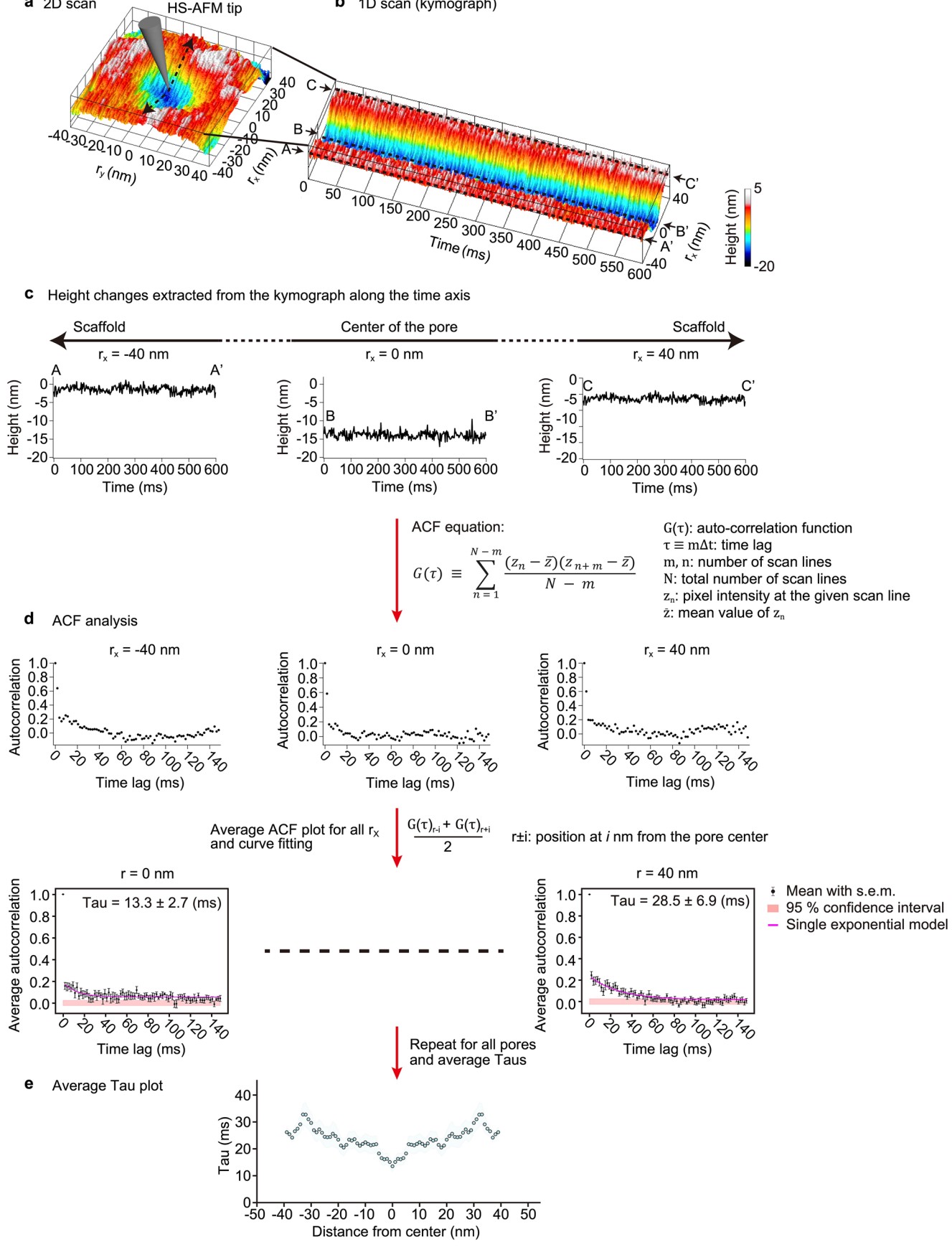

**Extended Data Fig. 2 | See next page for caption.**

**Extended Data Fig. 2 | Autocorrelation function (ACF) analysis workflow.**
**a**, HS-AFM image of an NPC showing the location where the HS-AFM tip captures continuous line scans (HS-AFM-LS) at a rate of 1.875 ms/line (dashed line with arrow heads). Note: Only unidirectional data (for example, left-to-right) is captured. **b**, Corresponding kymograph capturing time-dependent height changes along the specified measurement line. **c**, Three representative time courses (height fluctuations over time) extracted along the black dashed lines in **b** denoted by AA', BB' and CC'. **d**, Normalized autocorrelation coefficients were computed from individual time courses using the ACF equation, averaging between values that correspond to similar radial positions (for example, AA' and CC'). The decay time constant (Tau) was obtained by fitting the ACF by a single-exponential decay model. Only coefficients above the 95% confidence threshold (red band) were fit to ensure reliability. **e**, Average Tau plot calculated across multiple NPCs. See Methods for further details.

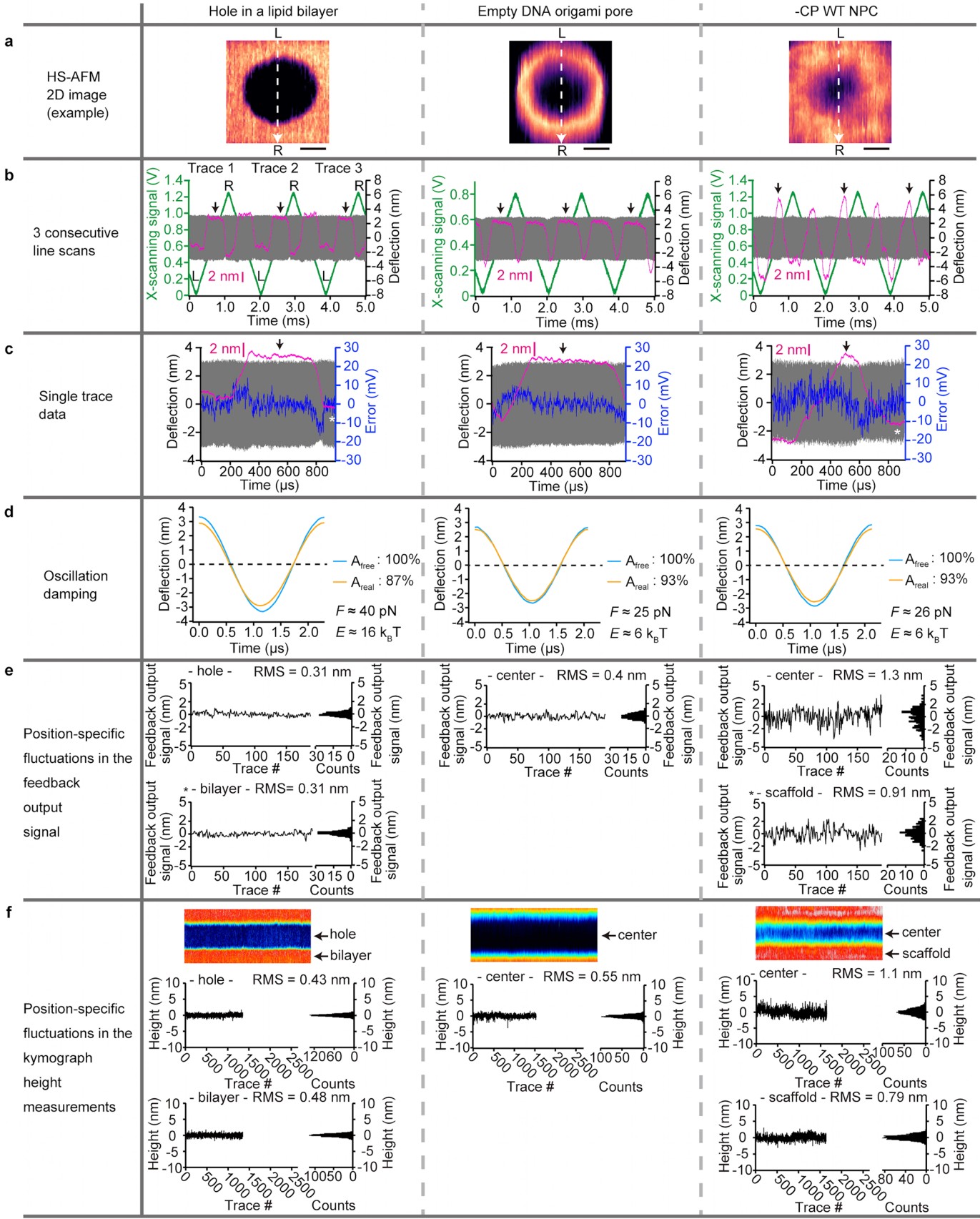

**Extended Data Fig. 3 | See next page for caption.**

**Extended Data Fig. 3 | HS-AFM-LS signal characterization.** All signals were obtained on (i) holey supported lipid bilayers (on mica), (ii) bare DNA origami pores, and (iii) −CP WT NPCs. **a**, Example HS-AFM image indicating the position and left (L)-to-right (R) trace direction of HS-AFM-LS (white dashed arrow). Scale bar, 20 nm. **b**, Plot of feedback output (magenta) versus time over 3 consecutive line scans (1 line scan = 1 trace-retrace cycle; Trace = L to R; Retrace = R to L). Each trace is numbered. Also included are the corresponding deflection (grey) and X-scanning signals (green). Black arrows denote the position within each hole/pore where fluctuations in the feedback output signal are analysed. See **e**. **c**, Single trace data (Trace 1) showing feedback output (magenta), deflection (grey) and error (blue) signals. White asterisks denote a position outside the hole/pore where fluctuations in the feedback output signal are also analysed. See **e**. **d**, Average cantilever motion calculated over several consecutive oscillation cycles. The degree of oscillation damping at the setpoint ($A_{real}$; orange) is contrasted against the free oscillation of the same cantilever in solution ($A_{free}$; cyan). The applied force ($F$) and the energy loss per tap ($E$) are shown in the figure. See Methods for details. **e**, Position-specific fluctuations in the feedback output signal obtained over several traces (Trace 1, Trace 2, Trace 3, …; black arrows in **b** and **c**; white asterisk in **c**). The signal distribution is shown in the accompanying histogram. The RMS amplitude measures the average magnitude of each fluctuation. **f**, Position-specific height fluctuations obtained during HS-AFM-LS as indicated in the accompanying kymographs. The height distribution is shown in the accompanying histogram. Note the general agreement between the RMS amplitude in the feedback output signal (**e**) and the RMS height values in the HS-AFM-LS kymographs, that is, the height signal is the mirror image of the feedback output signal. Source numerical data are provided.

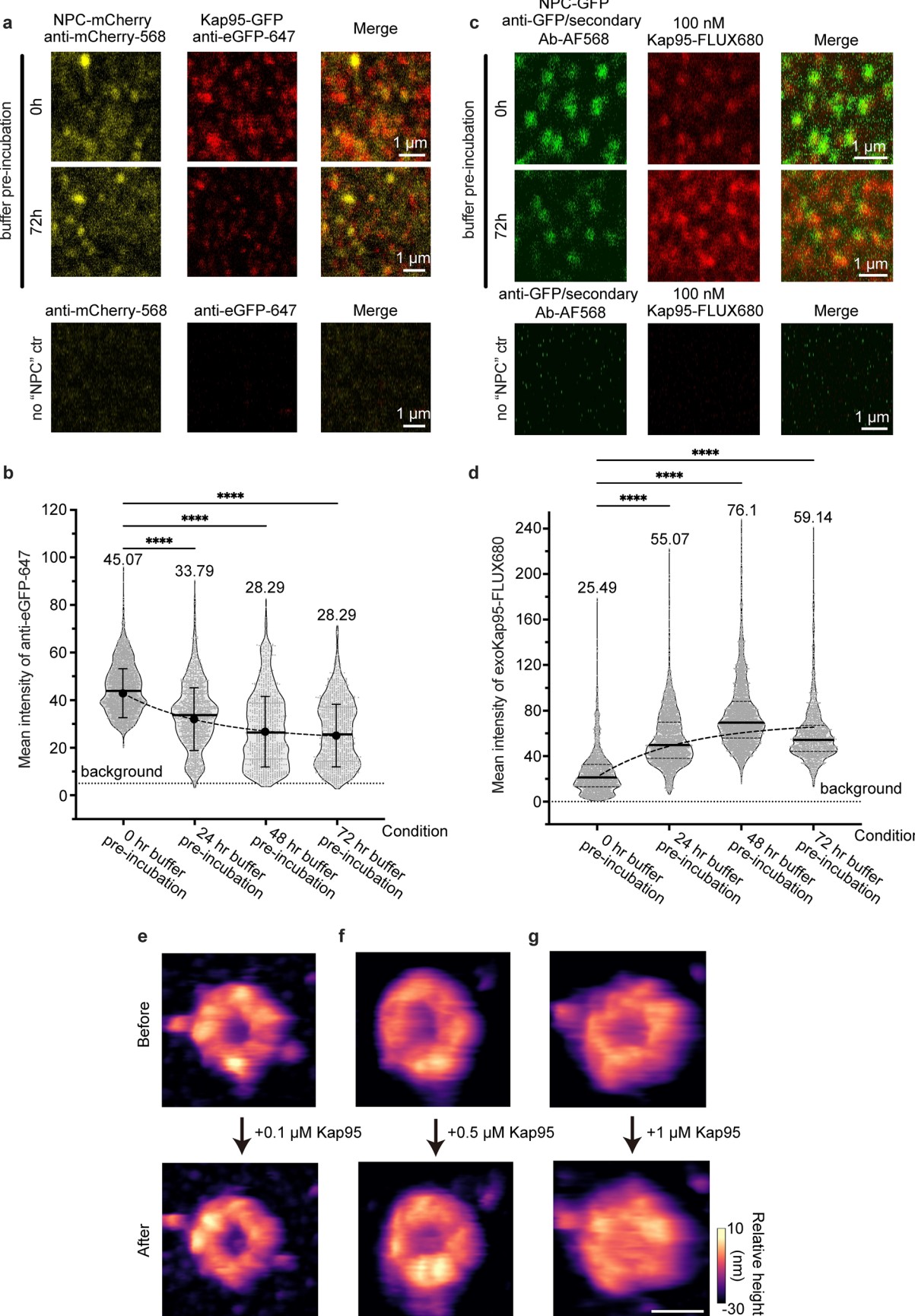

**Extended Data Fig. 4 | See next page for caption.**

**Extended Data Fig. 4 | Depletion of Kap95 from +CP WT NPCs and its rebinding restores the CP in −CP WT NPCs. a,b**, Loss of endoKap95-GFP at Nup84-mCherry NPCs determined after 0, 24, 48 and 72 h incubation in buffer.
**a**, Representative images at 0 and 72 h are shown together with a "no NPC" control to determine the background fluorescence caused by non-specific binding. **b**, Violin plots show mean intensities of anti-eGFP-647 that co-localized with anti-mCherry-568 at each pre-incubation. Horizontal dotted line indicates background fluorescence as shown in **a**. Mean values are indicated. $P$ values from left to right: $1.9 \times 10^{-10}$, $1.9 \times 10^{-10}$ and $1.9 \times 10^{-10}$. Statistical significance was analysed using ordinary one-way ANOVA (****$P$ < 0.0001). No. of biological replicates = 3; $n$ = 1,977, 1,299, 944 and 1,102 co-localized spots from left to right. The half-life of endoKap95-GFP depletion was determined as $t_{1/2}$ = 20.1 h by curve fitting. Error bars = s.d. **c,d**, Capacity for endoKap95-depleted Nup82−GFP NPCs to rebind exoKap95−FLUX680 after incubation in buffer for 0, 24, 48 and 72 h. **c**, Representative images at 0 and 72 h are shown together with a "no NPC"

control. **d**, Violin plots show mean intensities of Kap95−FLUX680 at each time point that co-localized with anti-GFP/secondary anti-mouse-568 after each pre-incubation. Horizontal dotted line indicates background fluorescence as shown in **c**. Mean values are indicated. p values from left to right: $1.8 \times 10^{-10}$, $1.8 \times 10^{-10}$ and $1.8 \times 10^{-10}$. Statistical significance was analysed using ordinary one-way ANOVA (****$P$ < 0.0001). No. of biological replicates = 3; $n$ = 1,794, 2,341, 3,430 and 1,354 co-localized spots from left to right. Close agreement between the predicted behaviour (overlaid dashed curve) and experimental data indicates the NPC's capacity to rebind Kap95−FLUX680 following endoKap95 depletion, consistent with the slow apparent dissociation rate of the CP-forming Kap95 fraction. See Methods for details. **e**−**g**, Representative HS-AFM images showing −CP WT NPCs before and after the addition of varying concentrations of Kap95 (**e**; 0.1 μM, **f**; 0.5 μM and **g**; 1 μM). Scale bar, 50 nm. Source numerical data are provided.

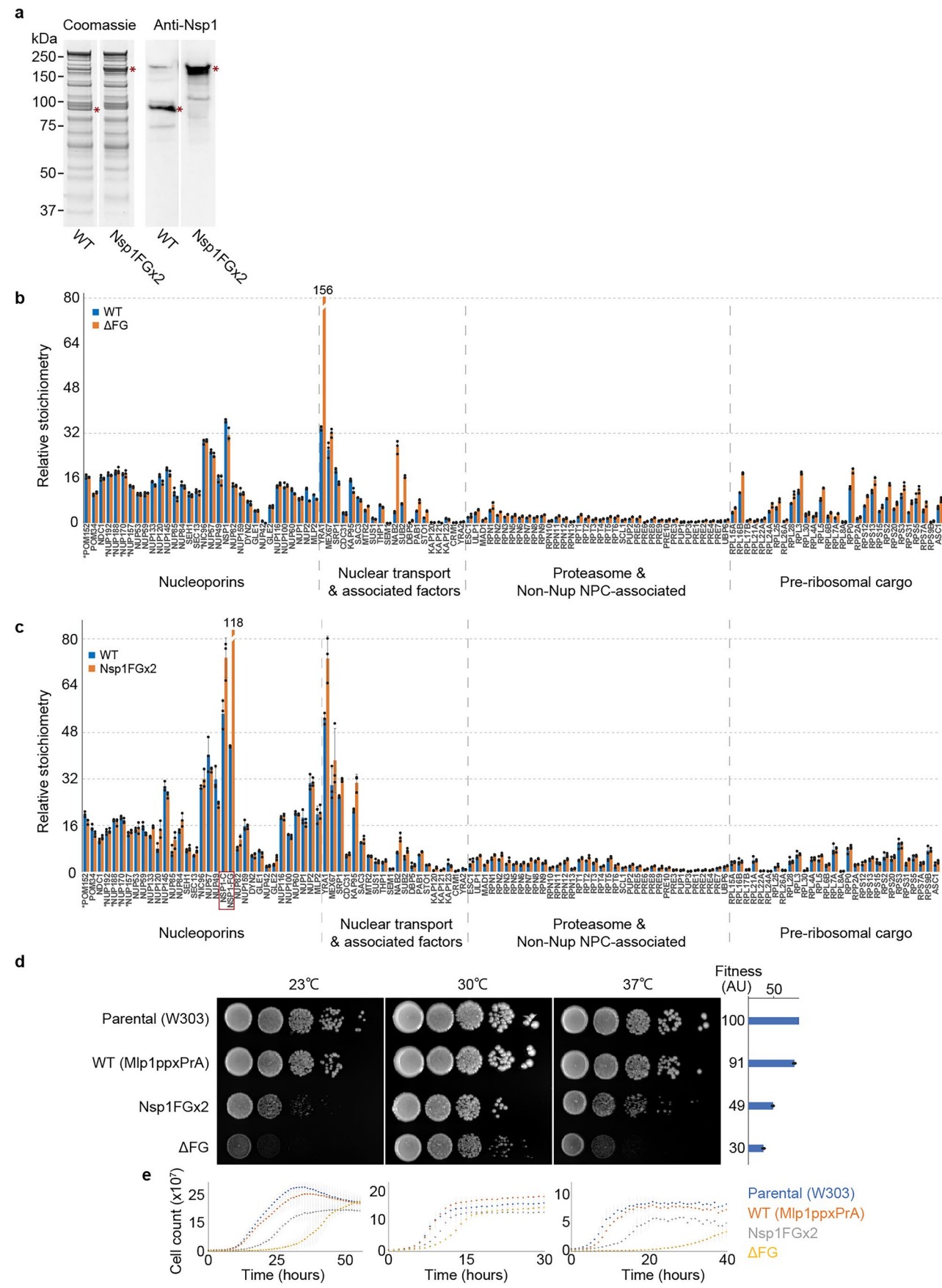

**Extended Data Fig. 5 | See next page for caption.**

**Extended Data Fig. 5 | Characterization and fitness of Nsp1FGx2 mutant cells compared with WT and ΔFG strains. a**, Expression and molecular weight of the Nsp1FGx2 mutant protein in affinity-purified NPCs shown by Coomassie staining (left) and western blot using an anti-Nsp1 antibody (right). Asterisks indicate the position of the wild-type Nsp1 and mutant Nsp1FGx2 proteins. **b**,**c**, Comparative relative stoichiometry of Nups and main associated proteins within affinity-purified NPCs[7] between wild-type (blue bars), (**b**) ΔFG and (**c**) Nsp1FGx2 NPC mutants (orange bars), as determined by label-free MS quantification (at least 5 peptides per protein). Proteins are grouped by functional categories or membership of discrete macromolecular assemblies (in some cases only a selection of components are shown). The bars for Nsp1 and Yra1 were shortened for presentation purposes, with their actual value indicated above. In the Nsp1FGx2 plot, two values are provided for Nsp1 (highlighted with red square), indicating the values obtained when only analysing peptides originated from the C-terminal anchor site of the protein (NSP1-C), or peptides originated only from the FG region of the protein (NSP1-FG), reflecting the levels of incorporated protein and FG repeats within the NPCs respectively.

The handle used for the affinity purification (Mlp1) is not shown. No. of biological replicates = 3. Error bars = s.d. **d**, Spot growth tests at different temperatures (23 °C, 30 °C and 37 °C) for the parental (w303) and mutant strains Mlp1ppxPrA, Nsp1FGx2 Mlp1ppxPrA and ΔFG Mlp1ppxPrA strains. Serial 10-fold dilutions of cells were spotted on YPD plates and grown at the indicated temperatures for 1–3 d. Each growing phenotype was quantified by semiquantitative methods (see Methods), and the obtained value (in arbitrary units [AU]) is shown on the right of each column. Plotted fitness value (mean value ± s.d.) for each measurement is shown on the right. **e**, Growth curves for the same strains and temperatures as in **d** measured in 96-well plates for the times indicated. $OD_{600}$ measurements were transformed to cells ml$^{-1}$ ($\times 10^7$) as indicated in the Methods section. Number of biological replicates for the parental (w303) strain = 5 (23 °C); 5 (30 °C); and 6 (37 °C). Number of biological replicates for the Mlp1ppxPrA strain = 7 (23 °C); 7 (30 °C); and 6 (37 °C). Number of biological replicates for the Nsp1FGx2 Mlp1ppxPrA strain = 4 (23 °C); 3 (30 °C); and 6 (37 °C). Number of biological replicates for the ΔFG Mlp1ppxPrA strain = 3 (23 °C); 3 (30 °C); and 6 (37 °C). Error bars = s.d. Source numerical data and unprocessed blots are provided.

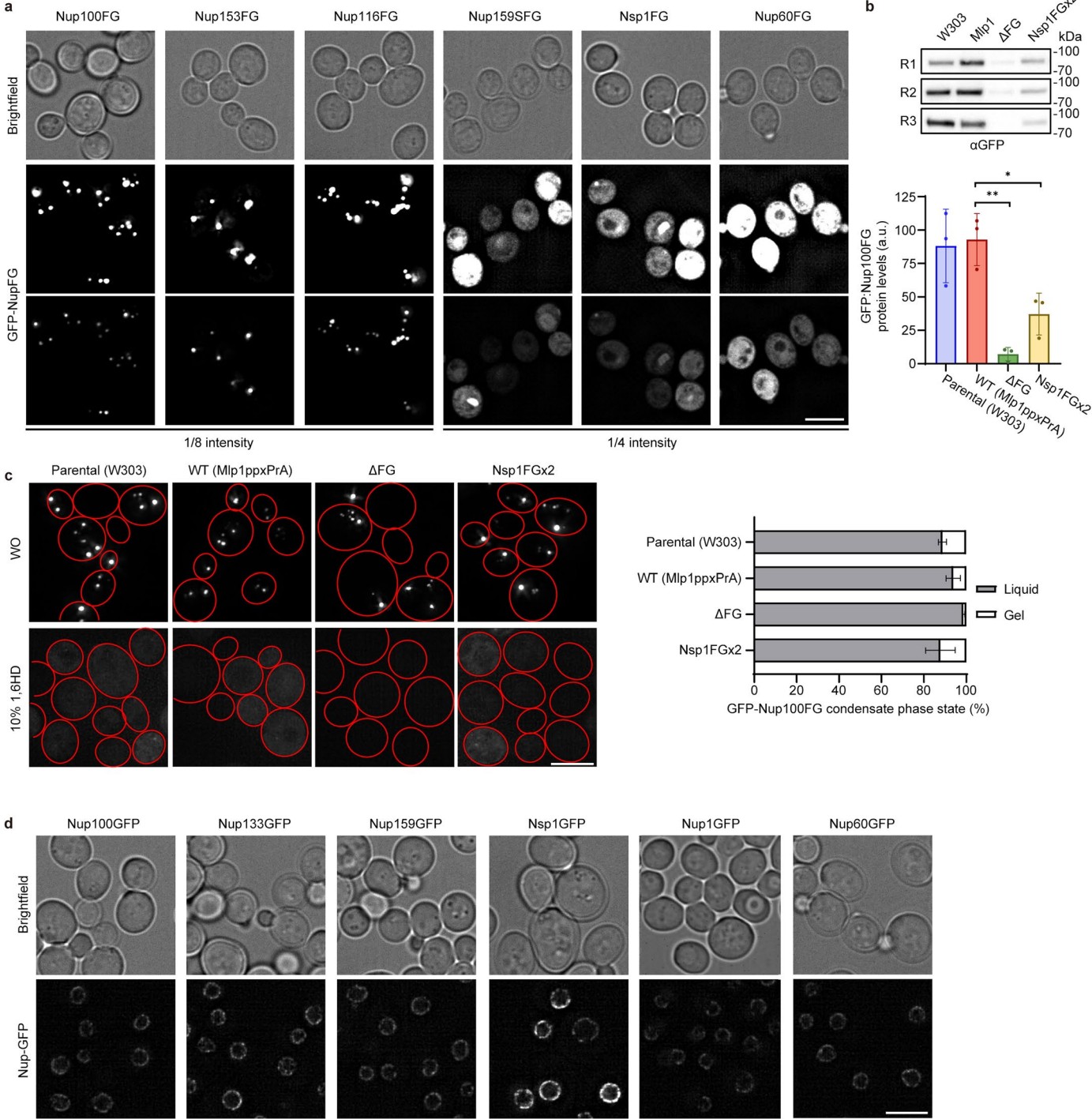

**Extended Data Fig. 6 | Characterization of in vivo FG Nup domains.**
**a**, Cellular localization of different FG domains tagged with GFP at the
N-terminus, overexpressed for 1 h. Maximal projection of four z-stacks, 0.2 μm
interval. Scale bar, 5 μm. The bottom images have 8-fold (Nup100FG, hNup153FG,
and Nup116FG) and 4-fold (Nup159SFG, Nsp1FG, and Nup60FG) lower intensity
compared to the upper panels. Upper panels serve to assess if a punctate NPC
signal becomes visible under the same imaging conditions used in **d** for imaging
full-length Nups. **b**, Western blot analysis of GFP–Nup100FG overexpressed
for 1 h in the indicated strains. Bar plot represents the quantification of GFP–
Nup100FG protein levels relative to protein loading (stain free staining) (no. of
biological replicates = 3; data are the mean value ± s.d.; P values from top to
bottom, $1.81 \times 10^{-2}$, $1.8 \times 10^{-3}$). Statistical analysis was performed by two-tailed
unpaired Student's t-tests. Primary ab: mouse anti-GFP (1:2,500); secondary
ab: anti-mouse (1:2,500). **c**, GFP–Nup100FG was overexpressed for 1 h in the

indicated strains and imaged without (WO) and after 10 min 10% 1,6-hexanediol
(1,6HD) treatment. Cells are outlined in red lines. Graph shows the percentage of
condensates that are in a liquid or gel-phase state (based on their dissolution by
the aliphatic alcohol 10% 1,6HD) (no. of biological replicates = 3; cell numbers are
as follows: W303 (WO, $n = 337$; 1,6HD, $n = 647$), Mlp1ppxPrA (WO, $n = 375$; 1,6HD,
$n = 631$), ΔFG (WO, $n = 548$; 1,6HD, $n = 667$), and Nsp1FGx2 (WO, $n = 576$; 1,6HD,
$n = 769$; data are the mean value ± s.d.; P values for W303, ΔFG and Nsp1FGx2 are
0.0882, 0.0745 and 0.2471, respectively). Statistical analysis was performed by
two-tailed unpaired Student's t-tests. All comparisons to the WT (Mlp1ppxPrA)
were non-significant. Scale bar, 5 μm. **d**, Localization of different full-length
FG Nups, tagged with GFP at the C-term, expressed under their endogenous
promoter. Note: A nuclear rim staining is also visible for Nup1GFP, which is
present in 8 copies per NPC[7]. Maximal projection of 4 z-stacks, 0,2 μm interval.
Scale bar, 5 μm. Source numerical data and unprocessed blots are provided .

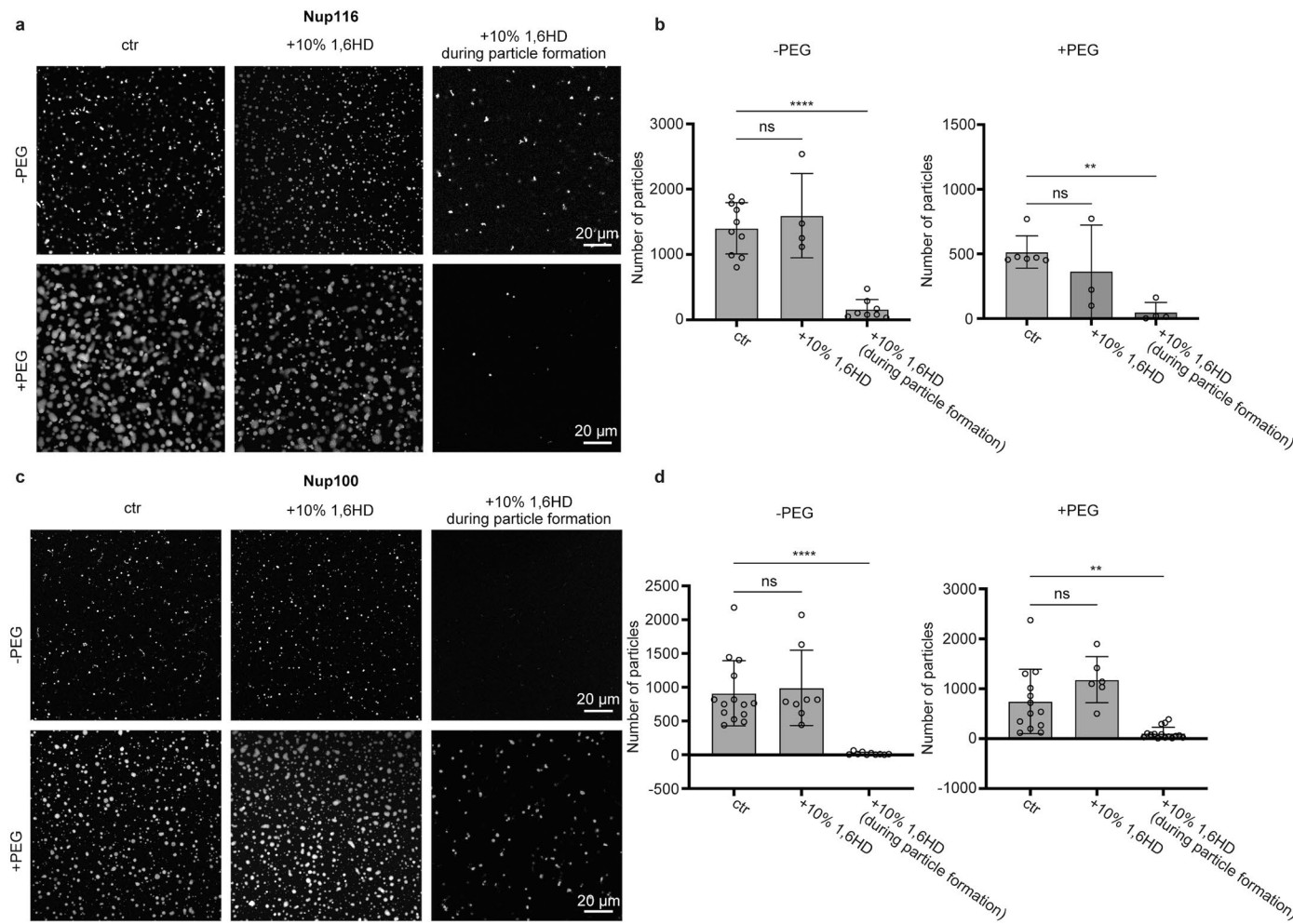

**Extended Data Fig. 7 | Effect of 1,6HD on Nup116FG and Nup100FG hydrogel particles during and after formation. a,c,** 10% 1,6HD has no effect on preformed Nup116FG (**a**) and Nup100FG (**c**) particles. However, 10% 1,6HD inhibits Nup116FG and Nup100FG particle formation. These behaviours are not affected by the absence or presence of PEG. **b,** Nup116FG particle counts for each of the conditions shown. No. of experimental replicates in the absence of PEG = 3; No. of hydrogel particles, $n = 10$ (ctr), 4 (+1,6HD), and 8 (+1,6HD during particle formation). No. of experimental replicates in the presence of 10% PEG = 3; No. of hydrogel particles, $n = 6$ (ctr), 3 (+1,6HD), and 4 (+1,6HD during particle formation). **d,** Nup100FG particle counts for each of the conditions shown. No. of

experimental replicates in the absence of PEG = 3; No. of hydrogel particles, $n = 14$ (ctr), 8 (+1,6HD), and 10 (+1,6HD during particle formation). No. of experimental replicates in the presence of 10% PEG = 3; No. of hydrogel particles, $n = 13$ (ctr), 7 (+1,6HD), and 15 (+1,6HD during particle formation). Error bars = s.d. $P$ values for Nup116FG without PEG are from left to right: 0.648 and $4.64 \times 10^{-6}$, and with PEG: 0.521 and $8.36 \times 10^{-3}$, respectively. $P$ values for Nup100FG without PEG are from left to right: 0.906 and $5.61 \times 10^{-5}$, and with PEG: 0.139 and $7.12 \times 10^{-5}$, respectively. Statistical analysis was performed by an ordinary one-way ANOVA test (ns refers to $P > 0.05$, **$P < 0.01$, and ****$P < 0.0001$). Source numerical data are provided .

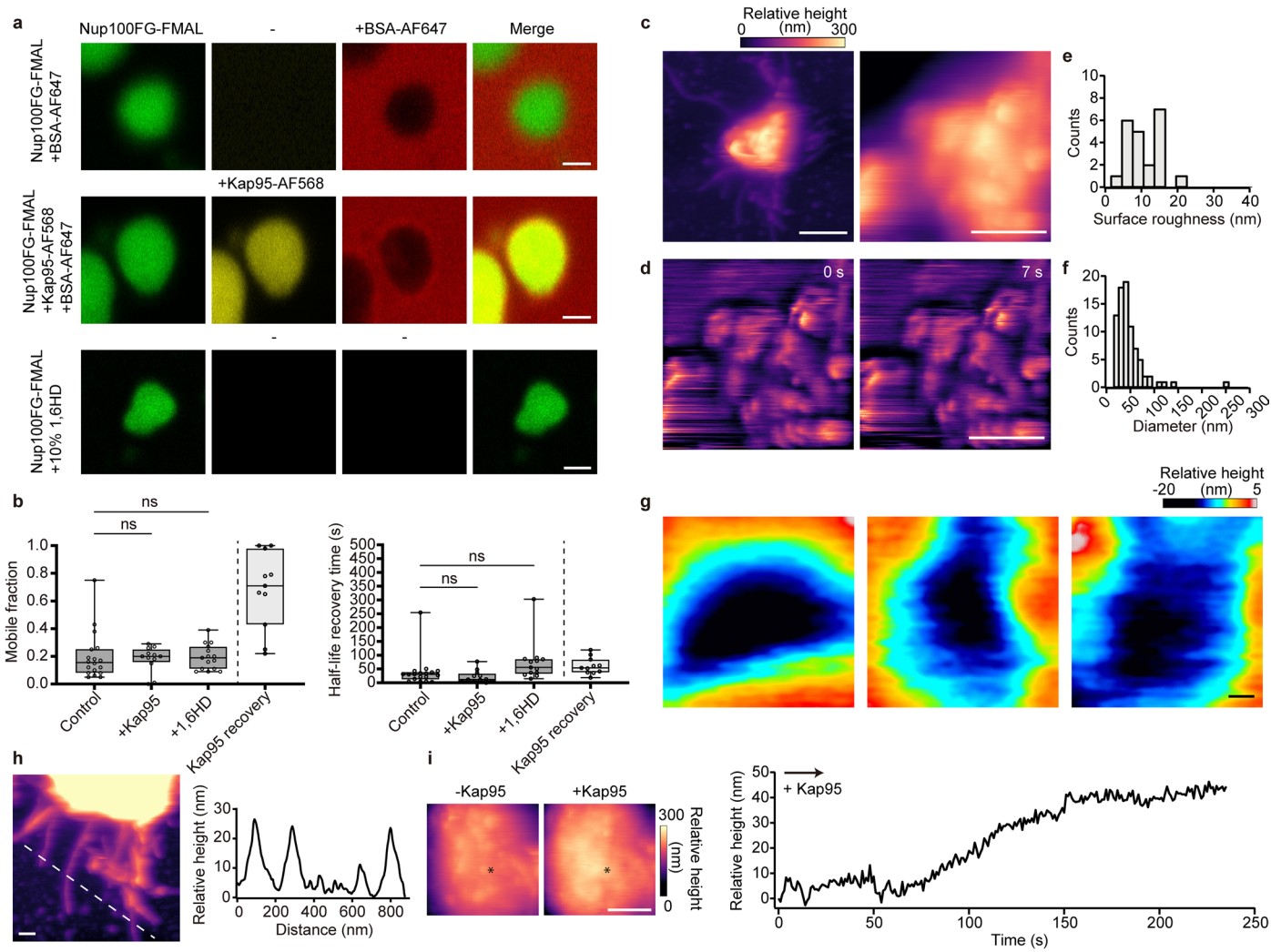

**Extended Data Fig. 8 | In vitro Nup100FG hydrogels. a**, Nup100FG domains (5% FMAL-labelled) form hydrogel particles that are selectively permeable to Kap95 but exclude BSA. Incubating Nup100FG hydrogels in 1,6HD has no discernible effect on their shape nor their exclusionary property against BSA. Scale bars, 2 µm. See Extended Data Fig. 7 for associated 1,6HD data. **b**, FRAP quantification of mobile fraction and recovery times of Nup100FG within Nup100FG hydrogels at the conditions indicated. Number of experimental replicates ≥ 3; No. of hydrogel particles, $n$ = 18 (ctr), 12 (+Kap95), and 13 (+1,6HD). Corresponding values for Kap95 within the Nup100FG hydrogels are also shown (Number of experimental replicates = 4; $n$ = 11 hydrogel particles). Boxplots denote the median, first and third quartiles. Whiskers represent the minimum and maximum values. $P$ values were obtained using an ordinary one-way ANOVA test (ns refers to $P$ > 0.05). $P$ values from left to right: 0.93418 and 0.99377 for mobile fraction, 0.69337 and 0.18885 for recovery times. **c**, Zoomed-out (left; scale bar, 500 nm) and zoomed-in (right; scale bar, 200 nm) HS-AFM images

of a Nup100FG hydrogel with protruding amyloid-like fibrils. **d**, HS-AFM images show that the surface morphology of Nup100FG hydrogels is static, irregular and holey. Same zoomed-out image as in **c** (right) but high-pass filtered. Scale bar, 200 nm. **e**, Surface roughness of Nup100FG hydrogels. No. of experimental replicates = 5; $n$ = 22 hydrogel particles; mean value ± s.d. = 11.8 ± 4.5 nm. **f**, Nup100FG hydrogel hole diameters. No. of experimental replicates = 5; $n$ = 82 holes; mean value ± s.d. = 52.4 ± 31 nm. **g**, Selection of HS-AFM images showing Nup100FG hydrogel holes of irregular size and shape (scale bar, 10 nm) obtained at 0.15 s/frame. **h**, Zoom-in of the amyloid fibrils and thickness as seen in the corresponding cross-sectional profile (white dashed line). Scale bar, 100 nm. **i**, Plot of Nup100FG hydrogel surface height versus time monitored by HS-AFM before (left image) and after incubation in 1 µM Kap95 (right image). Asterisks denote the measurement position. Scale bar, 50 nm. Source numerical data are provided .

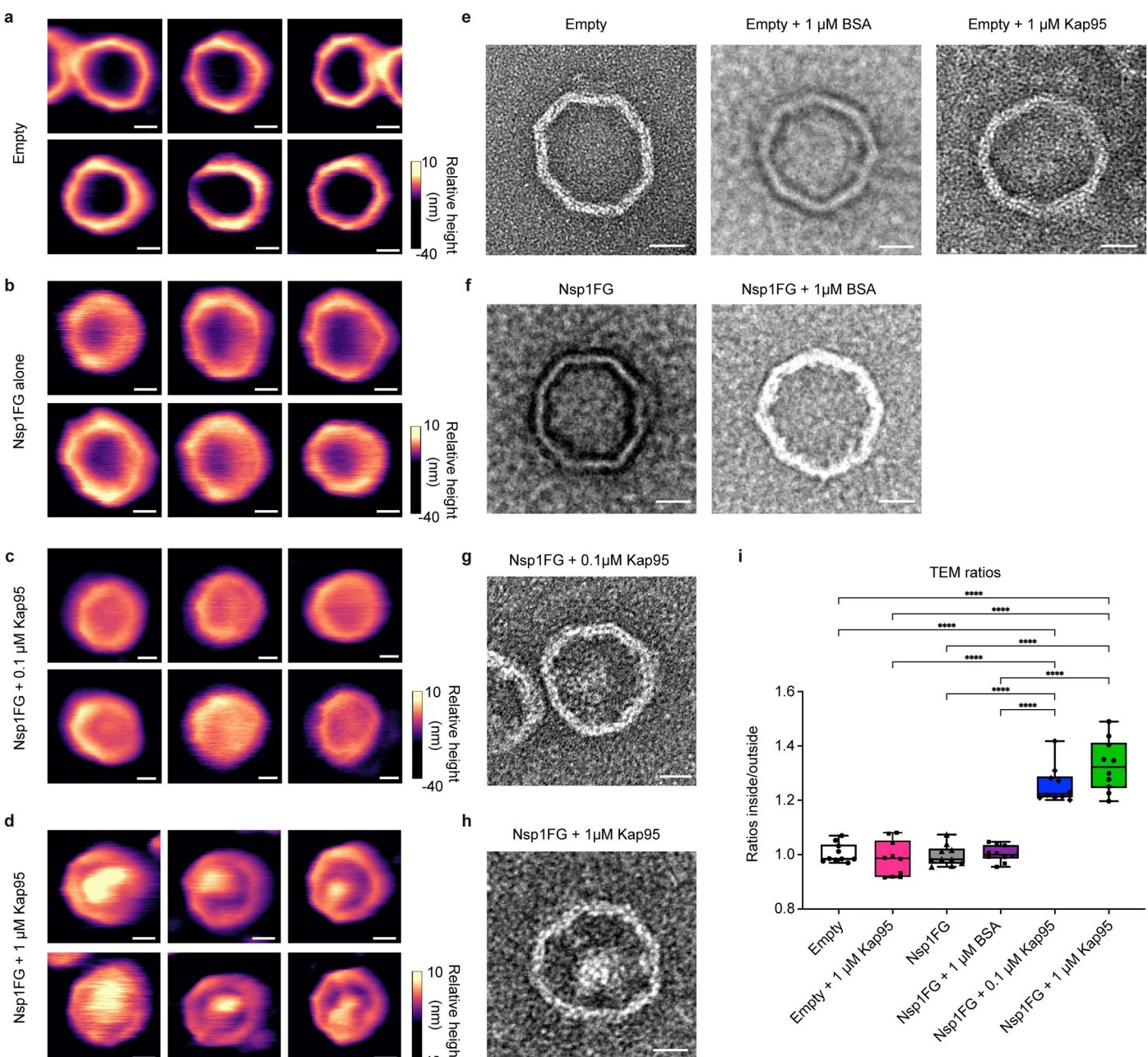

**Extended Data Fig. 9 | Visualization of NuPODs by HS-AFM and negative-stain TEM. a–d**, HS-AFM collages of **a**, empty DNA origami pores, **b**, Nsp1FG-functionalized DNA origami pores (NuPODs), **c**, NuPODs + 0.1 μM Kap95 and **d**, NuPODs + 1 μM Kap95. Scale bars, 20 nm. **e**, TEM images of empty, empty + 1 μM BSA, empty + 1 μM Kap95 DNA origami pores, showing that binding of BSA and Kap95 to empty pores was not observed. Scale bars, 20 nm. **f**, TEM images of NuPODs and NuPODs + 1 μM BSA, showing that binding of BSA to Nsp1FG was not observed. Scale bars, 20 nm. **g,h**, TEM images of **g**,

NuPODs + 0.1 μM Kap95 and **h**, NuPODs + 1 μM Kap95, showing that Kap95 binds to Nsp1FG and forms a CP-like structure in NuPODs. **i**, Inside/outside intensity ratios obtained for the conditions shown (*n* = 10 NuPODs for each condition; all experiments ≥3 independent replicates). Boxplots denote the mean, first and third quartiles. Whiskers represent the minimum and maximum values. Statistical significance was analysed using ordinary one-way ANOVA. Source numerical data are provided.

# Reporting Summary

## Statistics

For all statistical analyses, confirm that the following items are present in the figure legend, table legend, main text, or Methods section.

| n/a | Confirmed | |
|---|---|---|
| ☐ | ☒ | The exact sample size (*n*) for each experimental group/condition, given as a discrete number and unit of measurement |
| ☐ | ☒ | A statement on whether measurements were taken from distinct samples or whether the same sample was measured repeatedly |
| ☐ | ☒ | The statistical test(s) used AND whether they are one- or two-sided<br>*Only common tests should be described solely by name; describe more complex techniques in the Methods section.* |
| ☒ | ☐ | A description of all covariates tested |
| ☒ | ☐ | A description of any assumptions or corrections, such as tests of normality and adjustment for multiple comparisons |
| ☐ | ☒ | A full description of the statistical parameters including central tendency (e.g. means) or other basic estimates (e.g. regression coefficient) AND variation (e.g. standard deviation) or associated estimates of uncertainty (e.g. confidence intervals) |
| ☐ | ☒ | For null hypothesis testing, the test statistic (e.g. *F*, *t*, *r*) with confidence intervals, effect sizes, degrees of freedom and *P* value noted<br>*Give P values as exact values whenever suitable.* |
| ☒ | ☐ | For Bayesian analysis, information on the choice of priors and Markov chain Monte Carlo settings |
| ☒ | ☐ | For hierarchical and complex designs, identification of the appropriate level for tests and full reporting of outcomes |
| ☒ | ☐ | Estimates of effect sizes (e.g. Cohen's *d*, Pearson's *r*), indicating how they were calculated |

*Our web collection on statistics for biologists contains articles on many of the points above.*

## Software and code

Policy information about availability of computer code

| Data collection | Igor Pro (6.3), Chemidoc imaging system (BioRad), softWoRx (Cytiva, Version 7.0.0), PicoScope (7.1.50) |
|---|---|
| Data analysis | Python (2.7, 3.9), ImageJ (1.53c, 1.53t, 1.54p), Relion (3.1.2), easyFRAP software (web version), cryoSPARC (4), SpectroMine (2.8.210609.47784), Igor Pro (8.0), Prism software (GraphPad, 9.1.0, 10.2.3, 10.4.2, 10.6.1) |

For manuscripts utilizing custom algorithms or software that are central to the research but not yet described in published literature, software must be made available to editors and reviewers. We strongly encourage code deposition in a community repository (e.g. GitHub). See the Nature Portfolio guidelines for submitting code & software for further information.

## Data

Policy information about availability of data

All manuscripts must include a data availability statement. This statement should provide the following information, where applicable:
- Accession codes, unique identifiers, or web links for publicly available datasets
- A description of any restrictions on data availability
- For clinical datasets or third party data, please ensure that the statement adheres to our policy

Yeast strains and plasmids generated in this study will be distributed without restriction upon request.
Source data are provided with this study:
Source Data 1 - All numerical values used to generate plots.

Source Data 2 - Unprocessed images of gels and blots.

# Research involving human participants, their data, or biological material

Policy information about studies with human participants or human data. See also policy information about sex, gender (identity/presentation), and sexual orientation and race, ethnicity and racism.

| | |
|---|---|
| Reporting on sex and gender | NA |
| Reporting on race, ethnicity, or other socially relevant groupings | NA |
| Population characteristics | NA |
| Recruitment | NA |
| Ethics oversight | NA |

Note that full information on the approval of the study protocol must also be provided in the manuscript.

# Field-specific reporting

Please select the one below that is the best fit for your research. If you are not sure, read the appropriate sections before making your selection.

☒ Life sciences          ☐ Behavioural & social sciences          ☐ Ecological, evolutionary & environmental sciences

For a reference copy of the document with all sections, see nature.com/documents/nr-reporting-summary-flat.pdf

# Life sciences study design

All studies must disclose on these points even when the disclosure is negative.

| | |
|---|---|
| Sample size | No statistical methods were used to determine sample size. |
| Data exclusions | No data exclusions were made. |
| Replication | HS-AFM and TEM data were analyzed for structurally intact isolated WT or mutant NPCs. All replicates for in vivo assays were successful. |
| Randomization | Samples were not randomized in the experiments. |
| Blinding | Experiments were not blinded. |

# Reporting for specific materials, systems and methods

We require information from authors about some types of materials, experimental systems and methods used in many studies. Here, indicate whether each material, system or method listed is relevant to your study. If you are not sure if a list item applies to your research, read the appropriate section before selecting a response.

## Materials & experimental systems

| n/a | Involved in the study |
|---|---|
| ☐ | ☒ Antibodies |
| ☒ | ☐ Eukaryotic cell lines |
| ☒ | ☐ Palaeontology and archaeology |
| ☒ | ☐ Animals and other organisms |
| ☒ | ☐ Clinical data |
| ☒ | ☐ Dual use research of concern |
| ☒ | ☐ Plants |

## Methods

| n/a | Involved in the study |
|---|---|
| ☒ | ☐ ChIP-seq |
| ☒ | ☐ Flow cytometry |
| ☒ | ☐ MRI-based neuroimaging |

## Antibodies

| Antibodies used | Monoclonal antibody mouse Anti-NSP1 (1:5000 dilution) (Abcam, ab4641), Rabbit IgG affinity purified (Innovative Research, IRBIGGAP500MG), Monoclonal antibody mouse anti-GFP (1:2500 dilution) (Santa Cruz, sc-9996), Monoclonal antibody anti-mouse m-IgGκ BP-HRP (1:2500 dilution) (Santa Cruz, sc-516102), Monoclonal antibody mouse anti-mCherry (1:100 dilution) (ThermoFisher, (4B3)MA5-32977), Monoclonal antibody mouse anti-GFP (1:20 dilution) (Roche, 11814460001), Alexa Fluor 568-conjugated antibody anti-mouse (1:200 dilution) (ThermoFisher, A11004), Polyclonal antibody rabbit anti-eGFP (1:20 dilution) (ThermoFisher, CAB4211) |
|---|---|
| Validation | Antibodies were validated by the manufacturers. For Anti-NSP1, see https://www.abcam.com/products/primary-antibodies/nsp1-antibody-32d6-ab4641.html; for anti-GFP (Santa Cruz), see https://www.scbt.com/p/gfp-antibody-b-2; for anti-mouse m-IgGκ BP-HRP, see https://www.scbt.com/p/m-igg-kappa-bp-hrp; for anti-mCherry, see https://www.thermofisher.com/antibody/product/mCherry-Antibody-clone-4B3-Monoclonal/MA5-32977; for anti-GFP (Roche), see https://www.sigmaaldrich.com/CH/en/product/roche/11814460001?srsltid=AfmBOoo5-4C9eieJScKh75j3GNu1zIjf1U740_fR_PHBVxlkl7r4x09_; for Alexa568 anti-mouse, see https://www.thermofisher.com/antibody/product/Goat-anti-Mouse-IgG-H-L-Cross-Adsorbed-Secondary-Antibody-Polyclonal/A-11004; for anti-eGFP, see https://www.thermofisher.com/antibody/product/eGFP-Antibody-Polyclonal/CAB4211 |

