## [Peer Review File · Nature Cell Biology]

Karyopherins Remodel the Dynamic Organization of the Nuclear Pore Complex Transport Barrier

Corresponding Author: Professor Roderick Lim

Version 0:

Decision Letter:

Revise extended OD

*Please delete the link to your author homepage if you wish to forward this email to co-authors.

Dear Professor Lim,

Your manuscript, "Dynamic organization of the nuclear pore complex central transporter at millisecond transport timescales", has now been seen by 3 referees, who are experts in nuclear pore complex assembly (referee 1); nuclear pore complex function (referee 2); and nuclear transport (referee 3). As you will see from their comments (attached below) they find this work of potential interest, but have raised substantial concerns, which in our view would need to be addressed with considerable revisions before we can consider publication in Nature Cell Biology. I should stress that the referees' concerns about the interpretation that the central mass reflects the high density of Kap95 would need to be addressed with experiments and data, and reconsideration of the study for this journal and re-engagement of referees would depend on strength of these revisions.

Nature Cell Biology editors discuss the referee reports in detail within the editorial team, including the chief editor, to identify key referee points that should be addressed with priority, and requests that are overruled as being beyond the scope of the current study. To guide the scope of the revisions, I have listed these points below. We are committed to providing a fair and constructive peer-review process, so please feel free to contact me if you would like to discuss any of the referee comments further.

In particular, it would be essential to:

A-All reviewers questioned the interpretation that the central mass reflects a high density of Kap95. To support such a conclusion, additional analyses are required as highlighted by the revs:

Rev#1 points #1-2 and "To fully prove that the CP is..." paragraph

Rev#2 "The key advance of this paper is the "rescue" of CP by introducing Kap95 (Fig.1j-o). However, the improved HS-AFM-LS resolution is not key for interpreting this result and would have been resolved with previous ~100 ms HS-AFM measurement capabilities reported in Sakiyama et al. 2016. A more important experiment that would confirm the role of Kap95 in the formation of the CP would be a dose-response evaluation, which would involve titrating Kap95 and quantitating the percent of purified NPCs and DNA-origami pore mimics with a CP. Similarly, it would be interesting to assess if Kap95 recruitment is indeed a determining factor for CP retention by quantitating the kinetics of CP loss in the NPCs from three *S. cerevisiae* strains (Δ FG, Nsp1FGx2, and wildtype) upon incubation of extracted NPCs in buffer. "

Rev#3 requested discussion of alternative interpretations, which would also be important.

B-Address the reviewers' requests for appropriate controls and methodological clarifications (Reviewers#1 and #2).

C-Discuss alternative models that the data may support (Reviewer#3 second paragraph)

D- All other referee concerns pertaining to strengthening existing data, providing controls (Reviewers#1 and #2), methodological details, clarifications and textual changes, should also be addressed.

E- Finally please pay close attention to our guidelines on statistical and methodological reporting (listed below) as failure to do so may delay the reconsideration of the revised manuscript. In particular please provide:

- a Supplementary Table including all numerical source data in Excel format, with data for different figures provided as different sheets within a single Excel file. The file should include source data giving rise to graphical representations and statistical descriptions in the

paper and for all instances where the figures present representative experiments of multiple independent repeats, the source data of all repeats should be provided.

We would be happy to consider a revised manuscript that would satisfactorily address these points, unless a similar paper is published elsewhere, or is accepted for publication in Nature Cell Biology in the meantime.

In contrast, although we agree with referee 2 that testing the functional relevance of the CP structure for nucleocytoplasmic transport would provide valuable insights, we consider this point to be beyond the scope of the present study. Thus, addressing it experimentally will not be necessary for reconsideration of the manuscript at this journal.

- ensure that it conforms to our format instructions and publication policies (see below and www.nature.com/nature/authors/).
- provide a point-by-point rebuttal to the full referee reports verbatim, as provided at the end of this letter.
- provide the completed Editorial Policy Checklist (found here <https://www.nature.com/authors/policies/Policy.pdf>), and Reporting Summary (found here <https://www.nature.com/authors/policies/ReportingSummary.pdf>). This is essential for reconsideration of the manuscript and these documents will be available to editors and referees in the event of peer review. For more information see <http://www.nature.com/authors/policies/availability.html> or contact me.

Nature Cell Biology is committed to improving transparency in authorship. As part of our efforts in this direction, we are now requesting that all authors identified as 'corresponding author' on published papers create and link their Open Researcher and Contributor Identifier (ORCID) with their account on the Manuscript Tracking System (MTS), prior to acceptance. ORCID helps the scientific community achieve unambiguous attribution of all scholarly contributions. You can create and link your ORCID from the home page of the MTS by clicking on 'Modify my Springer Nature account'. For more information please visit <http://www.springernature.com/orcid>.

Link Redacted

We would like to receive a revised submission within six months. We would be happy to consider a revision even after this timeframe, however if the resubmission deadline is missed and the paper is eventually published, the submission date will be the date when the revised manuscript was received.

We hope that you will find our referees' comments, and editorial guidance helpful. Please do not hesitate to contact me if there is anything you would like to discuss.

Best wishes,

Sabrya Carim

Sabrya Carim, PhD
(she/her/hers)
Associate Editor, Nature Cell Biology
Nature Portfolio

Springer Nature
The Campus, 4 Crinan Street, London N1 9XW, UK
sabrya.carim@springernature.com
<https://orcid.org/0000-0001-9485-1938>

Reviewers' Comments:

Reviewer #1:

Remarks to the Author:

In this article, the authors explore the dynamics and the nature of the so-called 'central plug' (CP) of the yeast Nuclear Pore Complex (NPC). By performing High-Speed AFM (HS-AFM) on purified yeast NPCs, they observe a central mass within the NPC channel. This mass is dynamic, as are other fine structures in the channel which are designated by the authors as nucleoporins' FG-domains. The central mass is lost in a significant fraction of NPCs after 72h, but can be restored by addition of Kap95, a nuclear transport receptor. These data suggest that the CP is constituted or induced by Kap95 in the NPC central channel.

In addition, they study the morphology and dynamics of the CP in mutant NPCs, where the FG content has been increased or decreased, and correlate these effects with the permeability barrier efficiency in vivo. Finally, they address the importance of the spatial organization of FG-Nups by comparing two synthetic models: hydrogels spontaneously assembled from recombinant Nup116-FG domains and DNA-origami NPC mimics (NuPODs) that provide an 8-fold symmetry scaffold in which Nsp1-FG domains are grafted. This comparison highlights slower FG dynamics in hydrogels than in NuPODs. Moreover, addition of Kap95 in NuPODs promotes the formation of a

central mass that recalls the CP observed in purified NPCs.

Overall this study addresses the interesting question of the nature and role of the central plug, whose existence and significance has long been debated. Most importantly, it underlines the importance of the spatial arrangement of FG-Nups and how this influences their properties.

Methodologies include state-of-the-art HS-AFM, which provides fast kinetics and uses very small forces, ensuring minimal interference with the sample topography. In addition, implementation of line scanning provides even faster kinetics. Treating the resulting images with high-pass filters enables to detect very fine details and fluctuations which are presumably dynamics of the FG-Nups. The authors have developed an interesting approach to analyze these fluctuations and extract characteristic time constants. This approach is further validated by appropriate modeling. The work is supported by yeast biology, with generation of mutant strains, fluorescence microscopy, proteomics, and biochemistry. Very interestingly, the observations are partially recapitulated in a synthetic DNA-origami system in a bottom-up approach.

The cutting-edge methodologies gathered in this article provide high-quality data, which are presented in well-organized figures.

Statistical analyzes seem appropriate.

However, while the quantitative data are robust and properly analyzed, I am not fully convinced by their interpretation. My major concern regards the nature of the central mass that is observed in a fraction of purified NPC samples. While a series of observations suggest that this mass might be constituted of a high density of Kap95, straightforward proof of this is still missing and other interpretations are possible:

1) Kaps might induce a specific dense organization of FG-domains in the center, but are not necessarily concentrated in this region. In line with this, the HS-AFM simulations, run in the absence of Kaps, exhibit a central mass at the center of the ring that resembles closely the experimental images (supplementary figure 4).

2) The central mass could be the NPC basket, that would only be visible when NPCs are imaged from the nucleoplasmic face, but not from the cytoplasmic face. The authors argue that most purified yeast NPCs are devoid of baskets, but this is contradicted by the proteomic data that show that Mlp2 and Nup2 are present at their expected relative stoichiometry if the basket is present. Moreover Nup2 levels are highly decreased in the -CP particles, in a more dramatic effect than Kap95.

To fully prove that the CP is composed of a high density of Kaps (as illustrated in their model in fig 6), the authors could complement their data with fluorescence microscopy and correlate the presence of a central mass, measured by AFM, with the amount of Kap95 estimated by immuno-fluorescence or direct fluorescence using Kap95-mNeonGreen strains used in figure 2m. To further prove the local concentration of Kap95 within the CP region, super-resolution microscopy would be necessary.

In addition to this concern, I also have minor questions or remarks that should nevertheless be addressed:

- Line 122-123: isolated NPCs are in their contracted form, could you discuss how your current findings could be extrapolated to in situ dilated NPCs? Could you address this point using the DNA-origami model?

- How are the NPC and CP diameter measured and the height profiles rendered? I could not find this information in the methods.

- In figure 1b & d, it looks like the NPC is elliptical rather than circular and that the long axis rotates over time. Could you quantify this and compare +CP and -CP particles, or at least comment on this?

- In fig1 o-q, everything looks more dynamic than in the original +CP and -CP particles. This should be discussed.

- The comparison between FG-hydrogels and NuPODs would be stronger if the same FG domains were utilized in both cases.

- Many experiments rely on modulating Kaps levels, but Kap concentration is not measured or discussed. This is however a crucial point and should be taken into account in the interpretation of the results: more precisely in figure 1, what is the concentration of Kaps in purified NPC? how does it compare with the concentration added for rescue? in figure 2m, if I understand correctly Kap95-mNeonGreen is overexpressed while endogenous Kap95 is present, What is the ratio between endogenous and exogenous Kap95 levels? in figure 4, what are the local concentrations of Nup116-FG and Kap95? how do these relate to concentrations in purified NPCs? Same question for figure 5.

- The conclusions drawn in figure 2m (lines 283-286) are only valid if all images are acquired with the same acquisition settings (laser power, exposure time...) and rendered with the same contrast parameters. Adjusting parameters on very bright cytoplasmic foci may artificially dim NE levels.

- In supplementary fig 11, the quantification does not reflect the visual impression on the microscopy fields of view (comparison between -PEG and +PEG).

- Finally, I find some conclusions overstated: this is the case lines 260-261 or lines 286-287.

Besides this, the manuscript is very clearly written, the experimental details are in general well explained, with a complete description in the main text and/or in the methods section. Previous work is correctly cited.

Overall, the observations described here are very interesting, they rely on high-wire approaches and are highly relevant to the general understanding of the physical properties underlining the nucleo-cytoplasmic transport. Once the correlation between Kaps density of and the CP is directly assessed, I foresee a very strong and convincing article.

Reviewer #2:

Remarks to the Author:

Kozai et al. used high-speed atomic force microscopy (HS-AFM) and a complement of in vitro and in vivo experiments, to probe the dynamic organization of the permeability barrier, composed of natively unfolded protein regions rich in phenylalanine-glycine (FG) repeats, in the central transport channel of the nuclear pore complex (NPC) at millisecond timescales. The general goal of the study was to understand how structure and dynamics of the permeability barrier relate to nucleocytoplasmic transport. The corresponding author (Roderick Lim) had previously used HS-AFM to image *Xenopus laevis* NPCs from isolated nuclear envelopes (Sakiyama et al. 2016, Nature Nanotechnology). In that work, they observed the dynamic, fluctuating nature of FG regions and noted a "central plug" (CP) structure in ~40% of the NPCs' central transport channels, which they excluded further analyses. In the current study, Kozai et al. focused on characterizing this CP mass using *Saccharomyces cerevisiae* NPCs that were detergent-extracted according to protocols previously established by the corresponding author (Mike Rout) and are ~100% "plugged". The authors applied a "line scanning" data processing methodology (HS-AFM-LS) that assumes 8-fold rotational symmetry of the NPC. This method improved the temporal resolution of their measurements by a factor of ~50-100-fold, achieving single-digit millisecond timescales. Interestingly, a prolonged 72-hour incubation in buffer led to the loss of the CP in ~60% of NPCs. Using this method, they observed differences in the dynamicity of the permeability barrier between NPCs with and without the CP. CP loss correlated with the depletion of karyopherins and other factors, as quantitated by mass spectrometry analysis, similar to previous work from the senior author Brian Chait's group (Hakhverdyan et al. 2021, Mol. Cell). Conversely, adding back purified FG-interacting karyopherin Kap95 restored CPs and the dynamics of the permeability barrier associated with the presence of CP. The authors also applied HS-AFM-LS to a DNA origami hollow cylinder design with tethered FG

regions – a minimalist NPC mimic, as previously published by the senior author Chenxiang Lin's group (Stanley et al, 2019, ACS Nano). This approach replicated NPC-like dynamics and CP formation, albeit requiring a larger DNA origami cylinder's diameter compared to previous versions, which raises questions about the robustness of this system as an NPC mimic. The HS-AFM-LS analysis was extended to NPCs purified from a "ΔFG" *S. cerevisiae* strain in which ~50% of the FG mass is deleted. This strain had been used in previous work by the senior author Liesbeth Veenhoff's group (Strawn et al. 2007, NCB; Popken et al, 2015, MBoC). They also analyzed a novel "Nsp1FGx2" strain with ~30% extra FG mass due to duplicated Nsp1 FG regions. The mutations appeared to alter the dynamic fluctuations of FG regions as well as CP structures, though it is not clear what else might be different in these strains. Moreover, whereas both strains show fitness and transport function defects compared to wildtype, ΔFG and Nsp1FGx2 show decreased and increased Kap95 recruitment to the nuclear envelope in vivo, respectively. However, the relationship between the altered phenotypes and the altered dynamics of the permeability barrier observed by HS-AFM-LS is correlative and not mechanistically obvious. The authors also overexpressed various GFP-fused FG regions in vivo. They assert that the translocation of these condensate-forming FG regions into the nucleus as soon as 1 hour after expression is induced supports a dynamic permeability barrier. However, without a negative control, other interpretations are possible, such as karyopherins mediating the import of FG polypeptides that are not tethered to the NPC via a structural domain. Indeed, a long-standing question about the physical state of FG repeats in the field is whether they form a compact sieve-like mesh that can be "melted" by permeating karyopherins, or whether FG regions act more like a molecular brush, preventing free diffusion based on entropic excluded volume effects. The authors analyzed in vitro formed FG hydrogel particles by HS-AFM-LS, noting that they are heterogeneous, rigid, and holey, but that the holes are not induced by Kap95. Within the holes, they observed rare fluctuations that could be attributed to brush-like FG region movement like in the purified NPCs. The authors draw a distinction between the "stiff" hydrogel reconstituted in vitro and the dynamic barrier they observe in isolated NPCs. However, FG hydrogel particles are huge compared to the NPC's central transport channel and it is not clear what a native hydrogel on the NPC scale would behave like under an AFM. This lack of an appropriate positive control weakens the claim that the observed permeability dynamics are different from those expected for a hydrogel.

In summary, the study attempts to address a long-standing question about the nature of the CP, initially observed by Unwin and Milligan (1982, JCB). It is not clear what the benefit of the higher temporal millisecond-scale resolution of the HS-AFM-LS methodology is, especially considering that karyopherin-mediated transport events are orders of magnitude faster, typically ~5-10 ms. The key advance of this paper is the "rescue" of CP by introducing Kap95 (Fig.1j-o). However, the improved HS-AFM-LS resolution is not key for interpreting this result and would have been resolved with previous ~100 ms HS-AFM measurement capabilities reported in Sakiyama et al. 2016. A more important experiment that would confirm the role of Kap95 in the formation of the CP would be a dose-response evaluation, which would involve titrating Kap95 and quantitating the percent of purified NPCs and DNA-origami pore mimics with a CP. Similarly, it would be interesting to assess if Kap95 recruitment is indeed a determining factor for CP retention by quantitating the kinetics of CP loss in the NPCs from three *S. cerevisiae* strains (ΔFG, Nsp1FGx2, and wildtype) upon incubation of extracted NPCs in buffer. The rest of the results are mostly descriptive or correlative, without testing clear hypotheses, and thus hard to interpret in the broader context of NPC function. Many experiments are derivative from previous work by the authors.

Taken together while the technical advances in AFM are impressive, the study only contributes an incremental advance towards answering the question about NPC function and nucleocytoplasmic transport in the context of cell biology. The key question remains: is the CP structure per se functionally relevant for nucleocytoplasmic transport or is it an emergent phenomenon of spatially constrained FG regions and shuttling karyopherins? This paper is therefore more appropriate for a specialized journal.

Reviewer #3:

Remarks to the Author:

The nuclear pore complex (NPC) controls macromolecular movement between the cytoplasm and the nucleoplasm of eukaryotic cells. While the NPC scaffold has been well-defined by structural work over the last few decades, the permeability barrier that occupies and controls permeation through the pore is less well understood. This permeability barrier is generated by a dynamic assembly of intrinsically disordered proteins whose properties are modulated by the presence of various transport factors, most notably Kap95 (Importin beta). The primary emphasis of this paper is the use of high-speed AFM (HS-AFM) to probe the properties of the permeability barrier of isolated yeast NPCs. The central findings are that the central plug (CP) is a dynamic entity that undergoes lateral motion within the pore, that this CP is lost over the course of days, and that the CP feature can be reconstituted by the addition of Kap95. Satisfyingly, this picture is confirmed by a simple model system constructed from DNA origami. These are important new conclusions well-supported by the data. Using kymograph analysis of repeated single AFM linescans, they were able to monitor fluctuations of the permeability barrier surface with ~2 ms resolution (instead of ~150 ms for a full frame), which yielded autocorrelation times of ~20-40 ms. This indicates fluctuations on the order of translocation times previously measured at the single molecule level, and therefore consistent with transport through a liquid-like/flexible milieu. The authors also probed NPCs both with increased (+30%) and decreased (-51%) numbers of FG-repeats. In both cases, these are reported to have overall reduced dynamic signatures, though it is not clear how this conclusion is made since the autocorrelation times vary across the pore diameter and the profiles under the various conditions are not overlaid for direct comparison. A simple average autocorrelation time might not suffice, unless this is weighted by annular area. The Kap95 reconstituted NPCs have the most dynamic barrier with an essentially flat autocorrelation time profile across the pore, which is not discussed. Both FG-repeat mutants exhibit reduced nucleocytoplasmic partitioning, suggesting that the wildtype FG-concentration is optimized. Interestingly, in vitro hydrogels made from purified FG-repeat proteins are holey structures, visualized and interpreted as a solidified scaffold containing surface assemblies of FG-polypeptides that enable permeation via a similar mechanism to authentic NPCs. This is an important finding that adds to our understanding of in vitro hydrogels formed from FG-Nups. Though somewhat peripheral to the CP story, this nonetheless provides an additional new application of the HS-AFM technology. Finally, the authors examined the distribution of GFP-tagged FG-Nups in vivo, finding that these can penetrate into the nucleus but not accumulate at NPCs. The implications of these findings or their relevance to the CP story are unclear. These proteins could penetrate NPCs as monomers, multimers, or in combination with transport receptors. These data, therefore, seem to be best left for a subsequent paper as part of a more complete story. Overall, however, this is a well-written and interesting manuscript that provides important new information about the NPC permeability barrier.

In their model (Fig. 6), they assume that Kap95 accumulates in the center of the central transporter, which is a reasonable conclusion considering the reformation of the central plug. However, since they have not directly identified where Kap95 is located within the pore, I would give at least a moderate weight to the possibility of an alternate model. For example, one possibility is that Kap95 accumulates at the periphery of the pore (as promulgated in the original Kap-centric model of the Lim group), and this causes a remodeling and increased cohesiveness of the FG-Nups in the center of the pore, thus giving rise to the feature identified as a CP. This is not to argue that there are no Kap95 molecules in the center of the pore, but the density may not be highest here, as the authors assume.

Minor comments:

Title – the use of 'transport' in the title is awkward'; 'millisecond timescale' is sufficient. I believe the authors wanted to indicate that the transport occurs on the millisecond timescale, but the usage here is odd. 'Transport timescale' would also suffice.

Abstract – 'elucidate' is vague; perhaps 'probe the dynamics of'. The sentence following 'consequently' does not follow from the previous statement – this word can be deleted.

Line 70 – The Musser lab, which has extensively examined transport rates, was not referenced.

Lines 80-86 – The difference between BMCs and hydrogels is not clear. Solidified structures are often observed for condensates and are often considered a natural process of condensate aging. Thus, perhaps FG-Nup hydrogels are simply fast-maturing condensates. Following the view of Mittag and Pappu (2022, Mol Cell), such a fast phase transition could be considered 'phase separation coupled to percolation'.

Line 92 – The model system (yeast) should be indicated here, especially as a contrast with the previous work on *Xenopus* oocytes.

Lines 113-120 – This description of HS-AFM is very useful for the non-expert. However, it is not clear whether impulse (force*time) and pN forces refer to a single tapping event or the aggregate of tapping events at a single pixel. The effect of multiple rapid taps would have a cumulative effect. As per lines 120-121 of the supplementary, 80 pixels at 1.875 ms/line would be ~23 μ s/pixel. At 0.5 MHz, this would be ~10 taps/pixel, and due to the tip diameter, taps would be felt even if centered over a neighboring pixel.

Line 124 – 'transport' is not needed here. It is sufficient, clearer, and more accurate to simply state that the isolated NPCs are competent for binding Kaps.

Lines 342 – 'Nature' seems too strong a word. Something more specific and less general would be better.

Lines 358-359 – they have not shown that Kap95 binds to the ends of FG domains.

Fig. 1 – Are the scale bars in (b) and (d) accurate? If so, the pore in (b) is about twice the size of the pore in (d).

Supplementary:

Lines 119-121 – Clarity on dwell time between lines is needed, as this affects the time resolution. Also, clarity on whether the scanning is back and forth (Fig. 6b), or only one direction (Fig. 4c) is needed – this also affect time resolution.

Fig. S4 – (a) The diameter of the model NPC is over 2-fold less than that of authentic yeast NPCs (acknowledged as 'miniaturized version'). Considering that FG-Nup density affects permeability barrier properties, how does this influence the simulation results? (d-e) There appears to be a blue torus overlaid onto a grid pattern, suggesting that the simulations do not capture the effect of the NPC scaffold. Clarification is needed.

Fig. S5e – I do not understand the meaning of the 95% confidence interval and why this is used to eliminate and raise the baseline. It seems like this would substantially affect the estimated tau values.

Fig. S10b – It is not clear that the identified event is actually a fusion event, since there is no intermediate showing a partial outline of both puncta. This appears as an 'apparent fusion', as technically one of the puncta could disassemble and absorb the monomers from the other.

Line 910 – Is "CT" supposed to be "CNT"?

AUTHOR AFFILIATIONS – should be denoted with numerical superscripts (not symbols) preceding the names. Full addresses should be

included, with US states in full and providing zip/post codes. The corresponding author is denoted by: "Correspondence should be addressed to [initials]."

Methods should be written concisely, but should contain all elements necessary to allow interpretation and replication of the results. As a guideline, Methods sections typically do not exceed 3,000 words. The Methods should be divided into subsections listing reagents and techniques. When citing previous methods, accurate references should be provided and any alterations should be noted. Information must be provided about: antibody dilutions, company names, catalogue numbers and clone numbers for monoclonal antibodies; sequences of RNAi and cDNA probes/primers or company names and catalogue numbers if reagents are commercial; cell line names, sources and information on cell line identity and authentication. Animal studies and experiments involving human subjects must be reported in detail, identifying the committees approving the protocols. For studies involving human subjects/samples, a statement must be included confirming that informed consent was obtained. Statistical analyses and information on the reproducibility of experimental results should be provided in a section titled "Statistics and Reproducibility".

All Nature Cell Biology manuscripts submitted on or after March 21 2016 must include a Data availability statement at the end of the Methods section. For Springer Nature policies on data availability see <http://www.nature.com/authors/policies/availability.html>; for more information on this particular policy see <http://www.nature.com/authors/policies/data/data-availability-statements-data-citations.pdf>. The Data availability statement should include:

- Accession codes for primary datasets (generated during the study under consideration and designated as "primary accessions") and secondary datasets (published datasets reanalysed during the study under consideration, designated as "referenced accessions"). For primary accessions data should be made public to coincide with publication of the manuscript. A list of data types for which submission to community-endorsed public repositories is mandated (including sequence, structure, microarray, deep sequencing data) can be found here <http://www.nature.com/authors/policies/availability.html#data>.
- Unique identifiers (accession codes, DOIs or other unique persistent identifier) and hyperlinks for datasets deposited in an approved repository, but for which data deposition is not mandated (see here for details <http://www.nature.com/sdata/data-policies/repositories>).
- At a minimum, please include a statement confirming that all relevant data are available from the authors, and/or are included with the manuscript (e.g. as source data or supplementary information), listing which data are included (e.g. by figure panels and data types) and mentioning any restrictions on availability.
- If a dataset has a Digital Object Identifier (DOI) as its unique identifier, we strongly encourage including this in the Reference list and citing the dataset in the Methods.

We recommend that you upload the step-by-step protocols used in this manuscript to [protocols.io](https://www.protocols.io). More details can be found at <https://www.protocols.io/help/publish-articles>.

All imaging data should be accompanied by scale bars, which should be defined in the legend. Cropped images of gels/blots are acceptable, but need to be accompanied by size markers, and to retain visible background signal within the linear range (i.e. should not be saturated). The boundaries of panels with low background have to be demarked with black lines.

Splicing of panels should only be considered if unavoidable, and must be clearly marked on the figure, and noted in the legend with a statement on whether the samples were obtained and processed simultaneously. Quantitative comparisons between samples on different gels/blots are discouraged; if this is unavoidable, it should only be performed for samples derived from the same experiment with gels/blots were processed in parallel, which needs to be stated in the legend.

The total number of Supplementary Figures (not including the "unprocessed scans" Supplementary Figure) should not exceed the number of main display items (figures and/or tables (see our Guide to Authors and March 2012 editorial <http://www.nature.com/ncb/authors/submit/index.html#suppinfo>; <http://www.nature.com/ncb/journal/v14/n3/index.html#ed>). No restrictions apply to Supplementary Tables or Videos, but we advise authors to be selective in including supplemental data.

GUIDELINES FOR EXPERIMENTAL AND STATISTICAL REPORTING

REPORTING REQUIREMENTS – To improve the quality of methods and statistics reporting in our papers we have recently revised the reporting checklist we introduced in 2013. We are now asking all life sciences authors to complete two items: an Editorial Policy Checklist (found here <https://www.nature.com/authors/policies/Policy.pdf>) that verifies compliance with all required editorial policies and a reporting summary (found here <https://www.nature.com/authors/policies/ReportingSummary.pdf>) that collects information on experimental design and reagents. These documents are available to referees to aid the evaluation of the manuscript. Please note that these forms are dynamic 'smart pdfs' and must therefore be downloaded and completed in Adobe Reader. We will then flatten them for ease of use by the reviewers. If you would like to reference the guidance text as you complete the template, please access these flattened versions at <http://www.nature.com/authors/policies/availability.html>.

Version 1:

Decision Letter:

Dear Professor Lim,

Your revised manuscript "Dynamic Organization of the Nuclear Pore Complex Transport Barrier", has now been seen by 3 referees, who are experts in nuclear pore complex assembly (referee 1); nuclear pore complex function (referee 2); and nuclear transport (referee 3), and whose comments are pasted below. In light of their advice, we regret that we cannot offer to publish the study in Nature Cell Biology.

As you will see, although the reviewers find the work improved in revision and although Reviewer 1 sounds more positive in tone than Reviewer 2 and Reviewer 3, Reviewers 2 and 3 continue to have persisting concerns with regard to the Kap95 accumulation in the central mass and the model.

Although we cannot offer to publish your manuscript, I suggest that you consider Nature Communications or the EMBO Journal as a suitable venue for this work. To transfer your manuscript, please use our manuscript transfer portal. You will not have to re-supply manuscript metadata and files, unless you wish to make modifications. For more information, please see our http://www.nature.com/authors/author_resources/transfer_manuscripts.html

[WT.mc_id=EMI_NPG_1511_AUTHORTRANSF&WT.ec_id=AUTHOR](http://www.nature.com/authors/author_resources/transfer_manuscripts.html?WT.mc_id=EMI_NPG_1511_AUTHORTRANSF&WT.ec_id=AUTHOR) manuscript transfer FAQ page.

If you would be interested in the option to transfer the manuscript to Nature Communications or Communications Biology, please let me know if you would like me to initiate a consultation with my colleagues there to explore whether they would commit to take the manuscript forward with the existing peer review history.

We are very sorry that we could not be more positive on this occasion, but we thank you for the opportunity to consider this work.

With kind regards,
Sabrya Carim

Sabrya Carim, PhD
(she/her/hers)
Senior Editor, Nature Cell Biology
Nature Portfolio

Springer Nature
The Campus, 4 Crinan Street, London N1 9XW, UK
sabrya.carim@springernature.com
<https://orcid.org/0000-0001-9485-1938>

Reviewers' comments:

Reviewer #1 (Remarks to the Author):

The authors have significantly improved their manuscript, adding essential experiments that strengthen their conclusions. They also clarified some useful points. All my wariness was extensively addressed. The overall manuscript is strong and very interesting for a large scientific community. I support its publication at Nature Cell Biology.

Reviewer #2 (Remarks to the Author):

Kozai, Fernandez-Martinez, et al., are to be commended for adding further experimental evidence during the revision of this manuscript. Specifically, the confocal fluorescence microscopy data shown in Supplemental Figure 4 illustrate the gradual leeching of endogenous Kap95 from the central plug (CP) and the subsequent deposition of exogenous Kap95 into empty "-CP" NPCs to reform the CP. When considered alongside the experiments in Figure 3A, which demonstrate Kap95 accumulation at NPCs containing twice the number of FG repeats within Nsp1 (Nsp1FGx2), this new experiment offers a compelling quantitative analysis of the dynamic occupancy of Kap95 at the CP. Accordingly, it merits elevation into the main text Figure 3. However, while valuable, this experiment does not adequately address whether Kap95 concentration plays a role in CP formation. The authors compare the original NuPOD data collected in the presence of 1 μ M Kap95, against a new limiting concentration of 100nM Kap95. In this condition, HS-AFM indicates no CP formation, while TEM reveals a diffuse CP structure of similar lateral dimensions. In my view, the minimalistic NuPOD system represents the weakest model to elucidate concentration dependence, incorporating only the FG domain of Nsp1. This experiment would be more effectively conducted with isolated *S. cerevisiae* NPCs, incubated in buffer for 72hrs to ensure a homogenous -CP sample, followed by addition of exogenous Kap95 across a range of concentrations from 0 to 1 μ M. Unfortunately, a single ambiguous data point at a concentration that does not produce a consistent outcome between AFM and TEM analyses fails to clarify whether CP formation scales linearly with increasing Kap95 concentration or occurs abruptly at a threshold level.

I wish to emphasize that the technical advancements in AFM analysis presented here are impressive and certainly merit publication in a specialized journal. Nonetheless, the primary functional insights from this manuscript are that the CP (1) moves laterally within the NPC's central transport channel, and (2) is dynamically comprised at least in part with transport factors from the beta-karyopherin family. Since the CP has been recognized in EM imaging of the NPC since Unwin and Milligan (JCB, 1982) and has been demonstrated via AFM by Stoffer et al. (JMB, 2003) to leech from the central channel over time, the novel contribution of this manuscript is the demonstration that Kap95 alone can form a CP analog in vitro. Unfortunately, these findings do not align with the manuscript's title, "Dynamic Organization of the Nuclear Pore Complex Transport Barrier," as the paper provides little additional insight into the nature of the diffusion barrier, focusing instead on a region that other studies suggest is largely inert during karyopherin mediated transport.

This brings me to a significant caveat of this study, single-molecule digitonin-permeabilized NPC studies from the Musser group (Chowdhury et al., NCB, 2022; and very recently Sau et al., Nature, 2025) are not sufficiently discussed by the authors. These studies reveal that Kapbeta1 (the human homolog of Kap95), Kapbeta2, and Kapalpha-Kapbeta1 complexes utilize overlapping transport pathways through the NPC diffusion barrier. Importantly, they show that the central channel comprises three distinct annular zones with variable transport efficiency, with the CP localized in a central null zone where few Kapbeta1 transport events occur. A revised manuscript should therefore critically evaluate how these peripheral transport routes might impact the proposed functional role of the CP. For instance, if the CP does not facilitate active transport, might it function instead as a reservoir of beta-karyopherins ready for cargo binding?

Taken together, this revision does not adequately address the functional relevance of the CP on NPC function, particularly in the context of karyopherin-mediated nucleocytoplasmic transport. Thus, I am sorry to conclude that my initial impression remains unchanged, this manuscript is better suited for publication in a specialized journal.

Minor Comment:

Line 287: The statement, "(3) their lack of enrichment at the nuclear rim, unlike Kap95 (Fig. 3a), indicates that FG domains in the CNT do not form the same condensates as observed elsewhere in vivo," is unclear. Specifically, the phrase "as observed elsewhere in vivo" is ambiguous, does it refer to the GFP-FG domain fusions described in the current study, or condensates observed during either oocyte NPC assembly, or within aberrant NPC assemblies induced by Torsin-A/Brl1/Brr6 mutations? Given that comparisons between artificial GFP-FG domain fusions and naturally occurring FG-Nup condensates are inherently problematic, I recommend omitting or revising this statement for clarity.

Reviewer #3 (Remarks to the Author):

This manuscript is much improved and reads much better. In particular, the text is much more balanced and provides a complete and comprehensive story of how the current work fits in with prior data. It also portrays the complexity and challenges of the nuclear pore permeability barrier, and yet, nonetheless, demonstrates how they have made a significant advance. As described below, there are a few relatively minor issues that need attention, but these are readily addressable.

Supplementary Fig. 4 is new and uses confocal fluorescence microscopy to demonstrate the release and re-binding of Kap95 to isolated NPCs. Unfortunately, the data as presented is not convincing (the entire figure is poor resolution, which doesn't help). A key problem is that there appears to be significant non-specific binding raising questions about the quantification. There is no control slide in the absence of NPCs. In (a), it is entirely unclear what we should consider as fluorescence arising from a pore. In (d) at 72 h, the merge image suggests a significant pixel drift or mis-aligned images. It is not clear why mCherry- or GFP-tagged NPCs or Kap95-GFP cannot be imaged directly. The ratio and overall signals from pores and the bound Kaps fluctuate widely, as evident from the images and their violin plots. Most certainly, the binding of Kaps to NPCs occurs on the tens of seconds to minutes timescale, so the 20 h time constant from (f) is extremely misleading. This raises concerns about the similar 20 h time constant in (c). Due to their time resolution, they are essentially

fitting two points to the curve. In contrast, Kap95 binding to Nup100 hydrogels (Sup Fig. 12i) occurs on the minute timescale. They have not described the criteria for scoring a CP as present or not in their AFM experiments. Well-done fluorescence experiments could provide some nuance here, namely is the plug lost all at once (loss of a large object such as a large cargo) or is it lost gradually (stepwise loss of individual proteins). Nonetheless, I don't find Sup Fig. 4 as necessary for the manuscript.

In Fig. 6, they provide a complex model with substantial information that is not directly addressed by their data. Much of this information comes from a large body of work in the field. But the location/distribution of Kaps and those Kaps that are slow and fast moving is not known, and this is pure speculation. Likewise, the bidirectional transport through 8 peripheral channels (Sup Fig. 14) is speculation. Why do they assume transport through channels (Sup Fig. 14) rather than through an annulus (Fig. 6)? More description is needed about the motivation for these speculative points.

Minor comments:

In at least a few locations at first mention, the authors should indicate that the eight-fold symmetry is a 'rotational symmetry' (e.g., line 134 and in the Fig. 1 caption).

Lines 254-255 – This sentence is confusing. How does increased kinetic accessibility lead to decreased avidity?

Lines 268-271 – experimental references are missing for the statement at the end of the sentence. Ref. 77 is published (PNAS).

Fig. 1 – the axes in (f) appear to be labeled as -80°C and 4°C. What does an off-axis point imply – an intermediate temperature? This panel is small and the gray dots are almost invisible. While the color scheme/key in (k) (and in similar figure panels in the manuscript) is clear on my screen, it is not clear in my printout. Probably worse for color-blind individuals.

Fig. 3d – the four sample images above the figure to indicate the scoring are not mutually exclusive. What if a condensate is observed both inside and outside a nucleus?

Fig. 6b – the phrase '>>1 μm' is extreme. Condensate/hydrogel size is determined by the amount of material added and structures on the 100 nm scale are certainly feasible.

Fig. 5d,e and Sup Fig. 8 – the time between when Kap95 was added and when the images were obtained should be indicated.

**For Nature Portfolio general information and news for authors, see <http://npg.nature.com/authors>.

Version 2:

Decision Letter:

Our ref: NCB-A54964B-Z

18th August 2025

Dear Dr. Lim,

Thank you for submitting your revised manuscript "Karyopherins Remodel the Dynamic Organization of the Nuclear Pore Complex Transport Barrier" (NCB-A54964B-Z). It has now been seen by the original referees and their comments are below. The reviewers find that the paper has improved in revision, and therefore we'll be happy in principle to publish it in Nature Cell Biology, pending minor revisions to satisfy the referees' final requests and to comply with our editorial and formatting guidelines.

Please note that our articles must have 6 to 8 main figures and they can have up to 10 ED figures. If there are more figures than that, they become supplementary figures. Also supp figures, and supp notes and other supplementary materials, are included in the supplementary information PDF on the website. We know this is less accessed than our main and ED figures so we try to limit the use of supplementary figures as much as we can. Please ensure that all figures fit into a single standard page and adhere to a maximum page size of roughly 180mm wide x 200mm high, but also please use the full page space to fill the figure. At present several figures are too tiny to be legible once re-sized during the production process. To ensure legibility once figures are re-sized, please use a font size of no smaller than 6pt Arial or Helvetica throughout the figures.

We are now performing detailed checks on your paper and will send you a checklist detailing our editorial and formatting requirements in about 1-2 weeks. **Please do not upload the final materials and make any revisions until you receive this additional information from us.**

Thank you again for your interest in Nature Cell Biology Please do not hesitate to contact me if you have any questions.

Best wishes,

Sabrya Carim, PhD
(she/her/hers)
Senior Editor, Nature Cell Biology
Nature Portfolio

Springer Nature
The Campus, 4 Crinan Street, London N1 9XW, UK
sabrya.carim@springernature.com
<https://orcid.org/0000-0001-9485-1938>

Reviewer #2 (Remarks to the Author):

Kozai & Fernandez-Martinez et al., are to be congratulated on a revision that addresses all my technical and intellectual queries. As in previous submissions, the AFM data presented represents a high-water mark for the technique and a valuable addition to the literature. However, where the current revision makes significant strides is in reframing and focusing the description of the experimental advances presented.

Briefly, the authors have: (1) amended the title to better reflect the experimental focus of the study. (2) Introduced a summary paragraph at the end of the introduction that details the significance of their findings, making clear the novel aspects of the manuscript. (3) Provided proper credit to the recent single-molecule MINFLUX studies from the Musser group throughout the manuscript, with an informative and well-judged integration of their data in the discussion. (4) Expanded Figure 1G: The requested experiment increasing the range of Kap95 concentrations tested in the dose-response restoration of +CP-NPCs in purified *S. cerevisiae* NPC samples, is a missing-link addition that alleviates my lingering experimental reservations. (5) Revised Figure S6: Modifications made in response to Reviewer 3, adding controls and amending the labeling and visualization of the fluorescence microscopy analysis of CP leaching and exogenous Kap95 rescue. This represents a valuable data point that, in its current form, is more broadly understandable and merits inclusion in the final manuscript.

Overall, the authors have addressed all my concerns, and in redrafting sections of the manuscript, particularly the title, have made significant efforts to reframe the data presented in a way that better reflects the experiments presented. Provided the scope of the manuscript remains within NCB's editorial range, I recommend it should be published without further revision.

Version 3:

Decision Letter:

Dear Dr Lim,

I am pleased to inform you that your manuscript, "Karyopherins Remodel the Dynamic Organization of the Nuclear Pore Complex Transport Barrier", has now been accepted for publication in Nature Cell Biology. Congratulations!

You may wish to make your media relations office aware of your accepted publication, in case they consider it appropriate to organize some internal or external publicity. Once your paper has been scheduled you will receive an email confirming the publication details. This is normally 3-4 working days in advance of publication. If you need additional notice of the date and time of publication, please let the production team know when you receive the proof of your article to ensure there is sufficient time to coordinate. Further information on our

embargo policies can be found here: <https://www.nature.com/authors/policies/embargo.html>

Please note that *Nature Cell Biology* is a Transformative Journal (TJ). Authors may publish their research with us through the traditional subscription access route or make their paper immediately open access through payment of an article-processing charge (APC). Authors will not be required to make a final decision about access to their article until it has been accepted. [Find out more about Transformative Journals](https://www.springernature.com/gp/open-research/transformative-journals)

Authors may need to take specific actions to achieve compliance with funder and institutional open access mandates. If your research is supported by a funder that requires immediate open access (e.g. according to [Plan S principles](https://www.springernature.com/gp/open-science/plan-s-compliance) or the [NIH public access policy](https://www.springernature.com/gp/open-science/us-federal-agency-compliance)) then you should select the gold OA route, and we will direct you to the compliant route where possible. Because authors warrant under our subscription licensing terms that they haven't committed to licensing any version of their article under a licence inconsistent with the terms of our agreement – including the applicable embargo period – publication under the subscription model isn't suitable for authors whose funders require no embargo.

If you have not already done so, we strongly recommend that you upload the step-by-step protocols used in this manuscript to protocols.io (<https://protocols.io>), an open online resource that allows researchers to share their detailed experimental know-how. All uploaded protocols are made freely available and are assigned DOIs for ease of citation. Protocols and Nature Portfolio journal papers in which they are used can be linked to one another, and this link is clearly and prominently visible in the online versions of both. Authors who performed the specific experiments can act as primary authors for the Protocol as they will be best placed to share the methodology details, but the Corresponding Author of the present research paper should be included as one of the authors. By uploading your Protocols onto protocols.io, you are enabling researchers to more readily reproduce or adapt the methodology you use, as well as increasing the visibility of your protocols and papers. You can also establish a dedicated workspace to collect your lab Protocols. Further information can be found at <https://www.protocols.io/help/publish-articles>.

Nature Cell Biology encourages authors presenting evidence for cell, biological, molecular, and genetic interactions to consider communicating these findings using Biofactoid (<https://biofactoid.org/>). This tool helps users share a searchable representation of interactions (e.g. binding, gene expression, post-translational modification) between genes, gene products, or chemicals. Information added to Biofactoid, with author attribution, is shared on social media and public databases, such as Pathway Commons, where it can be discovered and analyzed in the context of a large and growing corpus of knowledge.

With kind regards,

Sabrya

Sabrya Carim, PhD
(she/her/hers)
Senior Editor, Nature Cell Biology
Nature Portfolio

Springer Nature
The Campus, 4 Crinan Street, London N1 9XW, UK
sabrya.carim@springernature.com
<https://orcid.org/0000-0001-9485-1938>

** Visit the Springer Nature Editorial and Publishing website at http://editorial-jobs.springernature.com?utm_source=ejp_NCB_email&utm_medium=ejp_NCB_email&utm_campaign=ejp_NCB for more information about our career opportunities. If you have any questions please click [here](mailto:editorial.publishing.jobs@springernature.com).

Reviewer #1:

Remarks to the Author:

In this article, the authors explore the dynamics and the nature of the so-called 'central plug' (CP) of the yeast Nuclear Pore Complex (NPC). By performing High-Speed AFM (HS-AFM) on purified yeast NPCs, they observe a central mass within the NPC channel. This mass is dynamic, as are other fine structures in the channel which are designated by the authors as nucleoporins' FG-domains. The central mass is lost in a significant fraction of NPCs after 72h, but can be restored by addition of Kap95, a nuclear transport receptor. These data suggest that the CP is constituted or induced by Kap95 in the NPC central channel.

In addition, they study the morphology and dynamics of the CP in mutant NPCs, where the FG content has been increased or decreased, and correlate these effects with the permeability barrier efficiency *in vivo*. Finally, they address the importance of the spatial organization of FG-Nups by comparing two synthetic models: hydrogels spontaneously assembled from recombinant Nup116-FG domains and DNA-origami NPC mimics (NuPODs) that provide an 8-fold symmetry scaffold in which Nsp1-FG domains are grafted. This comparison highlights slower FG dynamics in hydrogels than in NuPODs. Moreover, addition of Kap95 in NuPODs promotes the formation of a central mass that recalls the CP observed in purified NPCs.

Overall this study addresses the interesting question of the nature and role of the central plug, whose existence and significance has long been debated. Most importantly, it underlines the importance of the spatial arrangement of FG-Nups and how this influences their properties. Methodologies include state-of-the-art HS-AFM, which provides fast kinetics and uses very small forces, ensuring minimal interference with the sample topography. In addition, implementation of line scanning provides even faster kinetics. Treating the resulting images with high-pass filters enables to detect very fine details and fluctuations which are presumably dynamics of the FG-Nups. The authors have developed an interesting approach to analyze these fluctuations and extract characteristic time constants. This approach is further validated by appropriate modeling. The work is supported by yeast biology, with generation of mutant strains, fluorescence microscopy, proteomics, and biochemistry. Very interestingly, the observations are partially recapitulated in a synthetic DNA-origami system in a bottom-up approach.

The cutting-edge methodologies gathered in this article provide high-quality data, which are presented in well-organized figures. Statistical analyzes seem appropriate.

However, while the quantitative data are robust and properly analyzed, I am not fully convinced by their interpretation. My major concern regards the nature of the central mass that is observed in a fraction of purified NPC samples. While a series of observations suggest that this mass might be constituted of a high density of Kap95, straightforward proof of this is still missing and other interpretations are possible:

- 1) Kaps might induce a specific dense organization of FG-domains in the center, but are not necessarily concentrated in this region. In line with this, the HS-AFM simulations, run in the

absence of Kaps, exhibit a central mass at the center of the ring that resembles closely the experimental images (supplementary figure 4).

We thank the reviewer for the opportunity to clarify the simulation results and their interpretation, and we provide an analysis of simulations of the full NPC with and without Kaps.

First, we wish to clarify that the sole aim of the simulation shown in old Supp. Fig. 4 was to test for HS-AFM data asynchronicity (see Kato J. Chem. Theory Comput. 2023). We modeled Nsp1 FG domain behavior in a simplified 22 nm-diameter toroidal nanopore, which was not intended as a realistic representation of the full NPC so as to reduce computation time. The resulting FG domain concentration in the nanopore was 33% higher than the full NPC (see Table below for details), which likely explains why it resembles “a central mass at the center of the ring” when Kaps are absent. Hence, these simulations represent a simplified setting that is incompatible with the full NPC.

We have now revised the text to make this point clearer:

>> L178: “To specifically explore this effect, we modeled Nsp1 FG domains (Nsp1FG) in a simplified 22 nm-diameter toroidal nanopore, being distinct from the NPC, and sampled it computationally in the manner mimicking a HS-AFM (**Supplementary Fig. 5; Supplementary Video 3**). The simulated images displayed continuous features along horizontal lines, reflecting the capture of the collective motion of Nsp1FG along the HS-AFM fast scan axis (X-axis). In contrast, image data along vertical lines appeared more discontinuous due to asynchronicity along the HS-AFM slow scan axis (Y-axis). This demonstrates that HS-AFM more accurately captures dynamic molecular movements within individual horizontal lines than across them.”

Nonetheless, in a separate study by Raveh et al., we have developed a detailed integrative model of the transport through the full NPC, cross-validated by multiple sources of data (<https://www.biorxiv.org/content/10.1101/2023.12.31.573409v2>). Motivated by the referee's comment, we have used the new model to compare the simulated HS-AFM signal in -Kap95 vs. +Kap95 NPC, shown below (superimposed on the Kim et al., 2018 NPC structure). The resulting height profile supports the formation of a CP-like mass due to Kap95 enrichment in NPCs. The Kap95+ state corresponds to a few dozen nuclear transport receptors (NTRs) within the central channel, in accordance with the total NTR count using mass-spectrometry in the current manuscript (**Supp. Fig. 9b,c**), and the confidence intervals reflect uncertainty following sensitivity analysis of a single free parameter in the HS-AFM simulator. In the simulations of the NPC, the presence of Kap95 consistently lead to an elevated surface in the simulated HS-AFM. Moreover, the simulations clearly show that this central region is abundant with Kap95.

However, as this additional study extends beyond the scope of the Kozai manuscript and to avoid any further delays in its processing, we believe it is more appropriate to include this data in our upcoming work by Raveh et al..

The key differences between the 22 nm-diameter toroidal nanopore used in Kozai et al. and the full NPC simulations by Raveh et al. are listed in the Table below:

	Kozai et al.	Raveh et al.
Simulation	Coarse-grained Brownian dynamics simulation	Coarse-grained Brownian dynamics simulation, see Raveh et al. (bioRxiv)
Pore geometry	Simple toroidal ring	Yeast NPC scaffold (Kim et al, Nature , 2018)
Pore diameter	22 nm	44 nm
Pore thickness	15 nm	30 nm
FG domain anchor points	4 layers, each consisting of 8 FG domains	Yeast NPC scaffold (Kim et al, Nature , 2018)
Number and type of FG domain	Nsp1 (32)	216 in total: Nsp1 (48), Nup100 (16), Nup116 (16), Nup159 (16), Nup49 (32), Nup57 (32), Nup145 (16), Nup1 (8), Nup2 (16), Nup60 (16)
Representation of FG domains	Spherical beads	Spherical beads
Bead diameter	1.6 nm	1.6 nm
Number of residues per bead	20	20
Number of beads per FG domain	Nsp1 (16)	Nsp1 (36), Nup100 (48), Nup116 (53), Nup159 (30), Nup49 (17), Nup57 (15), Nup145 (15), Nup1 (36), Nup2 (23), Nup60 (8)
FG domain (aa) concentration	0.21 g/cm ³	0.16 g/cm ³
Kaps	No	Yes

2) The central mass could be the NPC basket, that would only be visible when NPCs are imaged from the nucleoplasmic face, but not from the cytoplasmic face. The authors argue that most purified yeast NPCs are devoid of baskets, but this is contradicted by the proteomic data that show that Mlp2 and Nup2 are present at their expected relative stoichiometry if the basket is present. Moreover Nup2 levels are highly decreased in the -CP particles, in a more dramatic effect than Kap95.

There is a clear density in the center of mammalian NPCs that is distinct from the nuclear basket (Singh et al. Cell 2024, Fig. 2B; Fig – upper panel). Similar observations have also been reported in *Dictyostelium discoideum* NPCs (Beck et al. Science 2004; Nature 2007). We also know that nuclear basket components in yeast are generally dynamic (Hakhverdyan et al. Mol Cell 2021; Akey et al. Cell 2022; Singh et al. Cell 2024). However, we can go one step further. NPCs that lack Mlp1 and Mlp2 also lack a nuclear basket (Niepel et al., 2013). Thus, we imaged NPCs from a yeast strain lacking Mlp1 and Mlp2. Indeed, we see that many of the NPCs still bear a central plug-like density in a Nup84 protein A tagged Mlp1 and Mlp2 deletion mutant (Fig – lower panel), just as they do in WT NPCs. If the plug was the collapsed basket, we would expect to see no plugs at all. This leads us to conclude that the central plug is a structurally distinct feature that is not an effect of the presence or absence of the basket.

We have now revised the text to make this point clearer and included the figure as **Supp. Fig. 1p,q**:

>> L131: “CPs have been identified in both isolated and *in situ* NPCs^{4,5}, and their presence in NPCs lacking the core nuclear basket proteins Mlp1 and Mlp2⁶⁸ confirms that the CP and the nuclear basket are distinct⁶⁹ (**Supplementary Fig. 1**).”

>> Supp L9: “For the construction of the Δ Mlp1 Δ Mlp2 mutant, the genomic loci of W303 of Mlp1 and Mlp2 were replaced with the URA3 and LEU2 markers, respectively.

>> Supp L36: “For the Δ Mlp1 Δ Mlp2 and WT comparison NPCs, Nup84 was genomically tagged with Protein A and NPCs were purified as described above.”

We also apologize, as the writing might have been confusing about the basket. We have not claimed that the isolated NPCs are devoid of baskets, quite the opposite, we use the main basket component, Mlp1, as the handle to affinity isolate the NPCs (see section “Affinity-purification of endogenous *S. cerevisiae* NPCs” in the Materials and Methods, and the referred publications Kim et al. 2018, Akey et al. 2022 and 2023). Thus all our isolated NPCs should contain a basket. What we tried to indicate in the text is that, based on our previous extensive structural analyses of isolated NPCs (Kim et al. 2018, Akey et al. 2022 and 2023), and *in cellulo* cryo-ET analyses (Akey et al. 2022, Singh et al. 2024), we know that the basket

is not a structurally well-defined, stable feature in a majority of NPCs, possibly due to its flexible and heterogeneous nature. We have tried to clarify this in the text:

>>L136: “It was however, not possible to determine their nucleocytoplasmic orientation, as the nuclear basket is not a structurally well-defined, stable feature in a majority of NPCs^{4, 5, 7, 69}.”

Nup2 has been suggested to be a somewhat peripheral component of the nuclear basket (Cibulka et al. 2022), present in an average of only 8 copies per NPC in double outer ring NPCs and localized primary on the NPC periphery, near the nuclear outer ring (Singh et al. 2024). Additionally, Nup2 exhibits a very fast exchange rate in isolated NPCs (Hakhverdyan et al. 2021), which altogether makes it unlikely to be the main component of the CP. Nevertheless, we are not discarding the possibility that it is also involved in the formation of the native CP, as is surely the case of other nuclear transport factors and constituents, and we have included that caveat in the text (see also response to point #7):

>>L211: “Although Kap95 alone could restore a CP-like structure, the overall dynamics of this reconstituted structure were notably faster than those of +CP WT NPCs, suggesting that additional nuclear transport factors and other constituents are present in the native CP mass.”

3) To fully prove that the CP is composed of a high density of Kaps (as illustrated in their model in fig 6), the authors could complement their data with fluorescence microscopy and correlate the presence of a central mass, measured by AFM, with the amount of Kap95 estimated by immuno-fluorescence or direct fluorescence using Kap95-mNeonGreen strains used in figure 2m. To further prove the local concentration of Kap95 within the CP region, super-resolution microscopy would be necessary.

We thank the reviewer for suggesting these additional experiments. We have now conducted confocal microscopy experiments to ascertain the time-dependent loss/gain of Kap95 at isolated WT NPCs as detailed below:

NPCs were isolated from an Mlp1ppxPrA pNup84-mCherry strain expressing Kap95-GFP under its native promoter to examine the time-dependent loss of endogenous Kap95 after 0, 24, 48 and 72 hours of incubation in PBS buffer at 4 deg C. At each time point, NPC samples were deposited on glass slides, followed by BSA passivation and immunostaining using an anti-eGFP antibody labeled with Alexa Fluor-647 (rabbit, Sigma-Aldrich) and an anti-mCherry antibody labeled with Alexa Fluor-568 (mouse). Fluorescence images were captured using a point scanning confocal microscope with the same settings for all experiments (**Supp. Fig. 4a**; see also Methods).

Fluorescence image quantification shows a time-dependent decrease in the anti-eGFP-647 signal, indicating Kap95-GFP dissociation from NPCs (**Supp. Fig. 4b**). Notably, the observed signals were still several times stronger than the baseline anti-eGFP-647 signal when NPCs were absent, confirming specificity. The dissociation of the strongly-bound fraction of Kap95 from isolated NPCs follows the equation:

$$[\text{endoKap95}]_t = [\text{endoKap95}]_0 \cdot e^{-\frac{\ln 2}{t_{1/2}} t}, \quad (1)$$

where $[\text{endoKap95}]_t$ represents the remaining endoKap95 in NPCs after an incubation time t , $[\text{endoKap95}]_0$ is the initial amount of strongly bound endoKap95 at $t = 0$ and $t_{1/2}$ is the half-life of the bound complexes. Assuming that the anti-eGFP-647 signal linearly corresponds to $[\text{endoKap95}]_t$ at $t = 0, 24, 48$ and 72 hours, the fitted half-life is $t_{1/2} = 20.1$ hours, indicating that 50% of the strongly-bound fraction of endoKap95 remains in the isolated NPCs after 20 hours (Supp Fig. 4c).

It is worth noting that in vitro binding assays on FG domain layers (e.g., Nup62, Nup98 and Nup153) yielded similar half-lives of approx. 10 – 20 h for the strongly-bound fraction of Kap β 1, based on their kinetic dissociation constants (k_{off}) (see Fig. 7 Kapinos Biophys. J. 2014; Fig. S4 Kapinos JCB 2017).

Following the decrease of endoKap95, we tested how 100 nM exogenous Kap95 labeled with maleimide FLUX680 (exoKap95-FLUX860) would re-occupy NPCs, and if their binding depended on whether the NPCs had been incubated in PBS for t = 0, 24, 48 or 72 hours (**Supp Fig. 4d**). (Note: This concentration was chosen over 1 μ M Kap95 to minimize the non-specific background signal.) Using a Nup82-GFP strain, we observed an increase in the exoKap95-FLUX860 signal that approached saturation at approximately 72 hours (**Supp Fig. 4e**). The association of exoKap95 then follows the equation (**Supp Fig. 4f**):

$$[\text{exoKap95}]_t = [\text{exoKap95}]_0 + ([\text{exoKap95}]_\infty - [\text{exoKap95}]_0) \cdot (1 - e^{-\frac{\ln 2}{t_{1/2}} t}) \quad (2)$$

where $[\text{exoKap95}]_t$ represents the amount of exoKap95 that bind to NPCs at time t, $[\text{exoKap95}]_0$ is the initial amount bound to NPCs at t = 0, $[\text{exoKap95}]_\infty$ is an extrapolated value that estimates the saturated amount of bound exoKap95, and $t_{1/2}$ is fixed to 20 hours.

This analysis confirms that the loss/gain of the central plug observed by HS-AFM directly correlates with the kinetics of endoKap95 dissociation from NPCs, independently verifying that its half-life is consistent with in vitro measurements. These findings not only demonstrate that the recruitment of Kap95 is indeed a key determining factor for CP retention, but also show that the amount of exoKap95 gained is consistent with the amount of endoKap95 lost.

>> This figure has been included as **Supp Fig. 4**.

>> The above text has been adapted as a new section in the Materials and Methods entitled "***Time-dependent loss/gain of Kap95 at isolated WT NPCs***". (Supp. L412 - 464)

>>L160: "This behavior was confirmed by confocal microscopy measurements, which demonstrated a direct correlation between CP loss and the slow dissociation rate of this long-lived pool of Kap95 from the NPCs^{15, 48} (**Supplementary Fig. 4**)."

>>L203: "Confocal microscopy later confirmed that the amount of exogenous Kap95 gained by -CP WT NPCs matched the amount of endogenous Kap95 lost by +CP WT NPCs, underscoring the role of Kaps as integral constituents of the CP (**Supplementary Fig. 4**)."

In addition to this concern, I also have minor questions or remarks that should nevertheless be addressed:

4) - Line 122-123: isolated NPCs are in their contracted form, could you discuss how your current findings could be extrapolated to in situ dilated NPCs ? Could you address this point using the DNA-origami model ?

The theoretical FG domain concentration in dilated NPCs is ~20% less than constricted (isolated) NPCs (see **Supp. Table 2**), suggesting a weakening of the permeability barrier. We

hypothesize that Kap-related CP formation would help to reinforce the permeability barrier by counteracting the limited extensibility of the FG Nups in dilated NPCs. Indeed, our recently published cryo-EM data on dilated NPCs confirm that the CP remains intact and aligned along the central transport axis (Singh et al. Cell 2024, Fig. 2B; see also Q2 above).

Yes, we have examined this effect in a separate DNA origami model-based work (Feng Sci. Adv. 2024; <https://doi.org/10.1126/sciadv.adq8773>). Briefly, adding importin β 1 to Nup62 NuPODs with an inner diameter of \sim 45 nm induced the formation of a central CP-like cluster, whereas the cluster shifted from the periphery to the center in a dose-dependent manner in \sim 65 nm diameter NuPODs.

>> L424: “Indeed, dilated NPCs may recruit additional Kaps to maintain the CP along the central transport axis⁶⁹, counteracting the weakened transport barrier caused by a 20% reduction in FG domain concentration compared to constricted NPCs^{5, 8, 9, 105} (**Supplementary Table 2**), as observed in NuPODs¹⁰⁶.”

5) - How are the NPC and CP diameter measured and the height profiles rendered? I could not find this information in the methods.

We apologize for this oversight. The below text has now been added to the Methods.

>> Supp L241: ***Measurements of NPC and central plug diameters***

“NPC and CP diameters were measured from “zoomed-out” images of individual NPCs (e.g., Supplementary Fig. 1) using Fiji. Cross-sectional line profiles were extracted across opposing pairs of octants, yielding four “scaffold” profiles per NPC. NPC diameters were determined by measuring peak-to-peak distances across the highest points of the scaffold. CP diameters were measured from the same line profiles. The resulting NPC and CP diameter values were plotted as histograms and fitted with a single Gaussian function to obtain the mean and standard deviation.”

>> Supp L249: ***Average height profiles***

“The average cross-sectional profile of the NPC (Fig. 1g; black line) was obtained by averaging over all cross-sectional line profiles extracted from four opposing pairs of octants per NPC, across all NPCs. To generate the average height profile of the central transporter, height values were extracted from kymographs at each distance from the pore center and averaged along the time axis. This process was repeated across multiple NPCs. The resulting average height profile was then overlaid onto the scaffold’s average profile.”

6) - In figure 1b & d, it looks like the NPC is elliptic rather than circular and that the long axis rotates over time. Could you quantify this and compare +CP and -CP particles, or at least comment on this ?

Cryo-EM-based NPC structural models appear nearly perfectly symmetric and round because each tomogram is first segmented and reconstructed into a single spoke, and then expanded and fitted into an averaged density to generate an idealized 8-fold symmetric NPC.

In contrast, individual NPCs are inherently flexible structures (Akey JMB 1995) that can dilate, constrict (Zimmerli Science 2021; Schuller Nature 2021; Akey Cell 2022) and even split apart (Kreysing Cell 2025). As a result, they often deviate from a perfectly symmetric ring structure (Beck Nature 2007; Akey JMB 1995; Sabinina MBOC 2021), as noted by Beck et al (2007): *“Most NPCs assume a slightly elliptical shape that entails a small change of the pore diameter as compared to a perfectly symmetric ring structure (Fig. 1c, d).”*

Deviations to the shape of the central channel, such as those originating from dynamic movements of individual spokes (Sakiyama Sem. Cell Dev. Biol. 2017), are more pronounced in HS-AFM, as the NPCs are neither fixed nor frozen. We agree that these behaviors certainly warrant further research but are beyond the scope of the current submission.

7) - In fig1 o-q, everything looks more dynamic than in the original +CP and -CP particles. This should be discussed.

We focused our CP-reconstitution efforts on Kap95 after mass spectrometry analysis confirmed that most of it was displaced from -CP WT NPCs (new Fig. 1f). However, we cannot rule out the involvement of additional Kaps or other constituents in CP-formation, and indeed, there is strong evidence that this is the case (this ms and Kim et al., 2018). The ms has since been revised to make this point clear, differentiating between the CP and the Kap95-restored “CP-like structure”:

>>L211: *“Although Kap95 alone could restore a CP-like structure, the overall dynamics of this reconstituted structure were notably faster than those of +CP WT NPCs, suggesting that additional nuclear transport factors and other constituents are present in the native CP mass.”*

8) - The comparison between FG-hydrogels and NuPODs would be stronger if the same FG domains were utilized in both cases.

Our intention was to address similarities/differences between the FG-hydrogel and actual NPCs, as it has been widely proposed in the literature. We selected Nup100FG and Nup116FG because they are the yeast homologs of vertebrate Nup98, the FG repeat most commonly used to generate model hydrogels (Schmidt and Görlich, 2015). As we show, the FG hydrogel state neither reproduces the scaffold structure nor the discrete arrangement of the FG domains in the NPC nor the nanometer dimensions of the NPC, nor the observed dynamic behaviors of the FG domains and the CP. We then used the Nsp1 NuPODS as a model system (following similar Nsp1-nanopore studies before, e.g. Jovanovic-Talisman et al., 2009) to test for CP reconstitution (and not to compare against FG hydrogels).

9) - Many experiments rely on modulating Kaps levels, but Kap concentration is not measured or discussed. This is however a crucial point and should be taken into account in the interpretation of the results: more precisely in figure 1, (A) what is the concentration of Kaps in purified NPC ? (B) how does it compare with the concentration added for rescue ? (C) In figure 2m, if I understand correctly Kap95-mNeonGreen is overexpressed while endogenous Kap95 is present. (D) What is the ratio between endogenous and exogenous Kap95 levels ? (E) in figure 4, what are the local concentrations of Nup116-FG and Kap95? how do these relate to concentrations in purified NPCs? (F) Same question for figure 5.

We apologize for not making these points clearer.

(A) After affinity-purification, ~20 endogenous Kap95 molecules were found to reside in each +CP WT NPC (see mass spectrometric analysis in new Supp. Fig. 9b,c; Source Data Table; and Extended Data Fig. 3 in Kim Nature 2018). This is consistent with the reported number of Kap β 1 molecules per NPC in HeLa cells (Fig. 4D in Lowe eLife 2015), and corresponds to an effective concentration of ~200 μ M Kap95 within a +CP WT NPC. Such a concentration is markedly higher than the ~1 μ M cellular concentration of Kap95 in *S. cerevisiae* (Timney J. Cell Biol. 2006) reflecting its enrichment at NPCs, as evidenced by a nuclear rim “staining” in vivo (old Fig. 2m this work; Kalita JCS 2021; Kalita JCB 2022; many others) and ex vivo (Lowe eLife 2015; Kapinos JCB 2017).

>>L157: “Label-free quantitative mass spectrometry revealed that, upon isolation, +CP WT NPCs contain approximately 20 molecules of Kap95, the primary nuclear import factor, corresponding to an effective concentration of ~200 μ M⁷.”

(B) We incubated the -CP WT NPCs in 1 μ M exogenous Kap95, which is equivalent to its estimated cellular concentration (Timney J. Cell Biol. 2006), to reconstitute the central plug.

>>L201: “Strikingly, adding 1 μ M exogenous Kap95 to -CP WT NPCs restored a CP-like structure ...”

>>Supp L144: “For reconstitution experiments of the central plug-like feature, -CP WT NPCs were monitored *in situ* and in real time by HS-AFM while Kap95 was added to the imaging buffer, achieving a final concentration of 1 μ M Kap95.”

(C, D) With regards to old Fig. 2m (new Fig. 3a), we need to clarify that the Kap95-mNeonGreen construct was expressed under the control of its native promoter and locus and no other version of Kap95 is present in the strain (see also Supp. Table 3). Moreover, it should be noted that the yeast strain carrying the Kap95-mNeonGreen construct was not the same as the strains used to isolate the NPCs (WT, Δ FG and Nsp1FGx2).

(E) Response added to Supp.

>>Supp L312: “The NupFG concentration within hydrogel particles was estimated by fluorescence calibration of FMAL²⁷, yielding 9.9 ± 6.1 mM for Nup100FG and 16 ± 7 mM for Nup116FG...The final concentration of Kap95-568 in Nup116FG particles was 1.58 ± 0.2 μ M...”

The stark contrast between the concentration of Kap95 in the vastly larger hydrogels and the ~200 μ M Kap95 concentration in the nanoscopic NPCs highlights the significant amount of excluded volume within the hydrogel. This is consistent with the 10% fraction of mobile Nup116FG in the Nup116FG hydrogel when Kap95 is present (**Fig. 4b**).

(F) Response added to main text:

>>L333: “Nevertheless, adding Kap95 to NuPODs induced a mobile, CP-like structure with dose-dependent changes in height and dynamics (0.1 μ M versus 1 μ M), qualitatively resembling the behavior when 1 μ M Kap95 was added to -CP WT NPCs (**Fig. 5d-g; Fig. 1j**). This

indicates that the NPC transport barrier mechanism is critically dependent on the CNT's dimensions and geometry (**Fig. 5d-g; Supplementary Fig. 13**)."

As NuPODs are not our primary focus, we did not quantify the number of Kap95 molecules inside each NuPOD. Nevertheless, previous estimates by Dekker, Onck et al. indicate that incubating 1.9 μM Kap95 resulted in approximately 20 Kap95 molecules binding simultaneously within 55 nm-diameter Nsp1FG-coated NPC mimics (Fragasso Nanoscale Research 2022). We now write:

>>L338: "While our assay did not directly quantify the number of Kap95 molecules residing within the NuPODs, ~ 20 Kap95 molecules were found within Nsp1FG-coated NPC mimics of comparable size⁸², aligning with our findings in +CP WT NPCs⁷ (**Supplementary Fig. 9b,c**)."

10) - The conclusions drawn in figure 2m (lines 283-286) are only valid if all images are acquired with the same acquisition settings (laser power, exposure time...) and rendered with the same contrast parameters. Adjusting parameters on very bright cytoplasmic foci may artificially dim NE levels.

While old Figures 2m and Figure 3 were generated using different microscopes, such that we do not make direct comparisons between them, it remains true that NE signal was not observed on any these cells expressing GFP-Nup100FG (new **Fig. 3d/Supp Fig. 10a**). Here, the controls (new **Supp Fig. 10d**) were generated with the same microscope, settings and image processing parameters.

See **Supp Fig. 10a**, L949: "The bottom images have 8-fold (Nup100FG, hNup153FG, and Nup116FG) and 4-fold (Nup159SFG, Nsp1FG, and Nup60FG) lower intensity compared to the upper panels. Upper panels serve to assess if a punctate NPC signal becomes visible under the same imaging conditions used in (d) for imaging full length Nups."

>>L280: "However, Nup100FG also lacked the punctate nuclear rim staining typically seen with NPC-associated proteins such as GFP-tagged full-length FG Nups including Nup100GFP (**Supplementary Fig. 10d**). ~~or FG-interacting proteins like Kap95 (**Fig. 3a**).~~"

11) - In supplementary fig 11, the quantification does not reflect the visual impression on the microscopy fields of view (comparison between -PEG and +PEG).

In Supp Fig. 11, the hydrogel particles were analyzed by their "particle counts" (L972/L975), not their size or brightness. The y-axes in Supp. Fig. 11b,d are also labelled as "Number of particles".

12) - Finally, I find some conclusions overstated: this is the case lines 260-261 or lines 286-287.

>>L263: "These results show that additional FG domain density impairs nucleocytoplasmic transport by an *apparent* overtightening of the NPC permeability barrier."

>>L286: Deleted.

13) Besides this, the manuscript is very clearly written, the experimental details are in general well explained, with a complete description in the main text and/or in the methods section. Previous work is correctly cited. *Overall, the observations described here are very interesting, they rely on high-wire approaches and are highly relevant to the general understanding of the physical properties underlining the nucleo-cytoplasmic transport.* Once the correlation between Kaps density of and the CP is directly assessed, I foresee a very strong and convincing article.

We thank the referee for pointing out a series of details that we needed to make clearer and strengthen in our manuscript to support our claims and for acknowledging the relevance of our results. It was especially satisfying to find that the time-dependent gain of exogenous Kap95 was consistent with the time-dependent loss of endogenous Kap95, confirming that Kap95 recruitment is indeed a key determinant of CP retention.

Reviewer #2:

Remarks to the Author:

Kozai et al. used high-speed atomic force microscopy (HS-AFM) and a complement of in vitro and in vivo experiments, to probe the dynamic organization of the permeability barrier, composed of natively unfolded protein regions rich in phenylalanine-glycine (FG) repeats, in the central transport channel of the nuclear pore complex (NPC) at millisecond timescales. The general goal of the study was to understand how structure and dynamics of the permeability barrier relate to nucleocytoplasmic transport. The corresponding author (Roderick Lim) had previously used HS-AFM to image *Xenopus laevis* NPCs from isolated nuclear envelopes (Sakiyama et al. 2016, Nature Nanotechnology). In that work, they observed the dynamic, fluctuating nature of FG regions and noted a “central plug” (CP) structure in ~40% of the NPCs’ central transport channels, which they excluded further analyses. In the current study, Kozai et al. focused on characterizing this CP mass using *Saccharomyces cerevisiae* NPCs that were detergent-extracted according to protocols previously established by the corresponding author (Mike Rout) and are ~100% “plugged”. The authors applied a “line scanning” data processing methodology (HS-AFM-LS) that assumes 8-fold rotational symmetry of the NPC. This method improved the temporal resolution of their measurements by a factor of ~50-100-fold, achieving single-digit millisecond timescales. Interestingly, a prolonged 72-hour incubation in buffer led to the loss of the CP in ~60% of NPCs.

Using this method, they observed differences in the dynamicity of the permeability barrier between NPCs with and without the CP. CP loss correlated with the depletion of karyopherins and other factors, as quantitated by mass spectrometry analysis, similar to previous work from the senior author Brian Chait’s group (Hakhverdyan et al. 2021, Mol. Cell). Conversely, adding back purified FG-interacting karyopherin Kap95 restored CPs and the dynamics of the permeability barrier associated with the presence of CP. The authors also applied HS-AFM-LS to a DNA origami hollow cylinder design with tethered FG regions – a minimalist NPC mimic, as previously published by the senior author Chenxiang Lin’s group (Stanley et al, 2019, ACS Nano). This approach replicated NPC-like dynamics and CP formation, albeit requiring a larger DNA origami cylinder’s diameter compared to previous versions, which raises questions about the robustness of this system as an NPC mimic.

The HS-AFM-LS analysis was extended to NPCs purified from a “ Δ FG” *S. cerevisiae* strain in which ~50% of the FG mass is deleted. This strain had been used in previous work by the senior author Liesbeth Veenhoff’s group (Strawn et al. 2007, NCB; Popken et al, 2015, MBoC). They also analyzed a novel “Nsp1FGx2” strain with ~30% extra FG mass due to duplicated Nsp1 FG regions. The mutations appeared to alter the dynamic fluctuations of FG regions as well as CP structures, though it is not clear what else might be different in these strains. Moreover, whereas both strains show fitness and transport function defects compared to wildtype, Δ FG and Nsp1FGx2 show decreased and increased Kap95 recruitment to the nuclear envelope in vivo, respectively. However, the relationship between the altered phenotypes and the altered dynamics of the permeability barrier observed by HS-AFM-LS is correlative and not mechanistically obvious. The authors also overexpressed various GFP-fused FG regions in vivo. They assert that the translocation of these condensate-forming FG regions into the nucleus as soon as 1 hour after expression is induced supports a dynamic

permeability barrier. However, without a negative control, other interpretations are possible, such as karyopherins mediating the import of FG polypeptides that are not tethered to the NPC via a structural domain.

We carried out the *in vivo* transport assays (**Fig 3**) to answer a relatively simple question: Do scaffold-tethered FG domains within NPCs serve as nucleation sites for free-floating FG domains (NupFGs) to coalesce, mirroring *in vitro* assemblies? How do NupFGs compare to Kap95 (old Fig. 2m) in terms of their enrichment at NPCs?

This question is highly relevant, as several prominent NPC models are largely based on *in vitro* NupFG assemblies, such as macroscopic hydrogels (Frey Science 2006, Frey Cell 2007, Frey EMBO J 2009, Mohr EMBO J 2009, Ader PNAS 2010, Hülsmann Cell 2012, Labokha EMBO J 2013, Schmidt eLife 2015; several reviews in the phase separation field). However, their *in vivo* behavior has not been well studied. To make our intentions clear, we now write:

>>L268: “Next, we investigated whether GFP-fused FG domains expressed *in vivo* (here referred to as NupFGs) exhibit enrichment at NPCs comparable to expressed Kap95, given the FG domains’ known tendency to phase separate⁷⁷ *in vitro* into condensate and aggregate assemblies.”

Furthermore, new Fig. 3 now comprises of old Fig. 2m-o and old Fig. 3d, making this comparison clear.

Unlike *in vitro* NupFG assemblies, none of the tested NupFGs (Nup100FG, hNup153FG , Nup116FG, Nup159FG, Nsp1FG, and Nup60FG) accumulated at the NPC. This included Nup100FG, which we used as a reporter of phase separation in the NPC, given its tendency to form biomolecular condensates *in vivo*. In marked contrast, Kap95-mNeonGreen enriched at NPCs (old Fig. 2m), similar to importin β 1 in mammalian cells (Lowe eLife 2015; Kapinos JCB 2017; Kim Nature 2018; Kalita JCB 2022; etc). Notably, depleting the enriched pool of Kaps compromises NPC barrier integrity (Kapinos JCB 2017). This shows that Kaps have a more natural affinity to concentrate at NPCs, and play a crucial role in reinforcing the permeability barrier. In contrast, NupFGs did not readily concentrate at WT NPCs *in vivo*. Instead, they may traverse the NPC either assisted by Kaps, as noted by the reviewer, or unassisted (we cannot yet distinguish which), suggesting more transient interactions.

Moreover, Nup100FG accumulation still did not occur (new Fig. 3d) in Nsp1FGx2 NPCs where:

- (i) FG repeat concentration was 30% higher than WT
- (ii) Overall central transporter dynamics was slower
- (iii) Kap95-mediated import was defective (new Fig. 3b)

Hence, the absence of FG domain accumulation at WT NPCs suggests that its nanoscopic environment is not conducive for *in vivo* phase separation. Possible reasons might be:

1. mixed FG composition in NPCs does not promote phase separation
2. scaffold constraints i.e., tethered FG domains do not promote phase separation
3. molecular crowding and hindered exchange due to nanoscopic confinement
4. localized interactions with stronger binders such as Kaps.

These reasons have been addressed in the Discussion.

>>L377: “While in vitro NupFG condensates and hydrogels rely on homotypic interactions of single-component FG domains^{28, 29}, the NPC environment is defined by a diverse ensemble of FG domains and Kap-cargo complexes engaging in multifaceted, interconvertible interactions essential for CNT function...These characteristics underscore the context-dependent behavior of the NPC transport barrier, often overlooked in in vitro assemblies, that distinguish it from in vivo NupFG condensates.”

>>L287: “These results suggest that (1) only a subset of FG domains can form condensates in vivo, (2) their presence in both the nucleoplasm and cytoplasm indicates they can traverse the CNT, potentially aided by Kaps, and (3) their lack of enrichment at the nuclear rim, unlike Kap95 (**Fig. 3a**), indicates that FG domains in the CNT do not form the same condensates as observed elsewhere in vivo.”

Indeed, a long-standing question about the physical state of FG repeats in the field is whether they form a compact sieve-like mesh that can be “melted” by permeating karyopherins, or whether FG regions act more like a molecular brush, preventing free diffusion based on entropic excluded volume effects. The authors analyzed in vitro formed FG hydrogel particles by HS-AFM-LS, noting that they are heterogeneous, rigid, and holey, but that the holes are not induced by Kap95. Within the holes, they observed rare fluctuations that could be attributed to brush-like FG region movement like in the purified NPCs. The authors draw a distinction between the “stiff” hydrogel reconstituted in vitro and the dynamic barrier they observe in isolated NPCs. However, FG hydrogel particles are huge compared to the NPC’s central transport channel and it is not clear what a native hydrogel on the NPC scale would behave like under an AFM. This lack of an appropriate positive control weakens the claim that the observed permeability dynamics are different from those expected for a hydrogel.

We thank the reviewer for acknowledging the relevance of our results. While we fully agree that macroscopic FG hydrogels are vastly larger than NPCs and not comparable in scale – and this is a criticism that can be leveled at a large body other work published on such hydrogels - their proposed similarities are widely discussed in the literature, including multiple reviews within the phase separation field. Given that hydrogels form the basis of the NPC selective phase model, we found it essential to compare them with actual NPCs to better define their similarities and differences (Frey Science 2006, Frey Cell 2007, Frey EMBO J 2009, Mohr EMBO J 2009, Ader PNAS 2010, Hülsmann Cell 2012, Labokha EMBO J 2013, Schmidt eLife 2015, Frey Cell 2018, Ng Nat Comm 2021, Ng Nat Comm 2022, Najbauer Nat Comm 2022, Ng Nat Comm 2023).

We do welcome the opportunity to work with nano-hydrogels at the scale of an NPC. However, to our knowledge, no such verifiable hydrogel exists, and generating or characterizing one is beyond the scope of this study. However, many of the droplets of the phase separated FG Nups we form in vivo (see above) are of the same order of magnitude in scale as NPCs, and many are immediately adjacent – suggesting they are at least in a similar environment.

In summary, the study attempts to address a long-standing question about the nature of the CP, initially observed by Unwin and Milligan (1982, JCB). It is not clear what the benefit of the higher temporal millisecond-scale resolution of the HS-AFM-LS methodology is, especially considering that karyopherin-mediated transport events are orders of magnitude faster, typically ~5-10 ms. The key advance of this paper is the “rescue” of CP by introducing Kap95 (Fig.1j-o). However, the improved HS-AFM-LS resolution is not key for interpreting this result and would have been resolved with previous ~100 ms HS-AFM measurement capabilities reported in Sakiyama et al. 2016.

A more important experiment that would confirm the role of Kap95 in the formation of the CP would be (A) a dose-response evaluation, which would involve titrating Kap95 and quantitating the **percent of purified NPCs** and DNA-origami pore mimics with a CP. Similarly, (B) it would be interesting to assess if Kap95 recruitment is indeed a determining factor for CP retention by quantitating the kinetics of CP loss in the NPCs from three *S. cerevisiae* strains (Δ FG, Nsp1FGx2, and wildtype) upon incubation of extracted NPCs in buffer. The rest of the results are mostly descriptive or correlative, without testing clear hypotheses, and thus hard to interpret in the broader context of NPC function. Many experiments are derivative from previous work by the authors.

We thank the reviewer for suggesting these additional experiments, which have helped us clarify and strengthen our conclusions.

(A) We now compare between the cross-sectional height and dynamic behavior of the CP-like feature in NuPODs at 100 nM (new) and 1 μ M Kap95, revealing a clear dose-dependent effect. The CP-like feature is less pronounced and more dynamic at 100 nM Kap95 compared to 1 μ M Kap95. These observations qualitatively mirror the differences between -CP WT NPCs and +CP WT NPCs, reflecting their distinct composition and FG domain arrangement.

>>L333: “Nevertheless, adding Kap95 to NuPODs induced a mobile, CP-like structure with dose-dependent changes in height and dynamics (0.1 μ M versus 1 μ M), qualitatively resembling the behavior when 1 μ M Kap95 was added to -CP WT NPCs (**Fig. 5d-g; Fig. 1j**). This indicates that the NPC transport barrier mechanism is critically dependent on the CNT’s dimensions and geometry (**Fig. 5d-g; Supplementary Fig. 13**).”

(B) We kindly refer the reviewer to our response to R1Q3 above, where a similar question was posed. In summary, we conducted confocal microscopy experiments to ascertain the time-dependent loss/gain of Kap95 at isolated WT NPCs. These results show that the amount of exoKap95 gained, is consistent with the amount of endoKap95 lost, confirming that the recruitment of Kap95 is indeed a key determining factor for CP retention.

We did not conduct the same analysis for Δ FG and Nsp1FGx2 NPCs, as we believe the CP loss/gain analysis is most relevant for resolving WT NPC function and less so for these mutants. Nonetheless, both time-dependent HS-AFM and negative stain EM analyses indicate similar CP loss behaviors in Δ FG and Nsp1FGx2 NPCs (**Supp. Fig. 1**). We appreciate your kind understanding.

Taken together while the technical advances in AFM are impressive, the study only contributes an incremental advance towards answering the question about NPC function and nucleocytoplasmic transport in the context of cell biology. The key question remains: is the CP structure per se functionally relevant for nucleocytoplasmic transport or is it an emergent phenomenon of spatially constrained FG regions and shuttling karyopherins? This paper is therefore more appropriate for a specialized journal.

Reviewer #3:

Remarks to the Author:

The nuclear pore complex (NPC) controls macromolecular movement between the cytoplasm and the nucleoplasm of eukaryotic cells. While the NPC scaffold has been well-defined by structural work over the last few decades, the permeability barrier that occupies and controls permeation through the pore is less well understood. This permeability barrier is generated by a dynamic assembly of intrinsically disordered proteins whose properties are modulated by the presence of various transport factors, most notably Kap95 (Importin beta). The primary emphasis of this paper is the use of high-speed AFM (HS-AFM) to probe the properties of the permeability barrier of isolated yeast NPCs. The central findings are that the central plug (CP) is a dynamic entity that undergoes lateral motion within the pore, that this CP is lost over the course of days, and that the CP feature can be reconstituted by the addition of Kap95. Satisfyingly, this picture is confirmed by a simple model system constructed from DNA origami. These are important new conclusions well-supported by the data. Using kymograph analysis of repeated single AFM linescans, they were able to monitor fluctuations of the permeability barrier surface with ~ 2 ms resolution (instead of ~ 150 ms for a full frame), which yielded autocorrelation times of ~ 20 - 40 ms. This indicates fluctuations on the order of translocation times previously measured at the single molecule level, and therefore consistent with transport through a liquid-like/flexible milieu.

The authors also probed NPCs both with increased (+30%) and decreased (-51%) numbers of FG-repeats. In both cases, these are reported to have overall reduced dynamic signatures, though (i) it is not clear how this conclusion is made since the autocorrelation times vary across the pore diameter and the profiles under the various conditions are not overlaid for direct comparison. (ii) A simple average autocorrelation time might not suffice, unless this is weighted by annular area. (iii) The Kap95 reconstituted NPCs have the most dynamic barrier with an essentially flat autocorrelation time profile across the pore, which is not discussed.

We thank the referee for pointing out this detail, which we needed to make clearer and strengthen in our manuscript to support our claim.

- (i) The dynamic maps have now been replaced with overlaid Tau plots to simplify comparisons of autocorrelation times between WT and mutant NPCs. (see **Fig. 1k**, **Fig. 2f,l** and **Fig. 5g**)
- (ii) To clarify, HS-AFM kymograph measurements sample the dynamic motion within the NPC as a function of radius. From the kymograph, ACFs are generated at each radial position and analyzed for their respective autocorrelation times. Because each pixel in the kymograph has a radial size of ~ 1 nm, Tau plots report the mean and standard deviation of the autocorrelation times captured at each radial position (for $r = m, \dots, -3$ nm, -2 nm, -1 nm, 0 , 1 nm, 2 nm, 3 nm, \dots, n) over many pores. Hence, we believe this analysis eliminates the need for weightage by annular area. Please see revised Supp Fig. 6 for schematic explanation.
- (iii) Yes, because we focused our CP-reconstitution efforts on Kap95, we cannot rule out the involvement of additional nuclear transport factors or other constituents in CP-formation. We now explain:

>>L211: “Although Kap95 alone could restore a CP-like structure, the overall dynamics of this reconstituted structure were notably faster than those of +CP WT NPCs, suggesting that additional nuclear transport factors and other constituents are present in the native CP mass.”

Both FG-repeat mutants exhibit reduced nucleocytoplasmic partitioning, suggesting that the wildtype FG-concentration is optimized. Interestingly, *in vitro* hydrogels made from purified FG-repeat proteins are holey structures, visualized and interpreted as a solidified scaffold containing surface assemblies of FG-polypeptides that enable permeation via a similar mechanism to authentic NPCs. This is an important finding that adds to our understanding of *in vitro* hydrogels formed from FG-Nups. Though somewhat peripheral to the CP story, this nonetheless provides an additional new application of the HS-AFM technology. Finally, the authors examined the distribution of GFP-tagged FG-Nups *in vivo*, finding that these can penetrate into the nucleus but not accumulate at NPCs. The implications of these findings or their relevance to the CP story are unclear. These proteins could penetrate NPCs as monomers, multimers, or in combination with transport receptors. These data, therefore, seem to be best left for a subsequent paper as part of a more complete story. Overall, however, this is a well-written and interesting manuscript that provides important new information about the NPC permeability barrier.

We thank the reviewer for acknowledging the relevance of our results. We kindly refer the reviewer to our response to R2Q1 above, where a similar question was posed.

In their model (Fig. 6), they assume that Kap95 accumulates in the center of the central transporter, which is a reasonable conclusion considering the reformation of the central plug. However, since they have not directly identified where Kap95 is located within the pore, I would give at least a moderate weight to the possibility of an alternate model. For example, one possibility is that Kap95 accumulates at the periphery of the pore (as promulgated in the original Kap-centric model of the Lim group), and this causes a remodeling and increased cohesiveness of the FG-Nups in the center of the pore, thus giving rise to the feature identified as a CP. This is not to argue that there are no Kap95 molecules in the center of the pore, but the density may not be highest here, as the authors assume.

We thank the referee for pointing out these possibilities. In the Discussion we write:

>>L363: “Due to the NPC’s dynamic nature, long-lived Kaps may persist elsewhere in the pore, such as along the NPC scaffold periphery^{15, 84}, or transition between this region and the CP.”

We also illustrate in **Supp. Fig. 14** how a central plug model might enhance selectivity and transport efficiency compared to a central conduit model, based on basic geometric considerations.

Minor comments:

Title – the use of ‘transport’ in the title is awkward; ‘millisecond timescale’ is sufficient. I believe the authors wanted to indicate that the transport occurs on the millisecond timescale, but the usage here is odd. ‘Transport timescale’ would also suffice.

We thank the referee for pointing this out and have modified the title as follows:
“Dynamic Organization of the Nuclear Pore Complex Transport Barrier”

Abstract – ‘elucidate’ is vague; perhaps ‘probe the dynamics of’. The sentence following ‘consequently’ does not follow from the previous statement – this word can be deleted.

Corrected.

Line 70 – The Musser lab, which has extensively examined transport rates, was not referenced.

Done.

Lines 80-86 – The difference between BMCs and hydrogels is not clear. Solidified structures are often observed for condensates and are often considered a natural process of condensate aging. Thus, perhaps FG-Nup hydrogels are simply fast-maturing condensates. Following the view of Mittag and Pappu (2022, Mol Cell), such a fast phase transition could be considered ‘phase separation coupled to percolation’ (PSCP).

We thank the reviewer for raising this point. We fully agree that the difference between BMCs and hydrogels is not clear. We now write:

>>L403: “Yet, another uncertainty lies in how the NPC transport barrier mechanism can be unambiguously distinguished as a putative FG phase³⁵ from within the broader continuum of condensate behaviors, ranging from liquid- to solid-like, hydrogel, and aggregated amyloid states⁹⁷. Additionally, how these phases compare to macroscopic percolated network hydrogels that form independently of phase separation remains unclear⁴². Notably, highly idealized mutations can convert Nup98 from a solid hydrogel to a liquid-state⁹⁸. Hence, greater clarity is needed on these behaviors.”

Line 92 – The model system (yeast) should be indicated here, especially as a contrast with the previous work on *Xenopus* oocytes.

Added: “the budding yeast, *S. cerevisiae*”

Lines 113-120 – This description of HS-AFM is very useful for the non-expert. However, it is not clear whether impulse (force*time) and pN forces refer to a single tapping event or the aggregate of tapping events at a single pixel. The effect of multiple rapid taps would have a cumulative effect. As per lines 120-121 of the supplementary, 80 pixels at 1.875 ms/line would be ~23 μ s/pixel. At 0.5 MHz, this would be ~10 taps/pixel, and due to the tip diameter, taps would be felt even if centered over a neighboring pixel.

The following clarifications were added to the text:

>>L117: “To be precise, the HS-AFM tip oscillates at ~0.5 MHz resonance frequencies with 2 μ s periods, applying forces that are less than 50 pN per tap by attenuating the 2-3 nm free oscillation amplitude by ~10% per tap.”

>>Supp L154: “Line scans were recorded over 80 pixels...in the fast-scan axis at 1.875 ms/line. At this scan speed, the tip spends 0.9375 ms moving from left to right, then returns in another 0.9375 ms along the same line. As standard practice, only the left-to-right data was analyzed to ensure data consistency. At an oscillation frequency, f_c , of ~ 0.5 MHz with $2 \mu\text{s}$ periods, each ~ 1 nm-sized pixel is tapped approximately 6 times with a tip-sample contact time of ~ 200 ns per tap and ~ 0.1 nm lateral spacing between taps due to the continuous lateral scanning motion of the tip (**Supplementary Fig. 7d**).

>>Supp L282: “To assess the impact of tip-sample contact, the mean energy loss per tap of each HS-AFM cantilever oscillation, E , can be calculated using²¹

$$E = \frac{k_c A_{free}^2}{2Q} \left\{ 1 - \left(\frac{A_{real}}{A_{free}} \right)^2 \right\}.$$

For $A_{free} = 3$ nm, the average energy losses are $\sim 16 k_B T$ for lipid bilayer holes, $\sim 6 k_B T$ for bare DNA origami pores and $\sim 6 k_B T$ for -CP WT NPCs, respectively (where k_B is Boltzmann constant, and T is 298 K) (**Supplementary Fig. 7**). For comparison, the repulsive energy barrier associated with the NPC's FG domains is estimated to be $\sim 10 k_B T$ ^{22, 23}. This allows for the HS-AFM to reliably image the “soft” permeability barrier without energetically altering it. The energy transferred from the tip disperses among multiple degrees of freedom into the surrounding buffer^{15, 20}. Intrinsically disordered proteins, with more degrees of freedom than ordered proteins, distributing this energy more efficiently, reducing their susceptibility to tip-sample contact damage²⁴.”

Line 124 – ‘transport’ is not needed here. It is sufficient, clearer, and more accurate to simply state that the isolated NPCs are competent for binding Kaps.

Deleted.

>> L128: “They represent the contracted form of the normal range of NPC diameters, retain an intact configuration of FG domains, and bind Kaps⁶⁷.”

Lines 342 – ‘Nature’ seems too strong a word. Something more specific and less general would be better.

Deleted.

Lines 358-359 – they have not shown that Kap95 binds to the ends of FG domains.

Deleted.

Fig. 1 – Are the scale bars in (b) and (d) accurate? If so, the pore in (b) is about twice the size of the pore in (d).

Yes, -CP pores are more easily zoomed into than +CP pores.

Supplementary:

Lines 119-121 – Clarity on dwell time between lines is needed, as this affects the time

resolution. Also, clarity on whether the scanning is back and forth (Fig. 6b), or only one direction (Fig. 4c) is needed – this also affect time resolution.

Please see our response to your comment on “description of HS-AFM is very useful for the non-expert”.

Fig. S4 – (a) The diameter of the model NPC is over 2-fold less than that of authentic yeast NPCs (acknowledged as ‘miniaturized version’). Considering that FG-Nup density affects permeability barrier properties, how does this influence the simulation results? (d-e) There appears to be a blue torus overlaid onto a grid pattern, suggesting that the simulations do not capture the effect of the NPC scaffold. Clarification is needed.

We kindly refer the reviewer to our response to R1Q1 above, which addresses a similar question. Briefly, we modeled Nsp1 FG domain behavior in a simplified 22 nm-diameter toroidal nanopore (blue) to assess data asynchronicity. The model was intentionally simplified to optimize computational efficiency and was not meant to fully represent the NPC. As a result, the FG domain concentration in this model was 33% higher than in the full NPC.

>>Supp. Fig. 5 Caption: “**Assessing HS-AFM data asynchronicity. a-b**, BD snapshots presenting a side view (a) and top view (b) of Nsp1FG tethered to a 22 nm-diameter toroidal nanopore (blue).

Fig. S5e – I do not understand the meaning of the 95% confidence interval and why this is used to eliminate and raise the baseline. It seems like this would substantially affect the estimated tau values.

The 95% confidence interval (CI) is a statistical threshold used in autocorrelation function analysis that helps distinguish significant correlations from random fluctuations, meaning any coefficients within this interval are likely not statistically significant. Therefore, the CI does not raise the baseline, rather, it excludes insignificant autocorrelation values. This process filters out noise and eliminates unreliable values, ensuring reliable and robust decay time estimations, especially at longer lag times where accuracy diminishes. Without this confidence threshold, random noise could be mistaken for significant correlations, leading to inaccurate decay time estimations. It is calculated using the formula (<https://ch.mathworks.com/help/signal/ug/confidence-intervals-for-sample-autocorrelation.html>):

$$CI = \pm 1.96/\sqrt{N}$$

where N is the total number of data points.

Fig. S10b – It is not clear that the identified event is actually a fusion event, since there is no intermediate showing a partial outline of both puncta. This appears as an ‘apparent fusion’, as technically one of the puncta could disassemble and absorb the monomers from the other.

We have deleted this section as it is unessential and the time resolution was not good enough to capture the intermediate steps.

Line 910 – Is “CT” supposed to be “CNT”? Yes, thank you.

Many thanks for the reviewers' feedback. These have allowed us to re-focus the description of our work, to emphasize that this study represents a major breakthrough in our understanding of the NPC by directly visualizing, at millisecond resolution, the dynamic behavior of its intrinsically disordered FG-repeat domains in response to nuclear transport factors. These findings challenge the prevailing notion of a barrier governed solely by FG repeat behavior, and instead reveal a transport mechanism governed by rapid, regulated fluctuations in FG-network organization, including the formation of a dynamic central plug, whose nature has eluded description for decades but is now revealed as the dynamic product of FG repeat / transport factor interactions.

Reviewer #1 (Remarks to the Author):

The authors have significantly improved their manuscript, adding essential experiments that strengthen their conclusions. They also clarified some useful points. All my wariness was extensively addressed. The overall manuscript is strong and very interesting for a large scientific community. I support its publication at Nature Cell Biology.

We thank the reviewer for your support. We are pleased that the revisions addressed your concerns.

Reviewer #2 (Remarks to the Author):

1. Kozai, Fernandez-Martinez, et al., are to be commended for adding further experimental evidence during the revision of this manuscript. Specifically, the confocal fluorescence microscopy data shown in Supplemental Figure 4 illustrate the gradual leeching of endogenous Kap95 from the central plug (CP) and the subsequent deposition of exogenous Kap95 into empty "-CP" NPCs to reform the CP. When considered alongside the experiments in Figure 3A, which demonstrate Kap95 accumulation at NPCs containing twice the number of FG repeats within Nsp1 (Nsp1FGx2), this new experiment offers a compelling quantitative analysis of the dynamic occupancy of Kap95 at the CP. Accordingly, it merits elevation into the main text Figure 3.

Thank you.

2. However, while valuable, this experiment does not adequately address whether Kap95 concentration plays a role in CP formation. The authors compare the original NuPOD data collected in the presence of 1 μ M Kap95, against a new limiting concentration of 100 nM Kap95. In this condition, HS-AFM indicates no CP formation, while TEM reveals a diffuse CP structure of similar lateral dimensions. In my view, the minimalistic NuPOD system represents the weakest model to elucidate concentration dependence, incorporating only the FG domain of Nsp1.

Both the average cross-section height analysis (Fig. 5f) and the Tau plot (Fig. 5g) show that the formation of a CP-like feature scales with different Kap95 concentration (0, 0.1 μM and 1 μM). Furthermore, the CP-like feature in 0.1 μM Kap95 NuPODs is evident when contrasted against Nsp1FG-NuPODs alone (right; Supp Fig. 12B; Fig. 5c,d). For better clarity, we have updated the selection of 0.1 μM Kap95 Nsp1-NuPODs with examples whose height profiles represent the average behavior shown in Fig. 5f (right; Supp Fig. 12C).

We have also shown that this dose-dependent behavior is not limited to one kind of NuPOD, as it is consistent with CP-like formation by importin β 1 in Nup62-NuPODs (Fig. 3A-E and Fig. S7A-C; Feng Sci. Adv. 2024).

From new Supp Fig. 13

3. This experiment would be more effectively conducted with isolated *S. cerevisiae* NPCs, incubated in buffer for 72hrs to ensure a homogenous -CP sample, followed by addition of exogenous Kap95 across a range of concentrations from 0 to 1 μM . Unfortunately, a single ambiguous data point at a concentration that does not produce a consistent outcome between AFM and TEM analyses fails to clarify whether CP formation scales linearly with increasing Kap95 concentration or occurs abruptly at a threshold level.

We agree that such experiments in *S. cerevisiae* NPCs would significantly enhance our studies, and so this isolated NPC HS-AFM experiment has now been performed exactly as requested by R2, incubating -CP WT NPCs in 0, 0.1 μM , 0.5 μM and 1.0 μM exogenous Kap95. Crucially, our results confirm the scaling relationship between Kap95 concentration and CP formation, directly addressing this reviewer's core criticism. Moreover at 1.0 μM , Kap95 effectively replicated the vertical range of the +CP WT NPC transport barrier.

>> This data is included now in Fig. 1g. See also accompanying figure to Q5 below. \

>> L207: "Supporting this observation, exogenous Kap95 (0, 0.1 μM , 0.5 μM and 1.0 μM) restored a dose-dependent CP-like structure^{76, 77} in -CP WT NPCs, highlighting a scaling relationship between Kap95 concentration and CP formation (Fig. 1g, j-l; Supplementary Fig. 7). At 1.0 μM , Kap95 effectively replicated the qualitative behavior and vertical range of the +CP WT NPC transport barrier. However, the overall dynamics of this reconstituted structure were notably faster than those of +CP WT NPCs (Fig. 1h), suggesting that additional nuclear transport factors and other constituents contribute to the native CP mass."

>> Updated Supplementary Fig. 7

4. I wish to emphasize that the technical advancements in AFM analysis presented here are impressive and certainly merit publication in a specialized journal. Nonetheless, the primary functional insights from this manuscript are that the CP (1) moves laterally within the NPC's central transport channel, and (2) is dynamically comprised at least in part with transport

factors from the beta-karyopherin family. Since the CP has been recognized in EM imaging of the NPC since Unwin and Milligan (JCB, 1982) and has been demonstrated via AFM by Stoffler et al. (JMB, 2003) to leech from the central channel over time, the novel contribution of this manuscript is the demonstration that Kap95 alone can form a CP analog in vitro. Unfortunately, these findings do not align with the manuscript's title, "Dynamic Organization of the Nuclear Pore Complex Transport Barrier," as the paper provides little additional insight into the nature of the diffusion barrier, focusing instead on a region that other studies suggest is largely inert during karyopherin mediated transport.

We thank the reviewer for recognizing the technical advances in our study. Respectfully, we wish to clarify that our manuscript advances a contemporary understanding of the CP far beyond prior static structural observations, such as those made by Unwin and Milligan (JCB, 1982) and Stoffler et al. (JMB, 2003). While those studies provided valuable morphological insights, neither they nor any other studies addressed the CPs dynamic properties, nor its composition, nor identity, nor its remodelling of FG Nup behavior, nor distinguished its behavior from FG biomolecular condensates, brushes and FG hydrogels, nor established an understanding of its functional relevance to the NPC transport barrier.

Quoting Rush et al. Front. Cell Dev. 2023,

"Older EM data have presented a structural barrier residing within the center of the NPC known as the "central plug," and it was thought that passaging cargo had to either move around or interact with this plug (Talcott and Moore, 1999). This finding generated what is known as the "plug model" of the NPC...however, the composition of the plug and the apparent mobility relative to the scaffold of the NPC were unclear (Stoffler et al., 2003). More recent models produced by these techniques tend to just omit the inside of the pore, leaving approximately a 60-nm-diameter gap where the FG-Nups would otherwise be displayed (Brohawn et al., 2009; Von Appen et al., 2015; Schuller et al., 2021; Tai et al., 2023). This is not just a gap in depictions of the pore but a gap in our knowledge about the pore because even though it is now certain that the pore is filled with FG-Nups, it remains uncertain exactly how they are composed."

To reiterate, the main breakthroughs from this study include:

1. resolving the in situ NPC transport barrier at millisecond timescales - the timescale at which transport occurs. (Fig. 1)
2. revealing that dynamic FG Nup fluctuations are remodelled by a pool of Kaps - probably together with additional nuclear transport factors and constituents - form the CP. (Fig. 1)
3. revealing how in vivo NPC transport function is influenced by FG density and dynamics, shown through systematic comparisons between wild type and mutant NPCs. (Fig. 2 & 3)
4. clarifying the behavior of the transport barrier inside NPCs relative to FG biomolecular condensates and FG hydrogels. (Figs. 3 & 4)
5. reconstituted dynamic FG Nup fluctuations and CP formation (owing to Kap-binding) in DNA origami NPC mimics (NuPODs). (Fig. 5)
6. showing strong complementarity with the recent findings by Sau et al. (Nature 2025). See below.

To be clear, we articulate these points in the Intro para 3 (indexed as above):

>> L95-113: “Here, we used high-speed atomic force microscopy (HS-AFM)⁵⁶ and line scanning (HS-AFM-LS)⁵⁷, complemented by orthogonal in vivo and in vitro studies, **(1)** to elucidate the in situ dynamic organization of the budding yeast *S. cerevisiae* NPC transport barrier with a ~1 ms temporal resolution, closely matching transport timescales¹⁷⁻¹⁹. Previously, we used HS-AFM to visualize dynamic FG domain behavior within NPCs using spread *X. laevis* oocyte nuclear envelopes^{58, 59}. In this work, we investigated the CP’s identity, the impact of transport barrier dynamics on in vivo NPC function and corroborate these attributes using in vitro FG domain assemblies and in vivo assays. **(2)** We show that Kaps remodel FG domain behavior by stabilizing their dynamic fluctuations at the pore center, mediating the formation of a mobile nanosized cluster that accounts for the CP. **(3)** Mutant NPCs with increased FG content showed reduced barrier dynamics and impaired NLS-cargo transport in vivo, despite elevated Kap95 enrichment at the pore, suggesting its sensitivity to FG composition and an overtightening the transport barrier. In contrast, **(4)** overexpressed nucleoporin FG domains (NupFGs) did not readily concentrate at NPCs in vivo, indicating poor coalescence with the transport barrier. To benchmark our findings to in vitro NPC models, we found that in vitro macroscopic FG hydrogels were structurally static, differing markedly from the highly elaborate, dynamic nanostructures within the NPC. Rather, **(5)** NPC-like dynamics, including the formation of CP-like nanoclusters, was successfully replicated in DNA origami nanopores bearing scaffold-tethered FG domains⁶⁰⁻⁶⁴.”

>> To better frame the work, a new title is proposed: “Karyopherins Remodel the Dynamic Organization of the Nuclear Pore Complex Transport Barrier”.

5. This brings me to a significant caveat of this study, single-molecule digitonin-permeabilized NPC studies from the Musser group (Chowdhury et al., NCB, 2022; and very recently Sau et al., Nature, 2025) are not sufficiently discussed by the authors. These studies reveal that Kapbeta1 (the human homolog of Kap95), Kapbeta2, and Kapalpha-Kapbeta1 complexes utilize overlapping transport pathways through the NPC diffusion barrier. Importantly, they show that the central channel comprises three distinct annular zones with variable transport efficiency, with the CP localized in a central null zone where few Kapbeta1 transport events occur. A revised manuscript should therefore critically evaluate how these peripheral transport routes might impact the proposed functional role of the CP. For instance, if the CP does not facilitate active transport, might it function instead as a reservoir of beta-karyopherins ready for cargo binding?

We thank the reviewer for raising the important point regarding the new paper by Sau et al. (Nature 2025), which we could not cite in our previous submission as it had not yet been published at the time.

Nevertheless, we are genuinely excited by this paper, as it shows a strong complementarity with our work. While Sau et al. revealed Kaps traversing an annular region of the NPC (right, bottom panel, purple), with no movement along the central axis; it did not capture the transport barrier itself, a feature our study directly captures in detail. Here, our results independently show that the CP creates this exclusion zone (Fig. 6a; right, middle panel, dark blue) in a manner that scales with Kap95 concentration (Fig. 1g; right top panel; see also R2Q3), thereby suggesting that transport occurs through the surrounding dynamic FG domains (Fig. 6a; right, middle panel, light blue). As their experiments were performed in permeabilized cells, the presence of strongly bound endogenous Kaps in the CP likely also blocked exogenous Kaps from accessing the central exclusion zone (personal communication, S.M. Musser), further supporting this interpretation. Together, these studies generate orthogonal yet complementary data that collectively deepen our understanding of the transport mechanism. [Note: The right figure illustrates how these datasets and models align.]

Kozai et al. Fig. 1g (top) and Fig. 6a (middle) vs. Sau et al. Fig. 4f (bottom)

We have now modified the text to highlight this striking complementarity:

>> L77: “Despite having mapped the core scaffold structure⁴⁻⁹ and the trajectories of individual Kap-cargo complexes^{17, 20} with nanometer precision, the organization of the NPC transport barrier remains obscure due to its dynamic and disordered nature²²⁻²⁴.”

>> L366: “Our model strongly complements 3D MINFLUX tracking of Kaps through individual NPCs in permeabilized cells, which revealed transport-active Kaps traversing an annular region of the NPC but not along its central transport axis²⁰. While the approach by Sau et al. provided single molecule-resolution tracking, it did not directly resolve the structure or dynamics of the NPC transport barrier. In contrast, our study directly addresses these features in detail and offers a compelling mechanistic explanation wherein the CP – comprising of a highly persistent pool of endogenous Kaps - forms a sterically restrictive structure that establishes the central exclusion zone. In parallel, selective transport proceeds through the surrounding annular region of more dynamic FG domains. Given that their experiments were performed in permeabilized cells^{17, 20}, the presence of this CP-bound pool of endogenous Kaps likely prevented exogenous Kaps from accessing the central exclusion zone. These findings underscore the need to account for endogenous Kaps in future experiments. The CP may also serve additional roles, such as acting as a dynamic reservoir of Kaps that exchange with other regions of the NPC during large structural transitions such as pore dilation. Conversely, dilated NPCs may recruit additional Kaps to the CP to compensate for a 20% reduction in FG domain

concentration relative to constricted NPCs^{5,8,9,90} (Supplementary Table 2), thereby stabilizing the central exclusion zone and maintaining transport barrier function.”

6. Taken together, this revision does not adequately address the functional relevance of the CP on NPC function, particularly in the context of karyopherin-mediated nucleocytoplasmic transport. Thus, I am sorry to conclude that my initial impression remains unchanged, this manuscript is better suited for publication in a specialized journal.

Our study provides the first real-time visualization of FG Nup remodeling within intact NPCs, revealing dynamic features and composition of the transport barrier, identifying the CP, and defining its functional relevance. It further distinguishes CP behavior from that of bulk FG assemblies. No prior study has combined such an extensive set of orthogonal approaches to dissect NPC transport dynamics, avoiding confirmation bias. Together with the timely complementation with the Sau et al. paper, our data provide a coherent picture of NPC barrier organization, answering longstanding questions about the NPC, breaking new ground. We are confident that our research holds broad appeal and is highly newsworthy, making the work highly suitable for *Nature Cell Biology*.

Minor Comment:

7. Line 287: The statement, “(3) their lack of enrichment at the nuclear rim, unlike Kap95 (Fig. 3a), indicates that FG domains in the CNT do not form the same condensates as observed elsewhere in vivo,” is unclear. Specifically, the phrase “as observed elsewhere in vivo” is ambiguous, does it refer to the GFP-FG domain fusions described in the current study, or condensates observed during either oocyte NPC assembly, or within aberrant NPC assemblies induced by Torsin-A/Br11/Brr6 mutations? Given that comparisons between artificial GFP-FG domain fusions and naturally occurring FG-Nup condensates are inherently problematic, I recommend omitting or revising this statement for clarity.

Now corrected as follows:

>> L289: “...indicates that FG domains in the CNT do not form the same condensates as visible elsewhere in the same cell.”

Reviewer #3 (Remarks to the Author):

1. This manuscript is much improved and reads much better. In particular, the text is much more balanced and provides a complete and comprehensive story of how the current work fits in with prior data. It also portrays the complexity and challenges of the nuclear pore permeability barrier, and yet, nonetheless, demonstrates how they have made a significant advance. As described below, there are a few relatively minor issues that need attention, but these are readily addressable.

We thank the reviewer for your support. We hope the below revisions/explanations fully address your concerns.

2. Supplementary Fig. 4 is new and uses confocal fluorescence microscopy to demonstrate the release and re-binding of Kap95 to isolated NPCs. Unfortunately, the data as presented is not convincing (the entire figure is poor resolution, which doesn't help). A key problem is that there appears to be significant non-specific binding raising questions about the quantification. There is no control slide in the absence of NPCs. In (a), it is entirely unclear what we should consider as fluorescence arising from a pore. In (d) at 72 h, the merge image suggests a significant pixel drift or mis-aligned images. It is not clear why mCherry- or GFP-tagged NPCs or Kap95-GFP cannot be imaged directly. The ratio and overall signals from pores and the bound Kaps fluctuate widely, as evident from the images and their violin plots.

Most certainly, the binding of Kaps to NPCs occurs on the tens of seconds to minutes timescale, so the 20 h time constant from (f) is extremely misleading. This raises concerns about the similar 20 h time constant in (c). Due to their time resolution, they are essentially fitting two points to the curve. In contrast, Kap95 binding to Nup100 hydrogels (Sup Fig. 12i) occurs on the minute timescale. They have not described the criteria for scoring a CP as present or not in their AFM experiments. Well-done fluorescence experiments could provide some nuance here, namely is the plug lost all at once (loss of a large object such as a large cargo) or is it lost gradually (stepwise loss of individual proteins). Nonetheless, I don't find Supp Fig. 4 as necessary for the manuscript.

We apologise for the misunderstanding. We recognize that how the data was plot could have been misinterpreted as a real-time kinetic measurement (such as Supp Fig. 11i), which it is not. Rather, it is an ensemble-level immunofluorescence assay that is read out under different conditions. Each x-axis "condition" is the duration of NPC pre-incubation in buffer (0, 24, 48, 72 hr; prior to the addition of fluorescently-labelled antibodies). Subsequently, Supp Fig. 4b (new Supp Fig. 6b) shows the extent of endogenous Kap depletion following each incubation time point, which we fitted using the equation on Supp P16L476, yielding a half-life ($t_{1/2}$) of 20.1 hrs, whose value is consistent with:

1. Previous surface plasmon resonance (SPR)-measured values (of Kap β 1 vs. various FG Nups; Kapinos Biophys. J. 2014).
2. Mass spec analysis showing reduction in Kap95 (Fig. 1i)
3. HS-AFM analysis of WT NPCs (L150: "60% of isolated NPCs lacked detectable CPs after 72 hours in buffer ...")

On the other hand, Supp Fig. 4d (new Supp Fig. 6d) demonstrates the capacity for NPCs to rebind exogenous Kap95 following endogenous Kap95 depletion at the same pre-incubation time points (0, 24, 48, 72 hr). The dashed curve in Supp Fig. 4d is not a fit *per se*, but a model calculation that applies $t_{1/2} = 20.1$ hrs using the equation in Supp P16L486 to predict the rebinding behavior of exogenous Kap95. The close agreement between the dashed curve and experimental data in Supp Fig. 4d reflects the capacity for NPCs to rebind exogenous Kap95 following endogenous Kap95 depletion. Taken together, our analysis is consistent with a slow apparent dissociation rate of the strongly bound, CP-forming Kap95 fraction in the NPC.

For details, see section “Depletion and rebinding of Kap95 at isolated WT NPCs” in the Supplemental Information. In the main text, we now write:

>> L195: “In contrast, more than half of this long-lived fraction of endogenous Kap95, along with its adaptor Kap60, was displaced from -CP WT NPCs (Fig. 1i), following a slow apparent dissociation rate^{15, 50} (Supplementary Fig. 6).”

New Supp. Fig. 6

Given the wealth of orthogonal data showing Kap95 rebinding, we are indeed willing to drop Supp Fig. 4 (new Supp Fig. 6) if required. Still, for completeness and thoroughness, we have revised the figure for greater clarity, by:

1. including the control images of non-specific binding (new Supp Fig. 6a,c)
2. replacing “time” x-axis label with “condition”
3. distinguishing the data from kinetic measurements by combining old Supp Fig. 4c,f with new Supp Fig. 6b,d
4. removing the black data points in Supp Fig. 6d to clarify that it represents a curve overlay, not a curve fit.
5. explicitly writing “background” above the dotted horizontal line in each plot to make it more obvious
6. renaming the Supp section “Depletion and rebinding of Kap95 at isolated NPCs” (Supp P14L431)
7. including a description of how non-specific binding was assessed (Supp P15L443 and P15L456)
8. revising the Supp Fig. 6 caption (Supp P37L932: “Close agreement between the overlaid dashed curve and experimental data indicates the NPC’s capacity to rebind Kap95-FLUX680 following endoKap95 depletion, consistent with the slow apparent dissociation rate of the CP-forming Kap95 fraction.”

Related to the quality of the images we would like to clarify that the signal corresponds to clusters of NPCs rather than individual pores and that we used dye-labelled antibodies against eGFP and mCherry as a necessary compromise to achieve stable visualization as direct fluorescence suffered from rapid photobleaching.

3. In Fig. 6, they provide a complex model with substantial information that is not directly addressed by their data. Much of this information comes from a large body of work in the field. But the location/distribution of Kaps and those Kaps that are slow and fast moving is not known, and this is pure speculation. Likewise, the bidirectional transport through 8 peripheral channels (Sup Fig. 14) is speculation. Why do they assume transport through channels (Sup Fig. 14) rather than through an annulus (Fig. 6)? More description is needed about the motivation for these speculative points.

We thank the reviewer for pointing this out and apologize for the overreach. Fig. 6 has now been significantly simplified to stay true to our experimental findings and Supp Fig. 14 has been deleted.

New Fig. 6

Minor comments:

4. In at least a few locations at first mention, the authors should indicate that the eight-fold symmetry is a '**rotational symmetry**' (e.g., line 134 and in the Fig. 1 caption).

>> Corrected L134: "The overall size and eight-fold **rotational symmetry structure** of +CP WT NPCs were also analyzed using negative stain electron microscopy..."

5. Lines 254-255 – This sentence is confusing. How does increased kinetic accessibility lead to decreased avidity?

>> Corrected L254: "One possibility is that Δ FG NPCs may be more kinetically accessible to Kap95 due to fewer FG repeats, **resulting from** ~~in~~ diminished avidity."

6. Lines 268-271 – experimental references are missing for the statement at the end of the sentence. Ref. 77 is published (PNAS).

>> Corrected L267: "Next, we investigated whether GFP-fused FG domains expressed in vivo (here referred to as NupFGs) exhibit enrichment at NPCs comparable to expressed Kap95, given the FG domains' known tendency to phase separate⁷⁹ in vitro into condensates^{30, 31} and aggregate assemblies³²⁻³⁷."

7. Fig. 1 – the axes in (f) appear to be labeled as -80°C and 4°C. What does an off-axis point imply – an intermediate temperature? This panel is small and the gray dots are almost invisible. While the color scheme/key in (k) (and in similar figure panels in the manuscript) is clear on my screen, it is not clear in my printout. Probably worse for color-blind individuals.

We apologize because our labeling might have been confusing. "-80°C" identifies NPCs that

have been stored frozen for 5 days at -80°C , have not undergone dilution of their CP and are thus +CP. " 4°C " identifies the same NPC sample that have been stored at 4°C also for 5 days, getting their CP diluted and becoming -CP NPCs. The plot compares the relative fold enrichment of each treatment when compared to the original +CP NPCs sample processed immediately after being isolated. Thus, an off-axis point would identify a protein showing a change in their relative amounts between +CP and -CP NPCs. To clarify this, we have now modified the plot labeling, using "+CP" and "-CP" instead of temperatures to label the axes and maintain a coherence with the rest of figure nomenclature, and modified the figure legend as follows to better explain the data shown in this plot:

>> Fig. 1i caption: "Relative enrichment of proteins associated with affinity-purified isolated yeast NPCs incubated at 4°C for 5 days (-CP) or snap-frozen in liquid nitrogen and stored at -80°C for 5 days (+CP) as determined by label-free mass spectrometry quantification. Proteins showing 2 fold-higher depletion in -CP NPCs are shown in blue (proteasome components not labelled), Nups in orange. N = 2."

Additionally, as a reference we have added a subpanel showing the estimated copy number of Kap95 in -CP and +CP NPCs from this experiment.

>> L194: "... +CP WT NPCs contain approximately 15 to 20 Kap95 molecules, the primary nuclear import factor⁷... over half of this long-lived endogenous Kap95 fraction, along with its adaptor Kap60, was displaced from -CP WT NPCs (Fig. 1i), consistent with a slow apparent dissociation rate^{15, 50}.."

We also thank the reviewer for pointing out the issue with the size of the panel in printing. We have tried to increase both the font and the dots size to make them more obvious. We have also double-checked and the color palette used in the figure should be colorblind safe.

8. Fig. 3d – the four sample images above the figure to indicate the scoring are not mutually exclusive. What if a condensate is observed both inside and outside a nucleus?

To quantify the localization of GFP-Nup100FG condensates, each individual condensate was assigned to a single, mutually exclusive category. In cells with multiple condensates, each condensate was scored independently and categorized accordingly, regardless of how many were present in the same cell.

>> Fig. 3d caption, L810: "Condensates were classified into mutually exclusive categories; in cells with multiple condensates, each was scored independently."

9. Fig. 6b – the phrase ' $>>1\ \mu\text{m}$ ' is extreme. Condensate/hydrogel size is determined by the amount of material added and structures on the 100 nm scale are certainly feasible.

Deleted.

10. Fig. 5d,e and Sup Fig. 8 – the time between when Kap95 was added and when the images were obtained should be indicated.

>> Supp P5L146: "For central plug reconstitution experiments, -CP WT NPCs were monitored *in situ* and in real time by HS-AFM while Kap95 was added to the imaging buffer, achieving a final concentration of 0.1 μM , 0.5 μM or 1 μM Kap95. The images were obtained 30 min after the addition of Kap95."

>> Supp P7L195: "Dose-dependent binding was examined by incubating NuPODs in 0.1 μM and 1 μM Kap95, respectively, followed by rinsing prior to HS-AFM analysis and negative stain TEM. The images were obtained 30 min after the addition of Kap95."